# European tree cover during the Holocene reconstructed from pollen records

Luke Sweeney[1,2,*], Sandy P. Harrison[1,2], Marc Vander Linden[3]

1: Geography and Environmental Science, School of Archaeology, Geography and Environmental Science (SAGES), University of Reading, Whiteknights, Reading, RG6 6AH, UK. Email: l.sweeney@pgr.reading.ac.uk

2: Leverhulme Centre for Wildfires, Environment and Society, Imperial College London, South Kensington, London, SW7 2BW, UK. Email: s.p.harrison@reading.ac.uk

3: Institute for the Modelling of Socio-Environmental Transitions, Bournemouth University, Christchurch House, Talbot Campus, Poole, BH 12 5BB, UK. Email: mvanderlinden@bournemouth.ac.uk

*Correspondence to*: Luke Sweeney (l.sweeney@pgr.reading.ac.uk)

## Abstract

Changes in tree cover influence many aspects of the earth system. Recent regional changes in tree cover, as documented by remote-sensed observations, are insufficient to capture the response to large climate changes or to differentiate the impacts of human activities from natural drivers. Pollen records provide an opportunity to examine the causes of changes in tree cover in response to large climate changes in the past and during periods when human influence was less important than today. Here we reconstruct changes in tree cover in Europe through the Holocene using fossil pollen records, using the modelled relationship between observed modern tree cover and modern pollen samples. At a pan-European scale, tree cover is low at the beginning of the Holocene but increases rapidly during the early Holocene and is maximal at ca. 6,500 cal. BP, after which tree cover declines to present-day levels. The rapidity of the post-glacial increase in tree cover and the timing and length of maximum tree cover varies regionally, reflecting differences in climate trajectories during the early and mid-Holocene. The nature of the subsequent reduction in tree cover also varies, which may be due to differences in climate but may also reflect different degrees of human influence. The reconstructed patterns of change in tree cover are similar to those shown by previous reconstructions. Our approach is relatively simple, only requires readily available data and could therefore be applied to reconstruct tree cover globally.

## 1. Introduction

Tree cover in Europe has been expanding in recent decades (FAO, 2020; Turubanova et al., 2023), with potential implications for land-atmosphere energy exchanges, water and carbon cycles, and ultimately local and global climates (Bonan, 2008; Alkama and Cescatti, 2016). Changes in tree cover may reflect forest or woodland expansion, or the presence of trees in vegetation mosaics and urban settings. Tree cover both affects and is affected by the environment (Moyes et al., 2015; Abis and Brovkin, 2017) and this two-way relationship leads to complex interactions between both. For example, deforestation in the Amazon has been linked to changes in weather patterns, which subsequently affect the moisture available for rainforest maintenance (Staal et al., 2018; Leite-Filho et al., 2020). Similarly, increases in tree cover resulting from farm abandonment has been shown to change fire frequency in the Iberian Peninsula (Moreira et al., 2001; Viedma et al., 2015), which in turn affects vegetation cover and structure and results in further changes to the fire regime. Satellite data can be used to assess the environmental controls on tree cover but, since they only cover the past 20–30 years, are insufficient to look at the response to longer term changes in climate or relationships when the human influence of land-use and vegetation cover was less ubiquitous.

Pollen from sedimentary sequences provides a record of past vegetation changes. The relative abundance of arboreal pollen has often been used to infer changes in tree abundance at a site (e.g. Thorley, 1981; Eastwood et al. 1999; Gil-Romera et al., 2008). The relationship between pollen abundances and vegetation cover is not straightforward however, because it is influenced by differences between species in pollen productivity and transportability, and site characteristics that affect the pollen source area, such as basin size and type (Bradshaw and Webb, 1985; Prentice and Webb, 1986; Prentice, 1988; Sugita, 1993).

Several different techniques have been applied to reconstruct regional and sub-regional vegetation in Europe using pollen, including biomization/pseudobiomization (e.g. Fyfe et al., 2015; Binney et al., 2017) and the application of MAT using plant functional types (e.g. Davis et al., 2014). Other studies have made reconstructions combining different approaches (e.g. Roberts et al., 2018) or by combining pollen-based reconstructions with simulated potential vegetation (Pirzamanbein et al., 2014). However, the most recent quantitative pan-European pollen-based reconstructions of Holocene vegetation changes have been made using the REVEALS approach (Sugita, 2007b, a) or the Modern Analogue Technique (MAT) (Overpeck et al., 1985; Guiot, 1990; Jackson and Williams, 2004; Zanon et al. 2018). The REVEALS method calculates regional vegetation cover based on modelled relationships between pollen abundance, estimated differences in species level pollen productivity and pollen transport, and differences in site characteristics. Initially used at individual sites or small regions (e.g. Gaillard et al., 2010; Nielsen et al., 2012; Marquer et al., 2014), REVEALS was first applied at a pan-European scale

by Trondman et al. (2015) and later extended with additional sites, taxa and an improved temporal resolution by Githumbi et al. (2022). The most recent analysis by Serge et al. (2023), is based on 1607 records for 500-year intervals before 700 cal. BP and for the subsequent intervals of 700–350 cal. BP, 350–100 cal. BP and 100 cal. BP– present. They tested the impact of including 15 additional taxa (total n=46) on the vegetation reconstructions, producing maps of landcover and species abundance at record-containing 1º grid cells. In contrast, the MAT approach reconstructs past vegetation based on identifying modern analogues of fossil pollen assemblages, on the assumption that samples found in the fossil record that share a similar composition to those found in present-day pollen assemblages will have similar vegetation. Zanon et al. (2018) applied MAT to 2,526 individual fossil pollen samples from Europe to generate interpolated maps at 250-year intervals at 5 arc-minute resolution through the Holocene.

Each of these approaches presents challenges. The REVEALS approach requires, and is sensitive to, estimates of relative pollen productivity (RPP) and pollen fall speeds (FS) for individual species (Bunting and Farrell, 2022; Githumbi et al., 2022; Serge et al., 2023). Landscape-level reconstructions are problematic if RPP information is not available for relatively common taxa (Harrison et al., 2020). RPP values have been estimated for common taxa in Europe and China, and there are a limited number of studies from North America (see e.g. Wieczorek and Herzschuh, 2020). Studies have been conducted for some ecosystems in South America and Africa, but these only provide RPPs for a very limited number of taxa (e.g. Duffin & Bunting, 2008; Whitney et al, 2018; Gaillard et al., 2021; Hill et al., 2021; Tabares et al., 2021; Piraquive Bermúdez et al., 2022) and even this level of information is not readily available for other regions. The MAT technique requires a large modern pollen data set for training purposes, but such data sets are now available for all regions of the world. However, the application of MAT involves a number of arbitrary decisions including the choice of analogue threshold (i.e. how similar modern and fossil assemblages must be to be considered analogous), and the number of analogues used (Jackson and Williams, 2004). Techniques designed to minimise the number of samples for which no analogues are found, such as grouping species into plant functional types (PFTs) (see Davis et al., 2003), introduce further uncertainties since the allocation of pollen taxa to PFTs is often ambiguous (Zanon et al., 2018).

In this paper, we develop a simple calibration approach to derive estimates of tree cover at individual sites across Europe, capitalising on an extensive modern pollen data set, new remote-sensed data on tree cover, harmonised age models, and improved information about basin size for fossil records from Europe. We evaluate how well this method reconstructs modern tree cover compared to existing methods. We then reconstruct pan-European changes in tree cover through the Holocene and compare these reconstructions with existing reconstructions.

## 2. Methods

Tree cover during the Holocene was reconstructed by applying a statistical model describing modern tree cover to fossil pollen records from individual sites. There were three steps: 1) selection and treatment of data; 2) development of the predictive model relating modern pollen data to modern tree cover; and 3) application of this model to fossil pollen records (Fig. 1). All analyses were performed using the R Statistical Software (v4.3.3) (R Core Team, 2024).

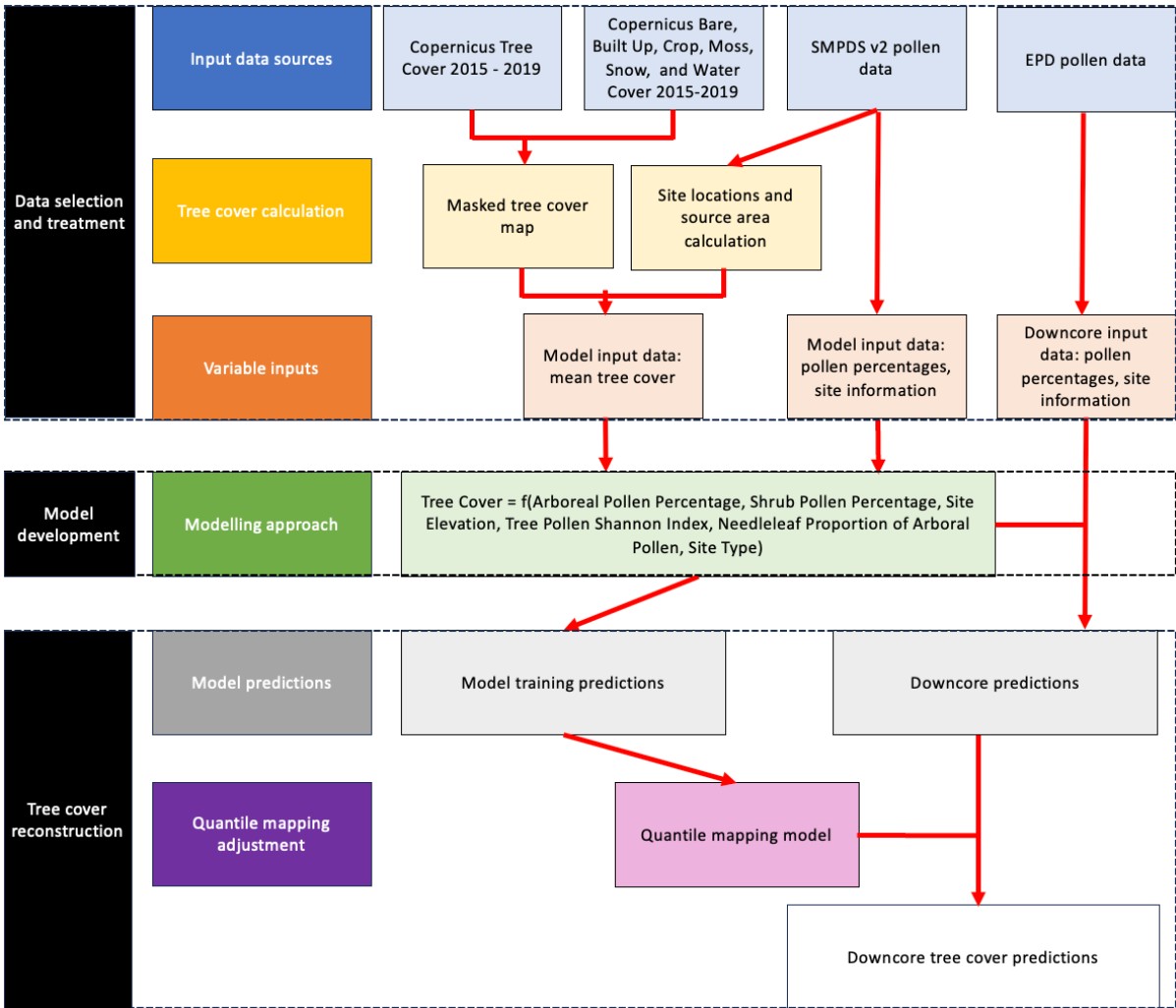

**Figure 1: Methodology to reconstruct European tree cover during the Holocene**

### 2.1 Selection and treatment of data

We used the Copernicus Dynamic Land Cover Dataset to source information on tree cover. This data set provides percentage land cover estimates for individual land-cover classes at a spatial resolution of 100m. We used the land-cover class designated as forest/tree cover, which at this spatial resolution can be regarded as an estimate of actual tree cover. A composite map of modern tree cover for the region 12°W to 45°E and 34–73°N was generated by averaging annual percentage forest/tree cover data from

Copernicus annual land cover maps from 2015 to 2019 (Buchhorn et al., 2020a, e, d, c, b), after removing cells dominated (> 50%) by other land-cover classes, which include bare ground, built up areas, moss or lichen, permanent water, snow, and crops (Fig. 2A). However, the Copernicus maps do not distinguish between natural forests and plantations and so the tree cover target may include planted species. This modern tree cover map has a resolution of 100m.

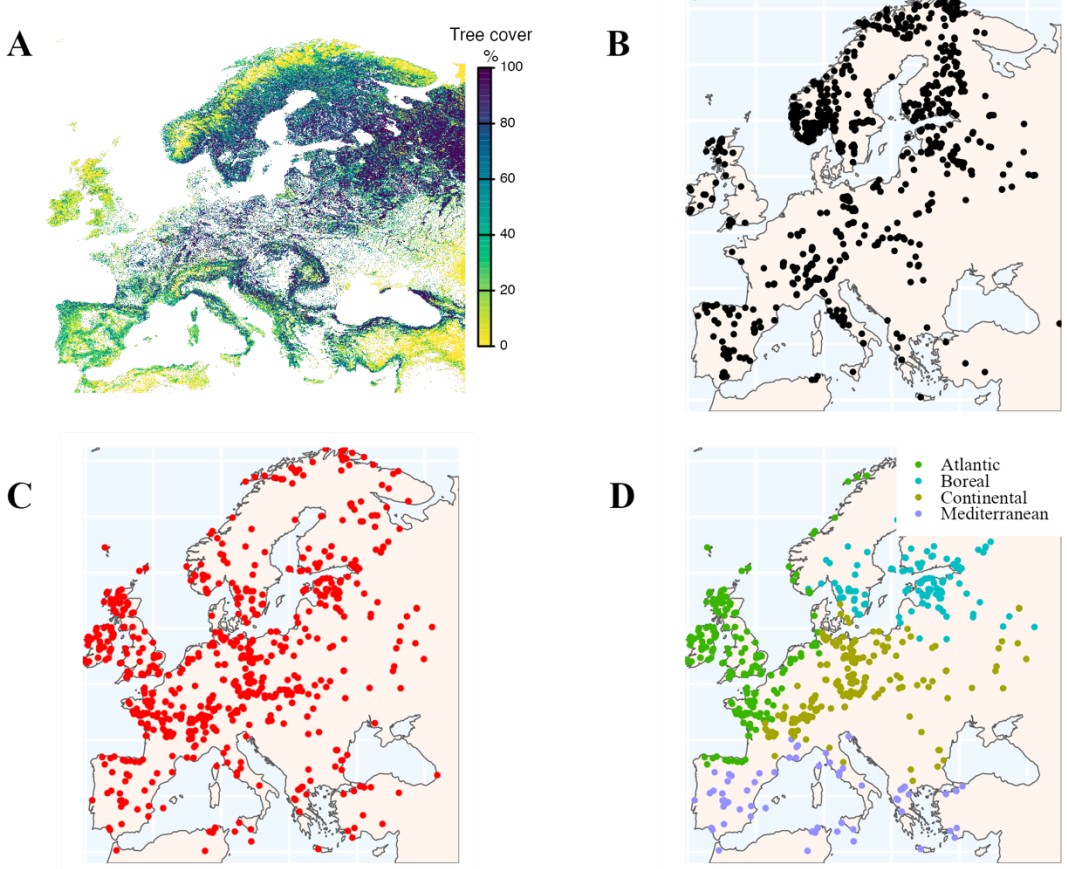

**Figure 2: A - Observed tree cover based on compositing annual tree cover maps from the Copernicus land cover data sets (2015-2019) and screening out cells where the dominant land cover was not natural; B - Modern pollen records used for model fitting; C - Fossil pollen sites used for tree cover reconstructions; D - Classification of the fossil pollen sites into climatic sub-regions**

Modern pollen data (Fig. 2B) was obtained from version 2 of the SPECIAL Modern Pollen Dataset (Villegas-Diaz and Harrison, 2022). This dataset was created from multiple different published regional datasets, from data repositories (Neotoma, PANGAEA) or directly from data collectors/authors (see Supplementary Information, Section S1,Table S1 for sources and citations) but employs a standardised taxonomy, and includes improvements to metadata and age models. The data set was further amended

for the current analysis by including updated meta information (see *Code and data availability*). The SMPDS contains pollen samples from the post-industrial era (post-1850 CE). We extracted samples that were explicitly dated to post-1950 and assumed that all samples characterised as modern but without an explicit age assignment also dated to post-1950 CE. We assume that at the regional scale these samples broadly reflect modern (2015–2019) tree cover values. Site metadata (elevation, site type, basin

size) from the SMPDSv2, along with the additional metadata updates, were used as explanatory

variables in the tree cover model (see below). Depauperate samples with Hill's N2 values (Hill, 1973) of < 2 were excluded, following Wei et al. (2021). Wei et al. (2021) found that low taxa diversity produced unreliable estimates of reconstructed variables, in this case temperature via tolerance weighted partial least squares estimation. Pollen counts from cores with multiple modern samples were averaged to prevent over-sampling. Likewise, where there were multiple cores from the same site, pollen counts were averaged so that there was only a single count from each site. Only samples from lakes and bogs were included, to ensure appropriate pollen source areas could be calculated; samples gathered via moss polsters or pollen traps were excluded as these generally reflect only the very local pollen rain. However, bog records with a radius ≥ 400m were excluded from the analysis because we included taxa that grow on bog surfaces in our analysis (see below), and the exclusion of large bogs reduces the potential for these to bias the regional vegetation reconstructions. Finally, since upslope pollen transport is known to increase the proportion of non-local pollen at high-elevation sites (Fall, 1992; Ortu et al., 2008, 2010), and the complex topography of mountainous areas also impacts pollen transport (Markgraf, 1980; Bunting et al., 2008; Marquer et al., 2020; Wörl et al., 2022), we excluded 236 site records above 1000m. To test the effect of the inclusion of bog sites and the restriction on site elevation, we ran three alternative models excluding bogs, including high elevation sites and applying a more restrictive 500m limit to site elevation, and examined summary statistics for each.

The percentage of arboreal pollen (AP%) and shrub pollen (SP%) were calculated based on the Total Terrestrial Pollen Sum (TTPS), and allocation of species into AP/SP or herbaceous (e.g. grass/ herb) pollen types in Europe (see Supplementary Information, Section S2, Table S2 and *Code and data availability*). To take into account broadscale differences in pollen productivity and pollen transport for species at a site level, the pollen data were also used to calculate the needleleaf share of the AP (%needleleaf), the share of tree pollen that are from species that are primarily wind pollinated (%wind pollinated) and the proportion of *Pinus* in the tree pollen (%*Pinus*). We also calculated the Shannon Index (SI) of tree species diversity from the pollen data. Species identified as not native to Europe, obligate aquatics and cereals were excluded from the TTPS, but Cyperaceae, Polypodiales and Ericaceae were included as characteristic of more open environments and to prevent the pollen assemblages from these environments being dominated by pollen transported from long distances.

The source area for each record, and hence the appropriate area for the calculation of mean tree cover, was calculated using Prentice's (1985) source area formula for 70% of pollen, and lake or bog area from the SMPDS, noting that the original formula makes no distinction for different site types. The selection of 70% is a simplification and other research has suggested lower percentages may be sufficient to capture most of the vegetation signal (e.g. ~30 – 45%; Sugita, 1994), so we also tested the implications of using 50 and 35% pollen source areas. The original source area formula used species-specific FS values but these are not available for all the taxa used in our analysis, so here we used the median FS

(0.03) from Githumbi et al. (2022) and Serge et al. (2023) since the tree cover map represents the broad species community around each record location. We assumed basin areas were circular to calculate the radius. For those sites that have no exact information on basin size but were categorised as small (0.1–1km$^2$), we assumed basin areas of 0.5km$^2$: there were 30 such sites included in the model construction. Source area radii varied in size from 5,026m to 418,894m for the largest lake, with a median of 28,316m. These values are reduced when using areas representing 50% and 35% of pollen: for 50% of source pollen the radii varied from 591m to 141,392m, with a median of 5293m; for 35% of source pollen the source radii varied from 150m to 72,919m, with a median of 1908m. Mean tree cover was calculated for each site from the tree cover map using the R package *exactextractr* (function*: exact_extract)* (Baston, 2023; version 0.10.0). There were 263 records where more than half of the contributing grid cells were masked as land-cover classes other than vegetation; these were excluded from the model construction. A total of 852 pollen records were included in the final model training dataset, of which 133 were bog records.

Fossil pollen data were obtained from the SPECIAL-EPD dataset (Harrison et al., 2024), a database of pollen records from Europe, the Middle East and western Eurasia. It builds on a pollen compilation covering the Middle East (EMBSecBIO database: Cordova et al., 2009; Harrison and Marinova, 2017; Marinova et al., 2017; Harrison et al., 2021), data available from public data repositories (NEOTOMA: https://www.neotomadb.org/; PANGAEA: https://www.pangaea.de/) and data provided by the original authors for the Iberian Peninsula (Liu et al., 2023) and some other regions. It includes 1,758 records from 1,573 sites. The data have been extensively quality-controlled and mistakes have been corrected and documented. New BACON Bayesian age models, based on the recalibration of radiocarbon ages using INTCAL2020 (Reimer et al., 2020) or Marine20 (Heaton et al., 2020) calibration curve as appropriate, are provided for all the records using the *rbacon* R package (Blaauw and Christen, 2011; version 3.4.2) in the *ageR* R package (Villegas-Diaz et al., 2021; version 0.2.0.900).We filtered the fossil data in the same way as the modern pollen data, by only using pollen records from lakes and bogs below 1000m in the region 12°W to 45°E and 34-73°N. Where there were multiple records from the same site, we selected a single record, prioritising the record with the maximum number of samples for the period 12,000 cal. BP to present (Fig. 2C). The pollen counts were used to calculate AP% and SP%, %needleleaf, %wind pollinated, %*Pinus* and the tree SI, using the species categorisation applied to the modern pollen data (see Supplementary Information, Section S2, Table S2 and *Code and data availability*).

## 2.2  Model development

A Beta regression, which is suitable for use with proportional data, was used to model tree cover using the R package *betareg* (Cribari-Neto and Zeileis, 2010; version 3.2-2). The explanatory variables included AP%, SP%, %needleleaf, site elevation, site type (lake or bog) and arboreal pollen SI values. The AP% and SP% were expected to explain most of the variability in tree cover. The %needleleaf was also included to reflect potential broad differences in both pollen productivity and transport between needleleaf and broadleaf species (see Table 1 from Serge et al., 2023), without having to limit the species considered to those for which there are RPP values. As an additional approach to address differences in pollen production and transport, we tested the implications of including broad pollen dispersal syndromes in the model in place of %needleleaf. We calculated the percentage of the tree pollen based on wind pollination as opposed to biotic (insect, bird) or dual (wind or biotic) pollination methods, following the categorisations by Tong et al. (2023), and Kling and Ackerly (2021) (see Supplementary Information, Section S2. Table S2). *Pinus* is the most ubiquitous pollen type recorded in the modelled data set, and is widely recognised as a potential contaminant in more open vegetation due to long-distance transport. We therefore also tested the impact of including %*Pinus* as a predictor in the regression model. Although records above 1000m were excluded from the data set, site elevation was included as an explanatory variable to capture any residual impacts of elevation on tree cover. We tested the implications of including sites of all elevations within the model, as well as a more restrictive 500m upper limit for sites. Site type (bog, lake) was included because there is a greater potential for pollen mixing prior to sedimentation in lake settings, which means that lakes may be more representative of the regional tree cover (Sugita, 1993; Githumbi et al., 2022). We also included tree SI, as a way to evaluate potential impacts of very localised tree cover or long-distance transport from a single taxa or few taxon influencing the recorded AP%. The final model was selected based on the Akaike Information Criteria (AIC: Akaike, 1974) and Cox-Snell $R^2$ value (Cox and Snell, 1989). We tested the inclusion of interaction effects associated with elevation, since the relationship between AP% and tree cover may be directly influenced by elevation due to upslope transport. In addition, given the potential importance of both RPP and transport, and landscape fragmentation, we tested the inclusion of second and third order polynomial coefficients for %needleleaf and tree SI. As Beta regression allows explicit modelling of the precision parameter as well as the mean (i.e. variance does not need to be consistent across observations) (see Simas et al., 2010), we tested the inclusion of regressors describing precision. Finally, as excluding cells dominated by crop cover in the modern observed tree cover map may have implications for downcore reconstructions, we also tested the implications of removing this crop cover restriction when calculating observed tree cover on the regression model fit.

The modern tree cover model was tested based on leave one out cross validation (LOOCV), and calculation of root mean squared error (RMSE), mean absolute error (MAE) and $R^2$ correlation between

observed and predicted values. A quantile mapping adjustment, using the R package *qmap* (Gudmundsson et al., 2012; version 1.0-6) was calculated on the LOOCV model predictions to account for compression of the reconstructions, with overestimation of low tree cover and underestimation of high tree cover values, following Zanon et al. (2018). This decompression was then used to adjust downcore reconstructions generated from the Beta regression.

## 2.3 Application of model to fossil pollen data

The tree cover model, adjusted to deal with the compression bias, was applied to the variables generated from 811 records from the SPECIAL-EPD with data for part or all of the interval between 12,000 cal. BP and the present day. Since the records cover different time periods and have different temporal resolutions, reconstructed tree cover values were binned in 200-year bins. Standard error estimates for site predictions were calculated through the application of a bootstrapping approach, with 1000 resamples of the model training data used to generate models, and equivalent quantile mapping adjustment, which were then applied to the fossil pollen data. We examined temporal trends in tree cover for the European region as a whole and for modern biogeographical regions as defined by the European Environment Agency (EEA) classification (European Environment Agency, 2016). We also produced maps of tree cover through time for 50km$^2$ grid cells by averaging reconstructed tree cover across all sites in the same cell. These reconstructions were compared with the Serge et al. (2023) and Zanon et al. (2018) reconstructions of Holocene tree cover. As basin size was not available for all record site locations, we extracted tree cover median values using a general 5km$^2$ buffer to maximise the number of site comparisons and to prevent edge effects.

## 2.4 Comparison of reconstructions

Our new approach shares some features with the methods used in previous reconstructions (Table 1), but is less data-demanding than the REVEALS-based technique and does not require *a priori* decisions about the selection of analogues. We compare our final predictions to both modern and fossil reconstructions of tree cover by Serge et al. (2023) and Zanon et al. (2018). Modern is defined as the interval 100 cal. BP and the present day in Serge et al. (2023) and between 125 cal. BP and present for Zanon et al. (2018). We make comparisons to the Serge et al. (2023) reconstructions based on the 31 taxa originally used by Githumbi et al. (2022) since Serge et al. (2023) show that this produces better results than using the expanded data set of 46 taxa.

**Table 1: Key elements of the reconstruction technique from this study, the REVEALS approach (i.e. Serge et al., 2023) and MAT (i.e. Zanon et al., 2018)**

| | This paper | REVEALS (Serge) | MAT (Zanon) |
|---|---|---|---|
| **Training data** | Modern pollen data<br>Modern tree cover | NA | Modern pollen data<br>Modern tree cover |
| **Training model** | Regression based model linking modern pollen to modern tree cover | NA, although defined relationship between RPP and FS per taxa from modern data underpins technique | Calibration of modern pollen assemblages to tree cover |
| **Downcore data requirements** | Pollen data;<br>Site characteristics | Pollen data;<br>Site characteristics;<br>RPP and FS per taxa | Pollen data |
| **Main Assumptions and Challenges** | Regression model applicability and included variables | Accuracy of RPP and FS values;<br>Limited set of taxa | Number of analogues used (commonly 3-5);<br>Threshold of similarity;<br>Non-analogues |
| **Scale** | Site-based | Typically 1º where sites are located | Site-based;<br>(Zanon: spatio-temporal interpolation) |

For each of the 1º grid cells in Serge et al. (2023), tree cover was calculated from the sum of the appropriate vegetation types. Time series of the change in median tree cover were constructed using median tree cover corresponding to the pollen source area of each of our individual modern reconstructions. As the Serge et al. (2023) and Zanon et al. (2018) data is available in gridded format, comparison with our site-based predictions is not straightforward. Where the site location source areas straddled multiple grid cells, a median was calculated, weighted by the proportion of grid cell coverage using R package *exactextractr (*function*: exact_extract)* (Baston, 2023; version 0.10.0). The tree cover time series for the Zanon et al. (2018) and Serge et al. (2023) data were initially constructed using all of the extracted tree cover values for each of our model training site locations. However, since there can be multiple sites within some of these grid cells, we tested whether this affected the comparisons by taking an average of extracted tree cover values for locations sharing the same grid cell values from Zanon et al. (2018) or from Serge et al. (2023), and using this to create new time series for these two reconstructions.

## 3. Results

### 3.1 Model fit and validation

The final model has a (Cox-Snell) pseudo-$R^2$ of 0.60, and LOOCV RMSE of 0.14 and MAE of 0.11, indicating a reasonable fit to the data. Variance inflation factor (VIF) scores are not readily interpretable

because of the inclusion of interaction terms, but a version of the model with the same variables but excluding the interaction terms has VIF values < 2 for all the explanatory variables, indicating that there is no multicollinearity and that all the explanatory variables represent independent controls on tree cover. The Cox-Snell $R^2$ for this reduced model was 0.54.

There is a positive relationship between tree cover and AP% and a negative relationship between tree cover and SP% (Table 2), as expected. However, the strength of each relationship is moderated by elevation, with increasing elevation reducing both the positive effect of AP% and the negative effect of SP% on tree cover. There is a negative correlation between %needleleaf and tree cover, although the significant positive quadratic term for needleleaf suggests this relationship becomes positive at higher abundances of %needleleaf. This negative relationship may be a reflection of longer distance pollen transport of needleleaf species (e.g. *Pinus*) to open environments. As tree cover increases, this may imply an increased diversity of species, including broadleaf species. The positive quadratic term indicates that this relationship becomes positive at higher levels of tree cover, potentially reflecting higher tree cover in boreal needleleaf forests. Replacing %needleleaf with either %wind pollination tree species or %*Pinus* as explanatory variables related to pollen transport resulted in a poorer model fit (Supplementary Information, Section S3, Tables S3 and S4). Increased tree SI is positively related to tree cover, with the effect decreasing with elevation. However, the negative correlation for the quadratic term for the SI suggests that the relationship has less of an effect on tree cover as tree SI increases. Again, this relationship may be explained in the context of open environments, where tree species diversity may be limited to species with longer distance pollen transport. For example, records in tundra tend to have a greater average disconnect between observed tree cover and AP%, as well as the lowest average tree SI values by biome (Supplementary Information, Section S4, Table S5, Fig. S1 and Fig. S2). Tree species diversity may then increase with tree cover, with the negative quadratic term implying that the highest levels of tree cover are represented by relatively uniform species types. There is no significant relationship between site type and tree cover, but the interaction term between them is significant, with a reduction in the effect of elevation on the likelihood of higher tree cover for lake sites. One possible explanation for this relationship is that the increasing likelihood of longer distance pollen transport with elevation affects bog sites more than lake sites in a relative sense, in that bog sites typically have a smaller source area. The Cox-Snell $R^2$ value increases slightly to 0.62 (from 0.60) if bogs are excluded from the model (Supplementary Information, Section S5, Table S6) but this causes a substantial reduction in spatial and temporal coverage. Conversely, including higher elevation sites (>1000m) within the model reduces the Cox-Snell $R^2$ to 0.50 (from 0.60) (Supplementary Information, Section S6, Table S7). Limiting the maximum elevation of sites had very limited impact on overall model fit (Supplementary Information, Section S7, Table S8). A likelihood-ratio test of the model with the inclusion of variables for precision against a model with a constant precision parameter shows that there is a significant improvement in the model with variable dispersion. Increases in %needleleaf and

SI are associated with increased precision of the tree cover reconstructions, whereas lake sites are generally less variable in terms of tree cover than bog sites.

**Table 2: Modern tree cover model coefficients**

| Coefficients (mean model with logit link) | Estimate | Standard Error | P Value |
|---|---|---|---|
| (Intercept) | -5.598 | 0.437 | 1.54e-37 *** |
| Tree pollen % | 2.374 | 0.223 | 1.56e-26 *** |
| Shrub pollen % | -3.458 | 0.630 | 4.06e-08 *** |
| Needle share of AP% | -1.317 | 0.456 | 0.004 ** |
| Needle share of AP%^2 | 3.009 | 0.514 | 4.80e-09 *** |
| AP Shannon index | 5.091 | 0.458 | 1.04e-28 *** |
| AP Shannon index^2 | -1.375 | 0.138 | 2.09e-23 *** |
| Lake or bog site | 0.031 | 0.132 | 0.815 |
| Elevation | 0.003 | 0.001 | 0.005 ** |
| AP pollen:elevation interaction | -0.001 | 0.000 | 0.003 ** |
| SP pollen:elevation interaction | 0.004 | 0.001 | 0.001 ** |
| AP Shannon:elevation interaction | -0.004 | 0.001 | 2.06e-04 *** |
| AP Shannon^2:elevation interaction | 0.002 | 0.000 | 2.10e-06 *** |
| Lake or bog site:elevation interaction | -0.001 | 0.000 | 5.57e-04 *** |
| | | | |
| **Precision submodel (log link; after variable selection^^)** | | | |
| (Intercept) | 0.407 | 0.256 | 0.112 |
| Needle share of AP% | 0.798 | 0.229 | 5.05e-4 *** |
| AP Shannon index | 0.840 | 0.121 | 4.04e-12 *** |
| Lake or bog site | 0.534 | 0.126 | 2.17e-5 *** |

Significance codes: 0 = '***'; 0.001 = '**'; 0.01 = '*'; 0.05 = '' 0.1; ' ' = 1

^^Only significant (at 5% significance) covariates were included

The influence of each variable on the quality of the statistical model is shown in Table 3, with the change in AIC value based on the removal of each variable, including the removal of associated interaction, polynomial and precision terms as applicable. Although AP and SP were expected to be the most important explanatory variables, the model only using only these two variables has an AIC value 568 greater than the final model, and a Cox-Snell $R^2$ of only 0.27 (Table 2). The poorer fit may reflect the observed limitation of using linear pollen percentages to represent tree cover ("the Fagerlind Effect"; Fagerlind, 1952; Prentice and Webb, 1986). Although pollen percentages provide a reasonable approximation of tree cover (Prentice and Webb, 1986), the inclusion of the other variables is important in fitting the final model.

**Table 3: Change in modern model AIC values and Cox-Snell R² model values when excluding specific variables (exclusion includes interactions, polynomials and precision variables) and for a model with only arboreal and shrub pollen percentage**

| Model | $\Delta$ AIC | Cox-Snell $R^2$ |
|---|---|---|
| Final model | 0 | 0.60 |
| excluding AP | 165 | 0.51 |
| excluding SP | 31 | 0.58 |
| excluding %needleshare | 121 | 0.55 |
| excluding AP Shannon index | 396 | 0.41 |
| excluding lake or bog site | 56 | 0.57 |
| excluding elevation | 204 | 0.48 |
| AP and SP model | 568 | 0.27 |

We tested the influence of excluding cells dominated by crop cover when calculating observed tree cover values for each record location by re-running the regression using the same variables but without excluding these cells. The number of records used to train the model increased to 1050 (from 852), but the model pseudo (Cox-Snell) $R^2$ was reduced from 0.60 to 0.47 (Supplementary Information, Section

S8, Table S9), supporting out decision not to include these cells in the model training. Using different pollen percentage values to calculate the source area for each record resulted in an increase in the number of records considered (935 for 35%, 934 for 50% compared to 852 in our original model) but the model fits were worse, with pseudo (Cox-Snell) $R^2$ reduced to 0.58 and 0.55 for 50% and 35% of pollen respectively (Supplementary Information, Section S9, Tables S10 and S11).

The application of the quantile mapping approach reduced the bias towards the mean, whilst preserving the general structure of the data (Supplementary Information, Section S10, Fig. S3). However, there is still a tendency for under- and over-estimation at low and high observed tree cover respectively in the final model (Fig. 3A). There is no obvious spatial patterning in the biases, except for a tendency to

375 overestimate tree cover in northernmost Scandinavia (Supplementary Information, Section S11, Fig. S4). The correlation between the final "decompressed" predictions and observed tree cover values is 0.80 (Fig. 3B). This correlation value compares favourably to the correlation between raw AP% and observed tree cover values (0.54): raw AP% values tend to overestimate observed tree cover (Supplementary information, Section S12, Fig. S5).

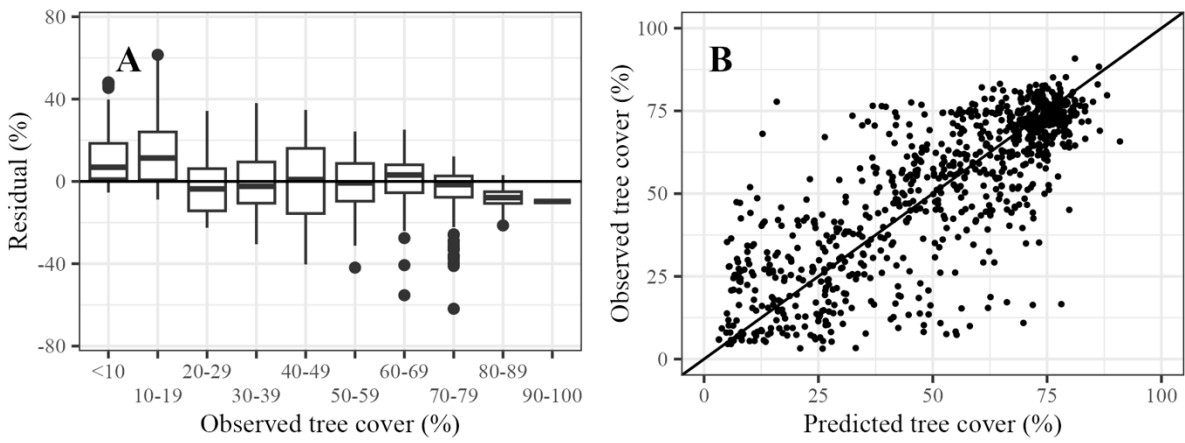

**Figure 3: Evaluation of final model performance. A – Differences between predictions and observations (residual), in bins of observed tree cover percentage; B – Predictions of tree cover compared to observed tree cover**

Our modern predictions differ from those of Serge et al. (2023) and Zanon et al. (2018) (Fig. 4). The correlation between the Zanon et al. (2018) reconstructions and observed tree cover is 0.78, which is similar to the correlation between our predictions and observed tree cover (0.8). The Zanon et al. (2018) reconstructions have fewer outliers, but nevertheless they underestimate tree cover at high levels of observed tree cover (Fig. 4A). The correlation between the Serge et al. (2023) predictions and observations is only 0.5. This is partly caused by the use of larger 1º grid cells, but even when taking this into account and comparing with an average value for each grid cell, the correlations between predictions and observations were still lower (0.59) than our predictions and those of Zanon et al. (2018) (Supplementary Information, Section S13, Fig. S6). Zanon et al. (2018) visually compare their interpolated modern tree cover map and the observed tree cover from Hansen et al. (2013), but they do not report a correlation or $R^2$ value. However, the correlation values for Serge et al. (2023) broadly align with those reported comparing REVEALS tree cover estimates, with observed tree cover derived from Hansen et al. (2013) ($R^2 = 0.15$; correlation ~ 0.4).

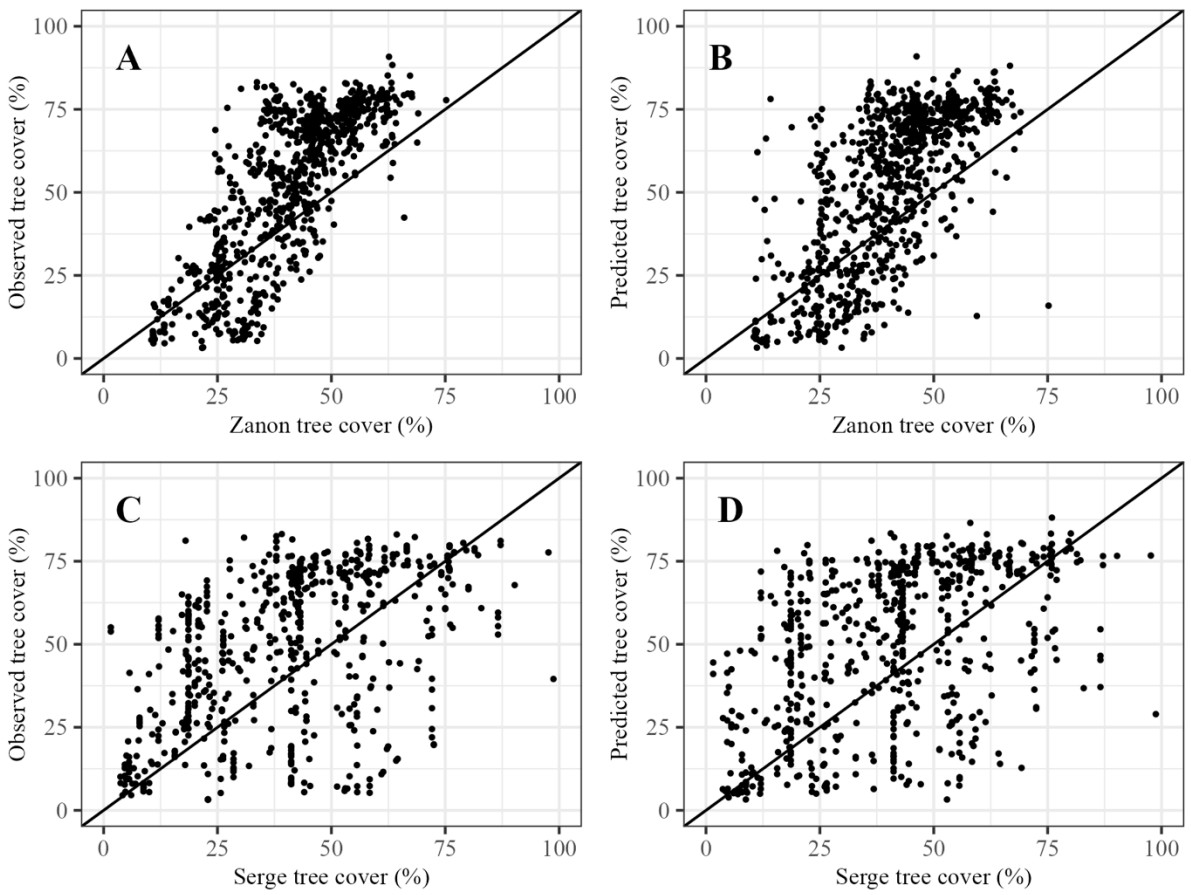

**Figure 4: Modern tree cover from Zanon et al. (2018) compared to (A) observed tree cover and (B) our predicted tree cover. Modern tree cover from Serge et al. (2023) compared to (C) observed tree cover values and (D) our predicted tree cover.**

## 3.2 Holocene changes in tree cover

We applied the tree cover model to reconstruct Holocene changes in tree cover at individual sites (see section *Data and code availability* for binned reconstructions at site level). The median tree cover value, considering Europe as a whole, increased rapidly between 12,000 to ca. 8,500 cal. BP, and then more

slowly to a peak at ca. 6,500 cal. BP. Median tree cover declined overall between 6,500 cal. BP and 4,000 cal. BP, albeit with some variability. Median tree cover declined steadily to present-day levels after ca. 4,000 cal. BP (Fig. 5A). This same pattern is shown when considering changes in mean tree cover (Supplementary Information, Section S14, Fig. S7), different LOESS smoothing (R package *locfit:* Loader, 2020; version 1.5-9.12) of the median tree cover value (Fig. 5B), and based on median

tree covers reconstructed with the model bootstraps used to generate reconstruction standard errors (Supplementary Information, Section S15, Fig. S8). The rapid warming at the end of the Younger Dryas (Alley, 2000; Cheng et al., 2020), and the changes associated with the 8,200 cal. BP event (Alley et al., 1997; Alley and Ágústsdóttir, 2005) and the 4,200 cal. BP event (e.g. Weiss et al., 1993; Bini et al., 2019) are not apparent at this pan-European scale.

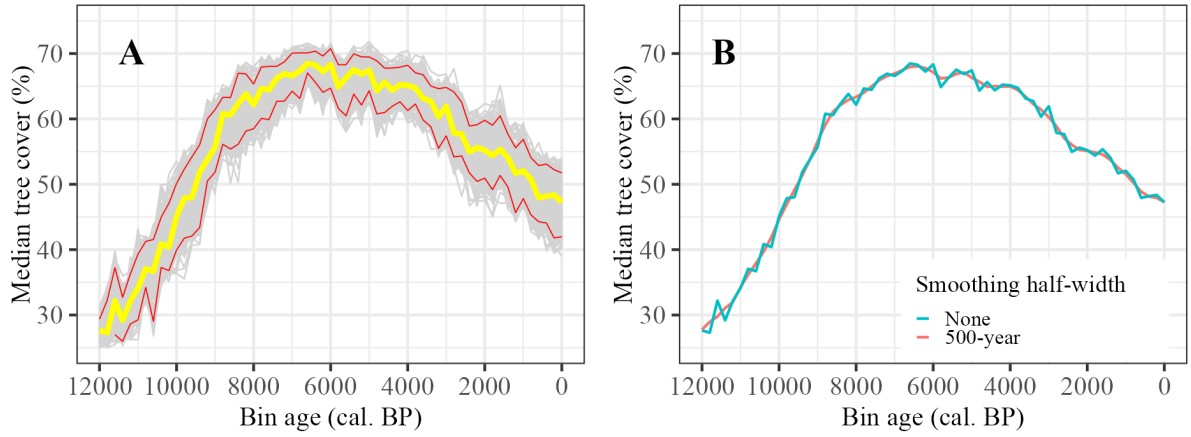

**Figure 5: A - Median reconstructed tree cover for Europe from 12,000 to 0 cal. BP, with 95% confidence intervals for 1000 bootstrap resampling of records; B - Median reconstructed tree cover for Europe from 12,000 to 0 cal. BP, with differing LOESS regression smoothing half-widths.**

Given the importance of the tree SI to the regression model (Table 2), we also ran the reconstructions

based on a model excluding this variable. Although the reconstruction followed the same broad mid-Holocene increase and decline in tree cover, median tree cover prior to 7,000 cal. BP was much greater than shown in Fig. 5, with a less marked increase to the mid-Holocene (see Supplementary Information, Section S16, Fig. S9 and Fig. S10). As shown in Fig. S10, there is a slight increase in tree SI through time, which may imply a slight underestimation in tree cover during the early part of the Holocene.

These pan-European trends mask considerable regional variability in tree cover at any given time and in trends through time (Fig. 6 and see Supplementary Information, Section S17, Figs. 11 to 19 for gridded maps of reconstructed tree cover).

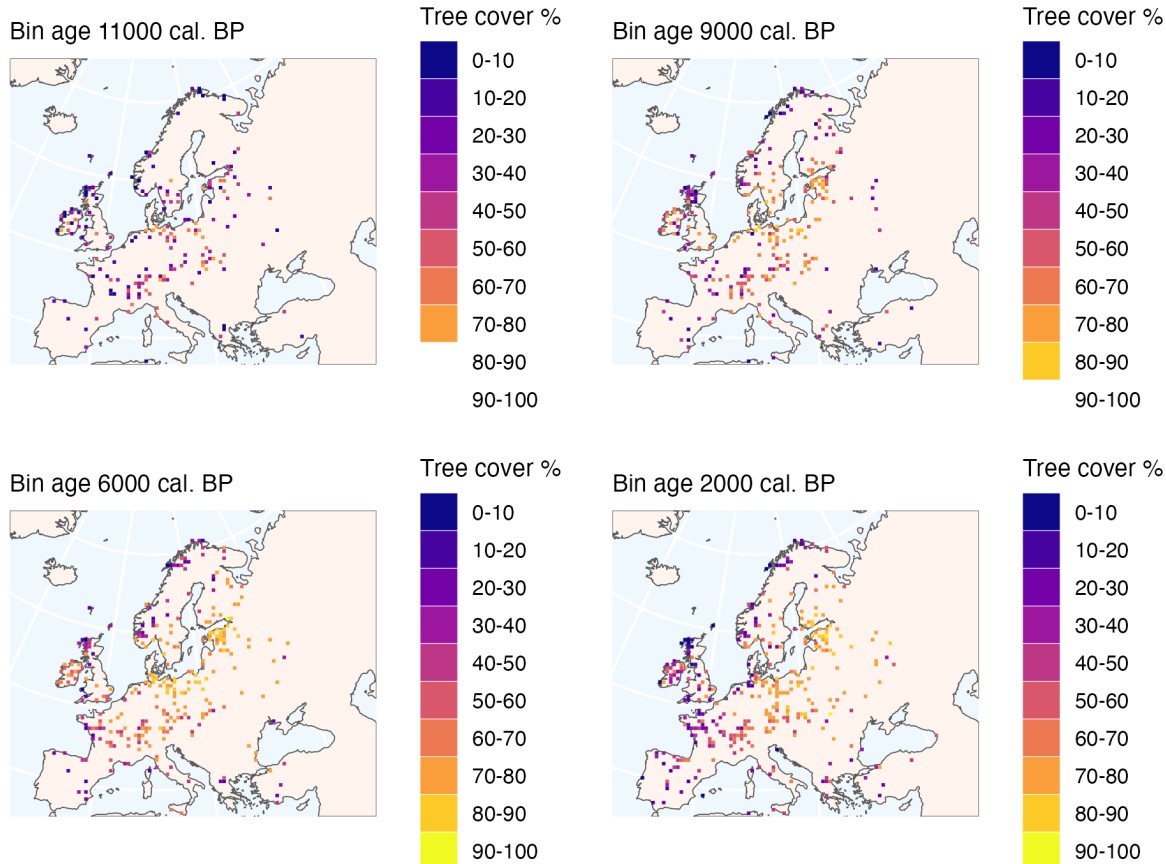

**Figure 6: Gridded maps of average reconstructed tree cover for selected periods, for 50km² grid cells. Bin ages are 200-years in width, with ages referring to mid-point of each bin.**

To examine these trends, we consider four biogeographical regions for which there is sufficient data: the Atlantic, Boreal, Continental and Mediterranean regions (Figure 2D; Supplementary Information, Section S18, Table S12).

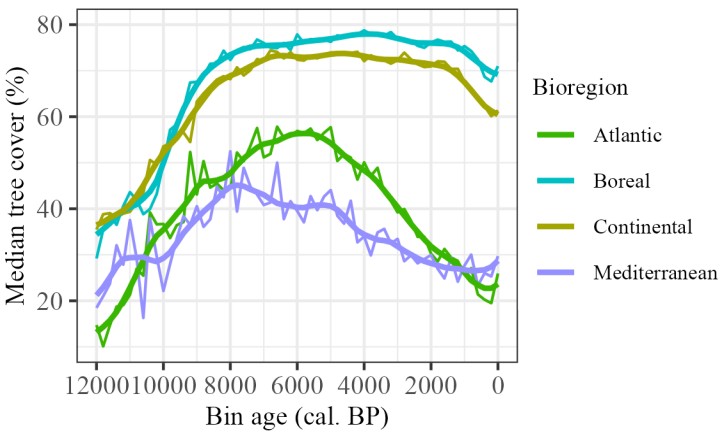

**Figure 7: Median tree cover values, for selected biogeographical regions. Smoothed lines reflect LOESS fitted regression with 1000-year halfwidth.**

The increase in tree cover at the beginning of the Holocene is shown in all four regions, but the trajectories are different (Fig. 7). There is an immediate rapid increase in the Atlantic region, but the

 increase is initially slow in the Boreal and Continental regions and only becomes more rapid after ca.

11,000 cal. BP in the Continental region and after ca. 10,300 cal. BP in the Boreal region. The initial increase in the Mediterranean region is interrupted by a decline between 11,000 and 10,000 cal. BP and only begins to increase again after ca. 9,800 cal. BP. The maximum in tree cover in the Mediterranean region is reached at ca. 8,000 cal. BP. After this there is a decline towards present-day levels, although this is interrupted by intervals of relative stability e.g. between ca. 6,000 and 5,000 cal. BP, again between 4,000 and 3,000 cal. BP and in the most recent millennium. There is also a well-defined maximum in tree cover in the Atlantic region, but this occurs at ca. 6,000 cal. BP. The subsequent decline is gradual until the last ca. 500 years. The maximum in tree cover is broader in the Continental and Boreal regions, with high levels of tree cover characteristic of the entire interval between ca. 8,000 and 1,000 cal. BP although the absolute maximum occurs at ca. 6,400 cal. BP in the Continental region and not until ca. 4,000 cal. BP in the Boreal region. Both regions are characterised by a rapid decline in the last millennium.

This broad pattern of increase, mid-Holocene maximum and then decline to present is consistent with previous reconstructions (Fig. 8). Initial levels of tree cover are similar in the three reconstructions (ca. 27.5-30%), but the increase in tree cover is more rapid in the Zanon et al. (2018) and Serge et al. (2023) reconstructions. Our reconstructed maximum cover is slightly lower (ca. 5–10%) than shown by the other reconstructions. However, the mid-Holocene timing of this maximum is broadly consistent across all of the reconstructions (although Zanon et al. (2018) show a double peak in tree cover, with an earlier peak at ca. 9,000 cal. BP) within the limitations of the age models and binning intervals used (see Supplementary Information, Section S19, Fig. S20, and Section S20, Fig. S21). These broad trends are maintained when calculating tree cover from Zanon et al. (2018) and Serge et al. (2023) data such that only a single value per grid cell is permitted in the calculation of the median (Supplementary Information, Section S21, Fig. S22). The biggest difference between our reconstructed tree cover and previous studies is that the decrease post-ca. 2000 cal. BP is less abrupt. Both Zanon et al. (2018) and Serge et al. (2023) show a steep decline to levels (ca. 35%) similar to those at the beginning of the Holocene whereas we show a decline to only ca. 47.5%. Based on pollen record locations and a $5km^2$ buffer, observed modern median tree cover is 46%, which suggests our estimated decline is more realistic.

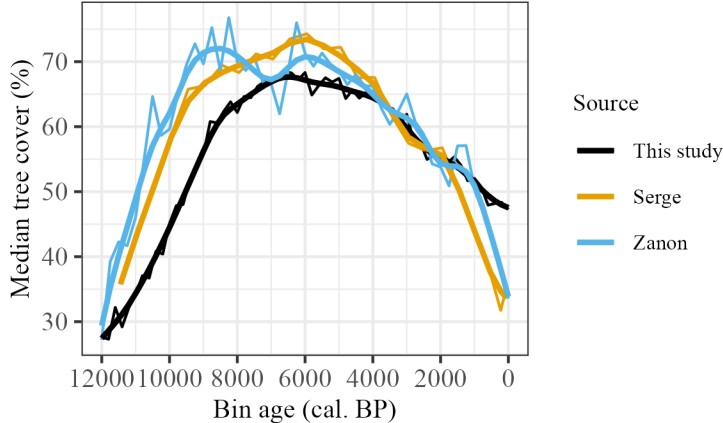

**Figure 8: Reconstructed median tree cover compared to equivalent extracted tree cover medians for Serge et al. (2023) and Zanon et al. (2018). Smoothed lines reflect LOESS fitted regression with 1000-year halfwidth.**

The three reconstructions also show some similarities at a sub-regional scale (Fig. 9), although comparison is more difficult because of differences in methodologies and coverage. As is the case for

the pan-European comparison, Zanon et al. (2018) and Serge et al. (2023) generally show higher tree cover than our reconstruction. The peak tree cover in the Atlantic region occurs earlier in these reconstructions, and Zanon et al. (2018) also show an earlier peak in the Continental region. The later Holocene decline in tree cover is similar across all three reconstructions in the Atlantic region, but the previous reconstructions show a steeper decline in the Boreal and Continental regions. The biggest

differences between the three reconstructions is in the Mediterranean region, where Serge et al. (2023) show a pronounced peak reaching 60% cover at ca. 5,500 cal. BP, whereas the other two reconstructions show comparatively muted changes in tree cover at around 40% throughout the mid- to late Holocene.

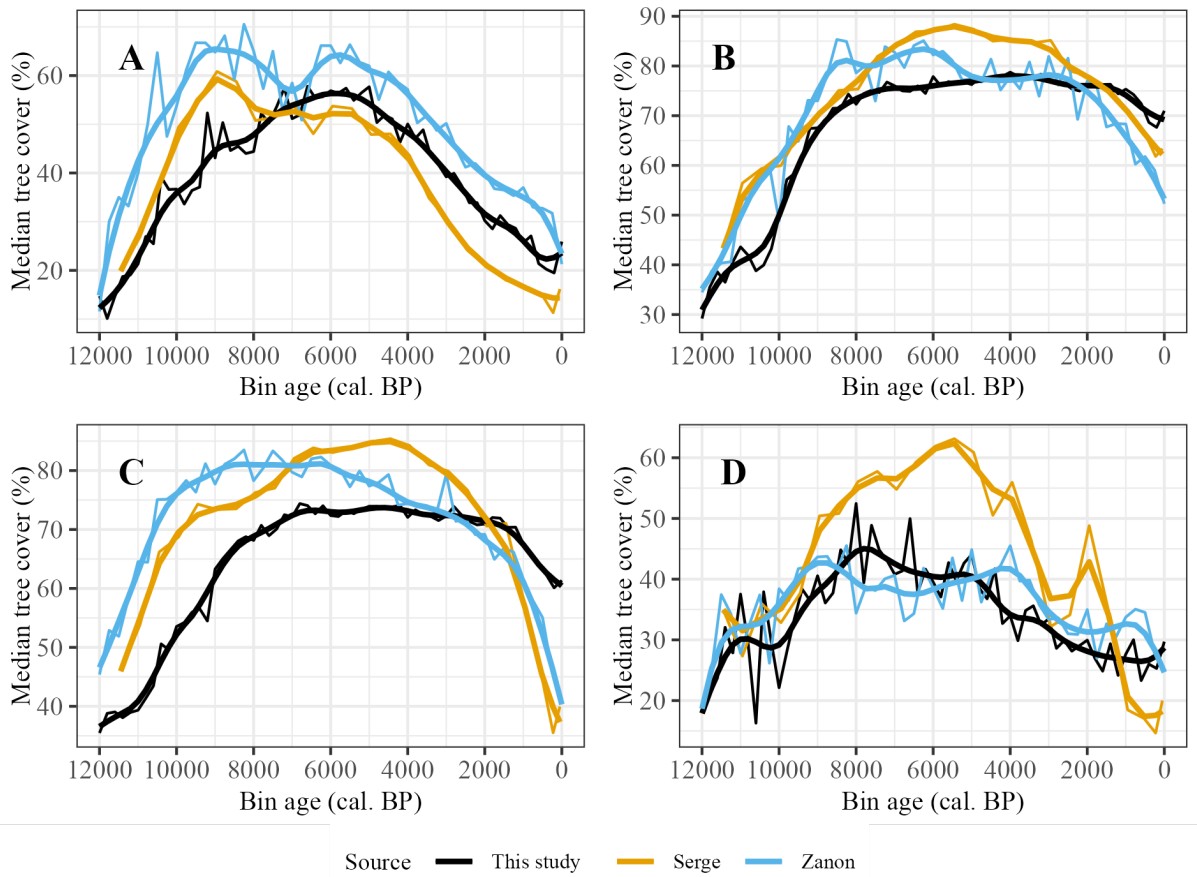

**Figure 9: Reconstructed median tree cover compared to equivalent extracted tree cover medians for Serge et al. (2023) and Zanon et al. (2018), for selected modern biogeographical regions. Smoothed lines reflect LOESS fitted regression with 1000-year halfwidth. A – Atlantic; B- Boreal; C- Continental; D – Mediterranean.**

Although the overall European median tree cover partially reflects the quantity of data for each region, recalculating the European level median tree cover based on just the four biogeographical regions of focus has very little material effect on the overall pattern of median tree cover for each reconstruction (Supplementary Information, Section S22, Fig. S23).

## 4. Discussion

Our reconstructions show that tree cover peaked in the mid-Holocene period, with median tree cover ca. 40% greater than at the beginning of the Holocene. This general pattern is shown by the REVEALS and MAT reconstructions, and is also visible in plant functional type (Davis et al., 2015) and pseudo-biomization reconstructions of vegetation cover (Fyfe et al., 2015). Despite the similarities in median values between our reconstructions and those calculated from the REVEALS and MAT reconstructions, our site-based estimates are not fully comparable with the gridded estimates provided by the REVEALS reconstruction and the gridded and interpolated values provided by the MAT reconstructions. Nevertheless, the similarities give some support to the overall robustness of our reconstructions. There

are differences in the timing and extent of changes between regions, although the general pattern of increase, mid-Holocene peak and subsequent decline is shown everywhere. There are some differences between our reconstructions and those from other studies. Firstly, the maximum tree cover from our reconstructions is around 5–10% less than the maximum calculated from the other reconstructions. This could reflect the conservative nature of our modern-day tree cover model, which underestimates tree cover at the high end despite the application of quantitative mapping adjustment to model predictions. However, Zanon et al. (2018) also underestimate tree cover at high levels of tree cover. Alternatively, the difference may reflect the exclusion of higher elevation records from fossil dataset in order to minimise the impact of upslope pollen transport, which was not done in the other two reconstructions and would tend to reduce overall median tree cover. Secondly, the timings of peak tree cover vary between the reconstructions, with the MAT-based estimate peaking 2000 years earlier and the REVEALS estimate ca. 500 years later than shown in our reconstruction. The small difference between the peak timing shown by REVEALS and our reconstructions likely reflect differences in coverage through time and differences in the binning procedure. This may also partially explain the difference with the MAT-based reconstructions. However, this reconstruction also shows a marked decline between ca. 8000 and 7000 yr BP and a second peak in tree cover slightly after the peak shown by our reconstructions. It is difficult to assess the robustness of the reconstructed decline, which is strongly affected by a single point, but this would affect the overall shape of the curve and hence the timing of peak cover. Thirdly, the increase in tree cover in the early Holocene is less rapid in our reconstruction than the other reconstructions. This could reflect an underestimation of tree cover because of the lower tree SI values in our reconstructions, but the rapid increase in tree cover in the MAT- and REVEALS-based reconstructions is more likely to reflect an overestimation of tree cover because of long-distance pollen transport into the more open landscape characteristic of the early Holocene. The major difference at the pan-European scale is the reduction in tree cover from ca. 2000 cal. BP to present, which is less marked in our reconstructions and more consistent with observed tree cover. The observed tree cover values used in the model construction exclude areas dominated by land-cover types such as built areas or areas dominated by crops. We account for this in defining modern source areas in our model. Not accounting for changes in these other land-cover types, which through anthropogenic land use have increased substantially over the past 1000 years (Klein Goldewijk et al., 2017) would result in a steeper decline in tree cover, as seen in the other two reconstructions.

We took an inclusive approach in defining the modern training data and for the fossil records to maximise spatial coverage. The inclusion of bogs, for example, reduced the goodness-of-fit of the model slightly but resulted in much better spatial coverage. Conversely, the exclusion of high-elevation sites did not have a major impact on spatial coverage but was necessary to improve model fit because of the tendency for upslope transport of pollen in mountainous areas. Nevertheless, there are still areas of Europe which lack data, or where there is a mismatch in coverage between the modern and fossil

records. Improved sampling of such areas would enhance our confidence in the reconstructions of Holocene tree cover.

Our reconstructions are consistent with understanding of Holocene climate patterns. The rapid increase in tree cover at the beginning of the Holocene shown in the pan-European reconstruction and in most of the sub-regional reconstructions reflects the marked warming after the Younger Dryas. This warming is less pronounced and more gradual in the Mediterranean region, consistent with the fact that the

545 Younger Dryas cool interval is not strongly registered over much of this region (Bottema, 1995; Cruz-Silva et al., 2023). Subsequent pan-European changes in tree cover broadly reflect changes in growing season temperature, which itself is a reflection of orbitally-forced changes in summer insolation. However, modelling studies have shown that the timing of maximum warmth during the Holocene in Europe was delayed compared to the maximum of insolation forcing as a consequence of feedbacks

associated with the presence of the Laurentide and Scandinavian ice sheets (Renssen et al., 2009; Blaschek and Renssen, 2013; Zhang et al., 2016, 2018). However, they consistently show that the warming was delayed in the region bordering the Atlantic (and northwest Europe more generally) by ca. 2,000 years compared to more continental regions, consistent with our reconstructions. The more muted changes in tree cover in the Mediterranean region compared to other regions of Europe is

consistent with previous reconstructions and is partly explained by the fact that these changes are largely driven by changes in precipitation and precipitation seasonality (Cruz-Silva et al., 2023). The late Holocene decline in tree cover is consistent with the orbitally-driven cooling. However, the more rapid decline in tree cover during the last millennium shown in the Boreal and Continental regions, is more difficult to explain as a function of climate changes: transient model simulations of the response to

changes in orbital and greenhouse gas forcing (e.g. Liu et al., 2009; Zhang et al., 2016; Braconnot et al., 2019; Dallmeyer et al., 2020) generally indicate muted changes in either summer or winter temperatures during the most recent millennia.

There are several other factors that could have influenced tree cover during the Holocene. Human

influence on the landscape has been identified in many regions of Europe from 6,000 cal. BP onwards (e.g. Roberts et al., 2018; Zapolska et al., 2023). Although this may have contributed to the observed decline in tree cover from the mid-Holocene onwards, the most rapid population growth occurred only during the past 2000 years (Klein Goldewijk et al., 2010, 2017). The recent decline in tree cover may therefore reflect this rapid growth and the consequent increasing human influence on the landscape in

some regions (see e.g. Marquer et al., 2017; Roberts et al., 2019). Climate-driven changes in disturbance (wildfires, windthrow) may also contribute to the inferred changes in tree cover. Changes in the frequency or intensity of storms has, for example, been shown in maritime Europe (e.g. Pouzet et al., 2018; Sjöström et al., 2024) during the late Holocene; storms are a major cause of widespread forest damage in Europe today (Senf and Siedl, 2020) and could have been important during the Holocene.

Changing wildfire regimes could also have been an important influence on tree cover (Marlon et al., 2013; Kuosmanen et al., 2014). Much of the debate about the relative importance of climate and human activities on the environment during the Holocene has been based on local-scale correlations; other contributing factors have been largely ignored. More formal modelling of these relationships, using quantitative information on climate, population size, and disturbance is required to assign the impact of 580 each on tree cover more confidently.

Our simple modelling approach yields a reasonably robust picture of changes in tree cover through the Holocene, largely consistent with known changes in climate. We have shown that both %needleleaf and the SI are important predictors of tree cover. These measures are, to some extent, surrogates for pollen 585 productivity and factors affecting pollen transport distance as explicitly addressed in the REVEALS-based reconstructions. Thus, their importance in the model is not surprising and we have demonstrated that the final model is able to capture modern day patterns. However, our approach is predicated on the assumption of stationarity between tree cover and the explanatory variables through time, as indeed are the other statistical reconstruction techniques considered here. This may be problematic for variables 590 such as elevation, where changes in elevational lapse rates (Mountain Research Initiative EDW Working Group, 2015) or atmospheric circulation patterns (Bartlein et al., 2017) could affect the relationship, but is less likely to be an issue for explanatory variables that reflect biophysical controls on pollen transport and deposition such as basin type or proportion of needleleaf trees. Tree diversity is somewhat more problematic, since the influence of long-distance transport into the more open landscapes likely 595 to have been characteristic of the colder, drier climate at the start of the Holocene may not be adequately captured in the modern training data. The relatively slow rate of the initial increase in tree cover may be a reflection of this. This could be explored using macrofossil of sedimentary DNA data, to discriminate between local diversity and potential long-distance contamination of the SI index. Nevertheless, the overall pattern of changes in tree cover during the Holocene appear to robust and 600 explicable, supporting the idea that modern relationships between tree cover and the explanatory variables provide a reasonable basis for reconstructions.

Our approach is less data-demanding that the REVEALS approach. Serge et al. (2023) have pointed out that lack of reliable information on species RPP values is likely to lead to less accurate reconstructions. 605 Even in Europe, there is limited RPP data and, as the comparison of reconstructions based on 31 and 46 species shows, some of that data may not be reliable. RPP data is more limited in many other regions of the world (Harrison et al., 2020). Given this, our simple modelling approach provides an alternative method that could be applied to reconstruct tree cover globally. It is difficult to compare our reconstructions with the MAT-based approach of Zanon et al. (2018) since they do not provide 610 individual site estimates and it is therefore not possible to determine the degree to which the patterns are influenced by the spatial and temporal interpolations they applied. Nevertheless, our simple

approach overcomes the methodological issues associated with MAT, such as how to measure the degree of analogy between assemblages, how to deal with non-analogues, and the sensitivity of the reconstructions to the number of analogues used. Thus, given the existence of global modern pollen training data sets and good remote-sensing based modern estimates of tree cover, our approach could be applied to other regions of the world to generate robust reconstructions of tree cover.

## 5. Conclusions

We have made use of modern pollen data and maps of tree cover percentage to build a simple model of tree cover. We then applied this model to fossil pollen records to reconstruct tree cover at a site level during the Holocene across Europe. At a pan-European level, tree cover increased from the early Holocene to the mid-Holocene, and then subsequently declined to the present day. There are regional variations in the speed of the initial increase, the timing of maximum tree cover, and the form of the subsequent decline. Our simple approach produces similar reconstructions of the trends in tree cover during the Holocene reconstructed using other methods, and since it only requires readily available data could be used to reconstruct tree cover in other regions of the world.

## 6. Code and data availability

The SMPDSv2 modern pollen database is available from the University of Reading Research Data Archive (https://researchdata.reading.ac.uk/389/) and/or via https://github.com/special-uor/smpds?tab=readme-ov-file
Metadata updates to SMPDSv2, including basin size updates are available at https://doi.org/10.5281/zenodo.15283096
The SPECIAL.EPD fossil pollen database is available from the University of Reading Research Data Archive (https://researchdata.reading.ac.uk/1295/)
The code used in generating tree cover, binned site-based reconstructions of tree cover, the code as cross-analysis of the data, and species categorisation tables are available at https://doi.org/10.5281/zenodo.15283096

## 7. Supplemental link

The supplement related to this article is available online.

## 8. Author contribution

LS, SPH and MVL conceived this study. LS carried out the analysis. LS and SPH wrote the first draft of the manuscript and all authors contributed to the final version.

## 9. Competing interests

The authors declare that they have no conflict of interest.

## 10. Acknowledgments

We thank colleagues in the Leverhulme Centre for Wildfires, Environment and Society (https://centreforwildfires.org/) and from the SPECIAL group at the University of Reading (https://research.reading.ac.uk/palaeoclimate/) for discussions during the development of this work. Much of the fossil data were obtained from the Neotoma Paleoecology Database (http://www.neotomadb.org) and its constituent databases: European Pollen Database (EPD) and The
Alpine Palynological Database (ALPADABA). The work of data contributors, data stewards, and the Neotoma community is gratefully acknowledged. We also thank members of the EMBSECBIO data community and other palynologists who have contributed records to the modern and fossil data set used here.

## 11. Financial support

LS acknowledges support from the Leverhulme Centre for Wildfires, Environment and Society. SPH acknowledges support from the ERC-funded project GC 2.0 (Global Change 2.0: Unlocking the past for a clearer future; grant number 694481).

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
