# Peer review of "European tree cover during the Holocene reconstructed from pollen records"

_EGUsphere, 2024_

## Community Comment (CC1)

[supplement omitted: unrelated document]

---

## Community Comment (CC2)

We thank the reviewer for their feedback on this article. Our responses to individual comments are in blue, with any proposed changes to the text in red.

**General comment**

Estimates of forest cover during the Holocene inferred from pollen data may be useful for e.g. climate modelling (e.g., REVEALS-based forest cover was used in regional climate modelling for Europe in Strandberg et al. 2014 in CP, 2022 in QSR, 2023 in CP). The REVEALS-based estimates of plant cover have the advantage (above MAT and pseudobiomization) to provide cover for each of the plant taxa used in the reconstruction, which allows to then calculate the cover of various groups of taxa that may be very useful in e.g. climate modelling. In e.g., Strandberg et al. studies, evergreen trees were separated from summer-green trees. Nonetheless, it is most useful to produce as many reconstructions as possible using different methods to test a) the effect of the method itself on the result, if results differ between methods, and b) the effect on e.g., climate of the differences between the various forest/tree cover products. Therefore, producing pollen-based estimates of forest/tree cover using different methods is of value even if the comparison can be done only for forest cover versus open-land cover and cannot be done for more detailed land-cover units. The study by Sweeney et al. is thus welcome.

I have read the review of Thomas Giesecke and agree with his concerns and comments. Therefore I won't repeat the questions and comments provided by Thomas but wish to add some additional points, questions, and information/references for consideration by the authors. One of my major concerns relates to the discussion on the comparison between the reconstructions from this study's model and the REVEALS model. The authors should acknowledge that their model reconstructs FOREST cover based on a specific definition of forest, i.e. A definition among OTHERS, while the REVEALS model estimates TREE cover. In Serge et al. REVEALS reconstructions TREES include the taxa *Buxus*, Ericaceae, *Juniperus*, *Phillyrea* and *Pistacia* that may not belong to the definition of FOREST you are using in your model. This should be considered in the discussion, most importantly for the Mediterranean region. I also comment on this issue in the comments below.

Our major goal in this paper was to develop a site-based model of tree cover using readily available data, and to test this in Europe and compare it with other available reconstructions as a measure of robustness. It was not our intention to provide a detailed explanation or critique of the other methods. We agree, however, that there should be more care taken to acknowledge the difficulty in comparing the datasets, given the different scopes, methods and data presentation for each. We have addressed this issue in our responses to the detailed comments below.

**Detailed comments – a mix of major and minor ones**

**R1:** 61-68: The REVEALS reconstructions by Trondman et al. (2015) and Githumbi et al. (2022) were produced for the study of land use as a climate forcing (biogeophysical forcing) in Europe during the Holocene using climate models (papers by Strandberg et al.). The scale of the reconstructions (1 degree) and protocol used was motivated primarily by the primary aim of these reconstructions. These authors produced estimates of plant cover for 25 respectively 31 taxa. These data can be freely accessed in PANGAEA (references to be found in Githumbi et al.). As said above, Strandberg et al. used the total cover of three groups of taxa, open land taxa, evergreen tree taxa and summer-green tree taxa. This is only one possible use of these datasets. Similarly, the REVEALS reconstruction by Serge et al. (2023) was produced for specific use in the European Terranova project (terranova-itn.eu). The focus of these reconstructions was NOT to just reconstruct open land versus forest cover. **The above should be clarified here**.

We agree that the aims of different studies may be different and that this may affect methodological choices, such as spatial scale or what is being reconstructed (e.g. tree cover, PFTs, individual taxon abundances). However, our purpose here is NOT to discuss the purpose of each study or indeed how this affected the methodology. Rather, we are trying to explain what pan-European datasets are available that can be compared to our reconstructions. We have modified this text, in response to comments by Reviewer 1 and Reviewer 2, as follows:

<L56>. Several different techniques that have been applied to reconstruct regional and sub-regional vegetation in Europe using pollen such as biomization/pseudobiomization (e.g. Fyfe et al., 2015; Binney et al., 2017) or the application of MAT using plant functional types (e.g. Davis et al., 2014). Other studies have made reconstructions combining different approaches (e.g. Roberts et al., 2018) and by combining pollen-based reconstructions with simulated potential vegetation (Pirzamanbein et al., 2014). However, the most recent quantitative pan-European pollen-based reconstructions of Holocene vegetation changes have been made using the Landscape Reconstruction Algorithm (LRA) REVEALS approach (Sugita, 2007b, a) or the Modern Analogue Technique (MAT) (Overpeck et al., 1985; Guiot, 1990; Jackson and Williams, 2004; Zanon et al., 2018). The REVEALS method calculates regional vegetation cover based on modelled relationships between pollen abundance, estimated differences in species level pollen productivity and pollen transport, and differences in site characteristics. Initially used at individual sites or small regions (e.g. Gaillard et al., 2010; Nielsen et al., 2012; Marquer et al., 2014), REVEALS was first applied at a pan-European scale by Trondman et al. (2015) and later extended with additional sites, taxa and an improved temporal resolution by Githumbi et al. (2022). The most recent analysis by Serge et al. (2023), is based on 1607 records for 500-year intervals before 700 cal. BP and for the subsequent intervals of 700-350 cal. BP, 350-100 cal. BP and 100 cal. BP- present. They tested the impact of including 13 additional taxa (total n=46) on the vegetation reconstructions, producing maps of landcover and species abundance at recordcontaining 1º grid cells. In contrast, the MAT approach reconstructs past vegetation based on identifying modern analogues of fossil pollen assemblages, on the assumption that samples found in the fossil record that share a similar composition to those found in present-day pollen assemblages will have similar vegetation. Zanon et al. (2018) applied MAT to 2,526 individual fossil pollen samples from Europe to generate interpolated maps at 250-year intervals at 5 arc-minute resolution through the Holocene.

**R2:** 70-87: data demanding versus less data demanding methods: I would not "classify" methods as such. They are ALL data demanding, if RPP is not necessary something else is needed, in your case a good remote-sensed data on tree cover! Then all methods have their assumptions (very thoroughly stated in the case of REVEALS, Sugita 2007a) and difficult decisions to take in terms of data handling and interpretation. The only difficult issue with REVEALS is the need of RPPs. Obtaining RPPs is indeed time consuming although it can be done relatively fast if some money for field work and "man/woman power" is available. This was possible in China for which there is to date many RPP values for most of the major taxa; these were produced within the last 6-7 years. Moreover, this work could be realized in much less time with the technology at hand today, i.e. drones for plant surveys and automatic pollen counting. MAT, PB and your new method use other ways to account for the inter-taxonomical differences in RPP. All methods assume that RPP were stable through time. All methods are challenging, and they all have their pros and cons. None of them can be judged as more or less robust. We should rather use them as "ensembles", as is done with climate model simulations using several different models. Four reconstructions using four different methods/models (Europe) can be considered as an "ensemble". More reconstructions would of course be better; I am sure new models will constantly be created and more parameter data will be produced (e.g. RPP). My advice is therefore to avoid evaluation of the methods "against" each other but rather do a synthesis evaluation considering/acknowledging all pros and cons of each method. Here (lines 76-80), do not speak of data demanding method (REVEALS) versus other methods. Similarly, in abstract, discussion, and conclusions, do not grade methods. I do not think there is ONE best method/model.

We are not trying to classify the methods and we agree that all the methods require data. However, there is a difference between methods for which all the necessary data is available (i.e. regional modern pollen training data sets, good remote-sensing products) and methods such as REVEALS which require and are sensitive to regional data on RPPs. As Marie-Jose rightly points out these RPP data could of course be produced, given sufficient people-power and funding, but there are many regions of the world for which they currently not available and are unlikely to be available in the near future. Since our focus here is on the potential issues involved in application of these techniques, we will rewrite this paragraph as follows:

<Line 77>. The REVEALS approach requires, and is sensitive to, estimates of relative pollen productivity (RPP) and pollen fall speeds (FS) for individual species (Bunting and Farrell, 2022;

Githumbi et al., 2022; Serge et al., 2023). Landscape-level reconstructions are problematic if RPP and FS information are not available for relatively common taxa (Harrison et al., 2020). RPP values have been estimated for common taxa in Europe and China, and there are a limited number of studies from North America (see e.g. Wieczorek and Herzschuh, 2020) but are not readily available for other regions of the world. The MAT technique requires a large modern pollen data set for training purposes, but such data sets are now available for all regions of the world. However, the application of MAT involves a number of arbitrary decisions including the choice of analogue threshold (i.e. how similar modern and fossil assemblages must be to be considered analogous), and the number of analogues used (Jackson and Williams, 2004). Techniques designed to minimise the number of samples for which no analogues are found, such as grouping species into plant functional types (PFTs), introduce further uncertainties since the allocation of pollen taxa to PFTs is often ambiguous (Zanon et al., 2018).

**Additional refererences:**

Harrison, S. P., Gaillard, M.-J., Stocker, B. D., Vander Linden, M., Klein Goldewijk, K., Boles, O., Braconnot, P., Dawson, A., Fluet-Chouinard, E., Kaplan, J. O., Kastner, T., Pausata, F. S. R., Robinson, E., Whitehouse, N. J., Madella, M., and Morrison, K. D.: Development and testing scenarios for implementing land use and land cover changes during the Holocene in Earth system model experiments, Geosci Model Dev, 13, 805–824, https://doi.org/10.5194/gmd-13-805-2020, 2020.

Wieczorek, M. and Herzschuh, U.: Compilation of relative pollen productivity (RPP) estimates and taxonomically harmonised RPP datasets for single continents and Northern Hemisphere extratropics, Earth Syst Sci Data, 12, 3515–3528, https://doi.org/10.5194/essd-12-3515-2020, 2020.

**R3:** 106-108: there are more vegetation classes that could be considered as "non-natural" such as planted forest, cultivated trees, even grazed land if strongly fertilized isn't "natural", as well as ley. You perhaps rather mean "non-pollen producing land" in the case of crops like most cereals (except rye) that do not release pollen as long as they are in the field, i.e. before collected and treated to get the grains. But there are other crops. Do the remote-sensed data provide details on the crops. Do all the crop areas you deleted correspond to areas that do not produce much pollen?

In our response to a comment from Reviewer 1, we amended the text to exclude the "non-natural" terminology, given its potential ambiguity as follows:

<L.106>. A composite map of modern tree cover for the region 12°W to 45°E and 34-73°N was generated by averaging annual percentage tree cover data from Copernicus annual land cover maps from 2015 to 2019 (Buchhorn et al., 2020a, e, d, c, b), after removing cells dominated (> 50%) by

other land-cover classes, including bare ground, built up areas, moss or lichen, permanent water, snow, and crops (Fig. 2A).

**R4:** 126: Githumbi et al. did not exclude large bogs, which would certainly have been better. But the decision came from Trondman et al. 2015 in which it was decided not to exclude sites based on this criterion; instead, cells including large bogs were emphasized as providing less reliable results. The correct reference for large bogs being not recommended for REVEALS reconstructions, even when multiple sites are used for one reconstruction, is Trondman et al., 2016 in VHA.

We did not mean to imply that Githumbi et al. (2022) excluded bogs, but rather that they urged caution in using these records, specifically "…REVEALS estimates of plant cover using pollen assemblages from large bogs should only be interpreted with great caution ...". Whether they should or should not be used for REVEALS-based reconstructions is not really the issue. The key point here is that we include taxa that grow on bogs and we felt that this could bias our tree cover reconstructions. In response to a comment from Reviewer 1, we have amended this text as follows:

<Line 126>. However, bog records with a radius $\geq$ 400m were excluded from the analysis. Githumbi et al (2022) indicated that caution was necessary in interpreting REVEALS vegetation reconstruction estimates based on large bogs and, given that we included taxa that grow on bog surfaces in our analysis (see below), we excluded large bogs to reduce the potential for these to bias the regional vegetation reconstructions.

However, given that we do not wish to comment on the applicability for REVEALS-based reconstructions but are simply justifying the exclusion of large bogs for our reconstructions we will further modify this to:

<Line 126>. However, bog records with a radius $\geq$ 400m were excluded from the analysis because we included taxa that grow on bog surfaces in our analysis (see below), and the exclusion of large bogs reduces the potential for these taxa to bias the regional vegetation reconstructions.

**R5:** 130: I would also refer here to Marquer et al. 2020, QSR 228-106049.

We will include this reference, as follows:

<Line 128>. Finally, since upslope pollen transport is known to increase the proportion of non-local pollen at high-elevation sites (Fall, 1992; Ortu et al., 2008, 2010), and the complex topography of mountainous areas also impacts pollen transport (Markgraf, 1980; Bunting et al., 2008; Marquer et al., 2020; Wörl et al., 2022), we excluded 236 site records above 1000m.

**Additional reference**

Marquer, L., Mazier, F., Sugita, S., Galop, D., Houet, T., Faure, E., Gaillard, M.-J., Haunold, S., de Munnik, N., Simonneau, A., De Vleeschouwer, F., and Le Roux, G.: Pollen-based

reconstruction of Holocene land-cover in mountain regions: Evaluation of the Landscape Reconstruction Algorithm in the Vicdessos valley, northern Pyrenees, France, Quat. Sci. Rev., 228, 106049, https://doi.org/10.1016/j.quascirev.2019.106049, 2020.

144-154:

- **R6:** It is true that the area represented by a REVEALS reconstruction of plant cover using pollen data from large lakes might correspond to Zmax as defined in Sugita 2007a "distance within which most pollen comes from", i.e. the maximum distance from a large lake from which pollen is coming and deposited in that lake (my words). However, this might not be true for small sites; in fact, we do not know as this has never been tested because single small sites are not appropriate for REVEALS applications. Small sites can only be used for REVEALS reconstructions if they are used together, i.e. within a grid cell, and a mean REVEALS estimate is calculated. That's fine you use Zmax for all your sites, but **you should acknowledge that you ASSUME that Zmax is also the area represented by a REVEALS reconstruction using pollen from a small lake, but this has never been tested**. A longer discussion on the "spatial scale" of a REVEALS reconstruction can be found in Li et al. 2020 in ESR, page 5, upper left column: "*In theory and practice, however, the strict definition of the pollen source area is difficult for REVEALS application. Sugita (2007a) defined it as the area within which most of the pollen comes from. Simulations and previous empirical studies (e.g. Sugita, 2007a, b; Hellman et al., 2008a; Sugita et al., 2010; Mazier et al., 2012) have indicated that, when the radius of the source area defined varies from 50 km to 400 km, the REVEALS results of regional vegetation reconstruction do not change significantly. The basin size is potentially important for REVEALS-based estimate of regional vegetation because differences in basin size among sites can lead to a significant site-to-site variation in the pollen assemblages. However, as long as the multiple study sites are located within a region that satisfies the first assumption as described above (no gradients in the overall vegetation composition), the averaged REVEALS estimates effectively represent the regional vegetation composition as demonstrated in Hellman et al., 2008a. The accuracy of the reconstructed vegetation against the observed vegetation composition was assessed for areas of 50 km × 50 km and 100 km × 100 km around each site in two regions of southern Sweden. The pollen records used are from 5 large lakes in each region, thus 10 lakes in total, that vary in size between 76 ha and 1965 ha. The results support the main conclusions and implications for the REVEALS application based on the theory and the simulations described in Sugita (2007a). Such evaluation is an essential step for credible application of the REVEALS model.*"

We are at a loss to see how these comments refer to the cited paragraph, which describes the use of the source area formula developed by Prentice (1985). The choice of the 70% threshold was also taken from that paper. We used median fall speeds dereived from Githumbi et al (2022) and Serge et al (2023) because the Prentice (1985) formulation used species specific values which were not

available for all taxa included in our analysis. We will make this point clearer by modifying the text as follows:

<Line 144>. The source area for each record, and hence the appropriate area for the calculation of mean tree cover, was calculated using Prentice's (1985) source area formula for 70% of pollen, and lake or bog area from the SMPDS. The original source area formula used species-specific FS values but, since these were not available for all the taxa used in our analysis, here we used the median FS (0.03) from Githumbi et al. (2022) and Serge et al. (2023) since the tree cover map represents the broad species community around each record location.

- **R7:** Why use 70% of pollen rather than 80% or 90%?

Please see response to R6. Note that we tested the 70% criterion and found that it gave plausible results.

- **R8:** Why use the median FS 0.03 and not simply estimate the distance for the lightest pollen type you use in your reconstruction, which would be Zmax? If you use 0.03 you get a Zmax for a vegetation composed of taxa such as *Pinus*, *Ulmus*, *Buxus*. But your total vegetation is composed of taxa with pollen grains that come from much longer distances (most herbs but also several common tree taxa with lower FS values). Why do you assume/think that this isn't important to define the area around each of your sites, i.e. the area that is represented by the pollen assemblages in those sites?

Our approach is based on matching the site source area to the underlying tree cover maps. The choice of a median FS is a pragmatic attempt to reflect the average vegetation composition surrounding the site area. We take your point that by using this median we may not be fully capturing the source of the lighter pollen, especially in more open environments. But taking a source area that is too large would also be problematic, as heavier pollen at the fringes of this area would not reach the site or be reflected in the pollen assemblage.

**R9:** 200-203: **I do not fully understand how you handled the Serge et al. REVEALS estimates of tree cover for comparison with your reconstruction**. There is doubtless a problem in comparing single site reconstructions within the site's "70% Zmax" with REVEALS reconstructions representing at minimum the area of the 1-degree grid cell including the pollen sites used in the reconstruction. Wouldn't it be fairer (for each method!) to calculate the median tree cover from your single site reconstructions covering +/- one or several 1-degree grid cells of the REVEALS reconstruction before comparing the results? The REVEALS estimates are mostly based on several sites in each grid cell. If you compare your results for each site with the median forest cover from several grid cells in the REVEALS reconstruction, you'll compare the vegetation cover between two areas of different sizes, smaller size for your reconstruction than for the REVEALS reconstruction (size of several 1- degree

grid cells). Please, acknowledge this issue. – Now I see in Supplement S6 that you seem to have done what I am suggesting above as fairer, although I am not sure.

Comparing the different reconstruction values is a challenge, given the different methods and scope of the reconstructions. The ideal comparison would be to use site-based estimates for each since this allows for potential variability between nearby sites that can be masked when using median values for a 1º grid cell, as in the REVEALS estimates. Supplement S6 investigates the implications of this to some extent, by averaging tree cover values for records that occur in the same grid cells. But this still leaves the problem of differences between reconstructions potentially being the result of having different numbers of sites in each grid cell. We will try to clarify this, both in this paragraph (L200-207), and in new text in the discussion, as follows:

<Line 200>. For each of the 1º grid cells in Serge et al. (2023), tree cover was calculated from the sum of the appropriate vegetation types. Time series of the change in median tree cover were constructed using median tree cover corresponding to the pollen source area of each of our individual modern reconstructions. As the Serge et al. (2023) and Zanon et al. (2018) data is available in gridded format, comparison with our site-based predictions is not straightforward. Where the site location source areas straddled multiple grid cells, a median was calculated, weighted by the proportion of grid cell coverage using R package *exactextractr (*function*: exact_extract)* (Baston, 2023). The tree cover time series for the Zanon et al. (2018) and Serge et al. (2023) data were initially constructed using all of the extracted tree cover values for each of our model training site locations. However, since there can be multiple sites within some of these grid cells, we tested whether affected the comparisons by taking an average of extracted tree cover values for locations sharing the same grid cell values from Zanon et al. (2018) or from Serge et al. (2023), and using this to create new time series for these two reconstructions.

We will modify the text in the discussion as follows:

<Line 363>. Our reconstructions show that tree cover peaked in the mid-Holocene period, with median tree cover ca. 40% greater than at the beginning of the Holocene. This general pattern is shown by the REVEALS and MAT reconstructions, and is also visible in plant functional type (Davis et al., 2015) and pseudo-biomization reconstructions of vegetation cover (Fyfe et al., 2015). Despite the similarities in median values between our reconstructions and those calculated from the REVEALS and MAT reconstructions, our site-based estimates are not fully comparable with the gridded estimates provided by the REVEALS reconstruction and the gridded and interpolated values provided by the MAT reconstructions. Nevertheless, the similarities give some support to the overall robustness of our reconstructions.

Figures 3 and 4:

- **R10**: I am not astonished that your reconstruction performs better than other reconstructions given that you use the same source of forest data to establish your model and test it – Note that I understand you haven't used the same sites to create the model and test the model, of course!!! – Your model is entirely dependent on the forest data you have used, and there must be good chances that your predictions will be relatively good when compared with the same data source of forest cover.

We compared the reconstructions to the Copernicus data set because this is considered to be the best available. There is no guarantee that the final model would reproduce the training data, as you point out. We used leave-one-out cross validation to test the model performance (Methods section 2.2).

- **R11:** Further, REVEALS (Fig. 4C) is closer to your reconstruction (Fig. 3B) than Zanon (Fig. A4) in terms of the spread of the points between low and high predicted % cover. Here, as said above, I am not sure to understand how you made the comparison between your results and Serge et al. The points in Figures 3 and 4 represent your sites. **I do not see how the REVEALS values can be constrained to represent the plant cover for your "pollen source area" (i.e. Z max for each of your site) as these REVEALS estimates are valid for at least the 1-degree grid cell including the sites used for the reconstruction, i.e. a much larger area; it is NOT valid for a much smaller area.** You write on line 269-271 "*The correlation between etc…. is only 0.5. This is partly caused etc….. but even when taking this into account etc…. were still lower (0.59)*". **From Supplement S6 I understand you took the scale of reconstructions into account, but I am not sure how**. In any case it increases the correlation, which is good news! **Could you please clarify**.

For each site location from our reconstruction, we extracted the median tree cover value from each time window for the area around each site. If a site source area is fully contained within an individual grid-cell, then the value for that cell is returned and subsequently compared to the observational value. If the source area straddles several grid cells, the value returned is the median, calculated to take into account the proportion of each grid cell value covered by the site-specific source area. Supplement 6 shows the implications of restricting the sites to one per grid-cell, to potentially limit some of the double counting of values from the REVEALS (and MAT) reconstructions. Here, where sites share the same grid-cell, the average of the median value for each site within a grid-cell is calculated. ==Please see our response to R9 for changes to the method text to clarify this==.

- **R12:** Does your model produce error estimates on the predicted forest cover? If so, why are they not provided in the graphs; similarly, why not use the error estimates provided with the REVEALS-based forest cover?

We didn't include error estimates for the site predictions. However, we agree that this would be a useful inclusion, both for the site estimates and as another way to reflect uncertainty in the median reconstructed tree cover. We have included the following additional text and supplementary figure: <Line 214> …binned in 200-year bins. Standard error estimates for site predictions were calculated through the application of a bootstrapping approach, with 1000 resamples of the model training data used to generate models, and equivalent quantile mapping adjustment, which were then applied to the fossil pollen data.

<Line 288> This same general pattern is shown when considering changes in mean tree cover (Supplementary Information: S7), different LOESS smoothing (R package *locfit* Loader, 2020) of the median tree cover value (Fig. 5B), and based on median tree covers reconstructed using the model bootstraps used to generate reconstruction standard errors (Supplementary Information: S8).

**S8: Median reconstructed tree cover, with bootstrapped models**

In order to generate reconstruction standard errors, the predictive model that linked observed tree cover to modern pollen data was generated 1000 times by bootstrapping the modern pollen data. These models were also used to generate the equivalent number of quantile mapping adjustments, by relating model predictions using the full dataset to observations. Together these elements were used to produce 1000 different reconstructions of tree cover for each fossil data sample, with prediction standard error calculated by sample and averaged by 200-year bin. As well as using the bootstrapped reconstructions to generate the standard error, we can use the median of these bootstraps as another way of assessing the confidence in the median reconstruction. Supplementary Figure 6 shows the median tree cover estimate, together with the bootstrapped medians of tree cover based on the different models and quantile mapping adjustment generated. Although the bootstrapped medians follow the same general pattern as the reconstructed median, maximum tree cover values for the reconstruction are generally on the higher side compared with the bootstrapped medians, implying that some training samples may have a larger influence on the generated model than others.

[Figure]

**Supplementary Figure 6: Median reconstructed tree cover for Europe from 12,000 to 0 cal. BP, with 95% confidence intervals for models generated through 1000 bootstrap resamples of model training data**

**R13:** Figure 5B: Can't see the different colors for the smoothing half-width in the graph, probably because the curves are very similar? Find another way to illustrate this, or have only -none and 500-year, and comment in the text for the other two?

The curves are very similar and we agree it is redundant to have the 1000 and 2000-year half-widths within in the figure. We will adjust the figure and amend the text as follows:

[Figure]

<Line 287>. This same pattern is shown when considering changes in mean tree cover (Supplementary Information: S7) and LOESS smoothing (R package *locfit* Loader, 2020) of the median tree cover value (Fig. 5B)

**R14:** Figure 7: Could the high values of forest cover in Zanon 10-8 k be due to over-representation of Pinus and Betula? In your reconstruction there is the same tendence.

Since Pinus and Betula are included in all three reconstructions, it is difficult to see why their over-representation between 10-8 ka in the Zanon et al. reconstruction would be due to this. However, diagnosing the cause of this peak is beyond the scope of this paper. We do not see the double peak shown by Zanon et al. in our reconstructions, and our reconstructions in general show lower values of tree cover than Zanon et al, despite including Pinus and Betula.

**R15:** 370-376: You write: "*This could reflect the conservative nature of our moder-day tree cover model*". YES, I think this might well be the major reason to this difference. Can't this also explain the same "phenomenon" in the MAT reconstruction (Zanon)?

Yes, as we mention in the subsequent sentence, Zanon et al. (2018) also indicate that they likely underestimate tree cover at higher levels of tree cover.

379-385:

- **R16:** I do not understand this reasoning. First, you did not account for all "anthropogenic land use", you only excluded non-pollen producing areas, including crops (see also my comment above on methods and in relation to crops). You write "We account for this (land use/" non-natural vegetation") in defining modern source areas in our model, since the pollen only provides evidence of the natural vegetation." This sounds very odd. Pollen provides evidence of natural vegetation AND human-influenced vegetation, i.e., not only crop cultivation but also grazing, managed woodland, planted forests etc…. It is the purpose of the REVEALS model and MAT and pseudo-biomization (PB). The purpose is NOT to reconstruct natural forest cover, but the "actual" forest cover. **Please clarify, this section is very confusing. What you did was to exclude non-pollen producing areas, which is good (I only have concerns about crops, see comments above). You did NOT exclude anthropogenic land use.**

We agree that this text could be clarified. We have amended the text as follows:

<Line 379>. The major difference at the pan-European scale is the reduction in tree cover from ca. 2000 cal. BP to present, which is less marked in our reconstructions and more consistent with the maps of observed tree cover. The observed tree cover values used in the model construction exclude areas dominated by land-cover types such as built areas or areas dominated by crops. We account for this in defining modern source areas in our model. Not accounting for changes in these other land-cover types, which through anthropogenic land use have increased substantially over the past 1000 years (Klein Goldewijk et al., 2017) would result in a steeper decline in tree cover, as seen in the other two reconstructions.

- **R17:** Note that in Roberts et al 2018 (Scientific Reports, Figure 2), the REVEALS and PB (closed sum) reconstructions agree with each other and agree with the Corine remote-sensed forest cover. Roberts et al. (2018) write "*A second means of testing the different forest reconstructions is to examine how well they match modern forest cover for the same grid cells. While remotely sensed estimates of forest might be expected to offer the clearest results and the most rigorous test, in fact the Corine and Forest Map 2006 data have strongly different outcomes, i.e. 45% and 29% modern forest cover, respectively. This inconsistency partly reflects the ontological question of "what is a forest?" Corine uses distinct land-cover classes, and land classified as forest may*

*include some open areas as the minimum required crown cover for a forested class is only 30%. The Forest Map 2006 is based on a minimum 50% tree crown cover with 5m used as a minimum height of trees. It also highlights the epistemological problem that differences in spatial resolution of measurement can fundamentally alter results30, in this case between 25m and 100m measured spatial resolution. An alternative data source for modern forest cover derives from surface pollen samples. We have transformed the surface pollen data set for Europe27 using both variants of the PBM, which leads to modern forest cover estimates of 49% (PBMsc) and 54% (PBMlcc). Overall, most estimates of modern forest cover for the grid cells used by Trondman et al. 10 are between 45% and 49%; that is, close to that reconstructed for the 100 to −65 BP REVEALS time window."* This is the kind of issues you should acknowledge in this discussion. Please see my major comment under "General comment" above, i.e. REVEALS reconstructs tree cover, while your model reconstructs FOREST cover based on a specific definition of FOREST.

==In fact, our reconstruction is of tree cover not forest cover. In response to a comment by reviewer 2, we have amended the text throughout to make this clear and have added an explanation of what we mean by tree cover in the Introduction.==

419-424:

- **R18:** In this discussion, you do not attempt to explain why REVEALS estimates > 65% tree cover around 6-5.5 k BP while both your model and Zanon's MAT predict forest cover < 50% (40-45%) 8.5- 5 ka (your model) respectively 9.5- 4 ka BP (Zanon). I would emphasize here the major difference between MAT, your model and REVEALS in terms of WHAT is reconstructed. REVEALS estimates TREE cover and NOT FOREST cover. The authors using REVEALS may define taxa as trees in various ways. You need to consider what taxa are defined as trees in Serge et al. and think about weather these taxa may belong to land-cover types your remote-sensed data define as non-forest vegetation.

==Please see response to comment above==.

- **R19:** You write (412-424) *"……the more rapid decline in tree cover during the last millenium shown ..... , and shown more dramatically in the Zanon …. And Serge …. reconstructions, is more difficult to explain - …… Human influence on the landscape MAY help explain etc….."*. I do not understand why you are so careful/doubtful on whether human impact may explain tree/forest cover decline from 6k, 5.5 k, 4k (depending on the region) and more so from 2k BP. This is documented and has been tested in a large number of publications by palaeoecologists, archaeologists, historians, etc… over past decades; I can't see what is problematic or controversial with this. In Strandberg et al. 2023 (CP), Figure 1 (based on Githumbi et al. REVEALS reconstruction, see figure copied below) shows clearly the increase in mean and median tree cover in three major biomes of Europe (note that we did not separate the Atlantic

region, which would indeed have been interesting) from mid Holocene, accelerating around 2 k BP, and more so around 1 k BP. Strandberg et al. (2023) write: "*The recent pollen-based reconstruction of land cover in Europe (spatial resolution of 1°; Githumbi et al., 2022) suggests that the earliest of the two major deforestation episodes before the start of the Modern period (1500 CE (0.45 ka) – present) took place between ca. 4 and 2.5 ka, i.e. the period during which the Bronze Age culture expanded from southeastern (Turkey, Greece) to central and western Europe (Mediterranean area included) and northern Europe (Champion et al., 1994; Coles and Harding, 1979). The second deforestation episode (before the Modern time deforestation) occurred ca. 0.9–0.5 ka, during the Middle Ages (ca. 500 (1.45 ka)–1500 CE in most of Europe, started 1050 CE (0.9 ka) in northern Europe) (Fig. 1a). The difference in open land cover between 4 and 2.5 ka of ca. 10 % (in either mean or median cover; Fig. 1a) is assumed to represent deforestation of Europe by Bronze Age cultures. This change in the land cover of Europe was also explained by deforestation for agriculture in the study of Marquer et al. (2017). If we consider …etc., the Bronze Age deforestation corresponds to an increase in open land cover by 200 % since 4 ka. etc. … (Githumbi et al., 2022).*" And "*The time around 3 ka (the Bronze Age) was also pinpointed as the time when "the planet [was] largely transformed by hunter-gatherers, farmers, and pastoralists", as suggested by an archaeological global assessment of land use from 10 ka to 1850 CE (ArchaeoGLOBE Project, 2019).*" There is no doubt that deforestation caused by land use did strongly influence forest/tree cover in Europe over the last ca. 3000 years.

[Figure]

**Figure 1.** Density plots of REVEALS-based open land (% cover) for **(a)** all European grid cells, **(b)** the boreal zone, **(c)** temperate Europe, and **(d)** Mediterranean Europe. Full black lines represent the mean value of open land cover, and dashed green lines represent the median value. Panel **(e)** is illustrative of how the density plots are derived for one time window, showing data values for all grid cells (opaque grey circles), the derived density function as a curve and expressed on the colour gradient, and the mean and median values. Key time intervals (6, 2.5 ka, and the PI period) are highlighted on panels **(a–d)** by the grey bars.

There have indeed been many studies that have attributed observed changes to human activities, but these are largely based on correlations between two variables, e.g. vegetation changes and archaeological records. Correlation is not causation. It is true that the Strandberg et al. paper cited above compares REVEALS-based reconstructions with potential vegetation reconstructions based on using a model (LPJ-GUESS), but these are (a) only as good as the vegetation model itself, (b) conditioned by the climate (and $CO_2$) inputs used to drive the model, and (c) do not account for climate-induced changes in vegetation disturbance such as fire. Note that over the past 12,000 years changes in $CO_2$ of themselves would have had a non-negligible impact on the vegetation through changing water-use efficiency. The only way to establish rigorously the causes of reconstructed changes in vegetation during the Holocene would be to model them quantitatively using robust reconstructions of all the variables that could impact the vegetation, i.e. climate changes, CO2 changes, changes in fire regimes and changes in human activity. This is a goal that partially underpins our efforts here to reconstruct tree cover. We are cautious about attributing any of the reconstructed changes to human activities until this quantitative assessment is made, although we acknowledge (given the seemingly robust evidence of human population changes over the last millennium available from HYDE) that at least on that time frame human activities look to be a plausible explanation.

**Abstract and conclusions:**

**R20:** Please revise the abstract and conclusions following the revisions you might consider making in response to the comments above. Among other, the statements "*our approach is more robust and less data-demanding than previously applied methods*" *(abstract) and* "*Our simple approach produces ….. etc…. using more complex methods, and thus provide a less data-demanding approach… etc… of the world*" (conclusions) should be revised. There are no less/least data-demanding method/model and/or best (most robust) method/model; there are several possible methods/models that all are data-demanding and have their pros and cons.

Please see our response to R2 above and the proposed modification to the text in the Introduction.

We will modify the last sentence of the abstract as follows:

==The reconstructed patterns of change in tree cover are similar to those shown by previous reconstructions, but our approach is relatively simple, only requires readily available data and could therefore be applied to reconstruct tree cover globally.==

We will modify the last sentence of the conclusions as follows:

Our simple approach produces similar reconstructions of the trends in tree cover during the Holocene reconstructed using more complex methods, and since it only requires readily available data could be used to reconstruct tree cover in other regions of the world.

---

## Author Comment (AC1)

We thank the reviewer for their constructive comments on this article. Our responses to individual comments are highlighted in blue, with any proposed changes highlighted in red.

General

**Reviewer comment 1 (R1):** There is a need for assessing past forest cover change from pollen regionally and the REVEALS model requiring knowledge on pollen productivity may not be the method of choice where that information is lacking or incomplete. Thus it is useful to explore other avenues and the manuscript by Sweeney et al. does that. It also adds interesting comparisons between the European estimates produced applying the REVEALS model and the modern analogue technique. While I welcome the attempt of applying a regression, I have my doubts on the choice of predictor variables. The authors should demonstrate how % needle leave and the Shannon index improve a regression model for overall tree cover. I can see how elevation improves the model in the current situation but have my doubt that this variable will improve past reconstructions. Instead using information on over and underrepresented pollen could perhaps make this a real winner.

The choice of predictor variables was explained in the paper (lines 168-177), but we agree that it would be useful to show how the inclusion of variables contributes to improve the model. We will include an additional table - Table 2 (see below)- highlighting the change in model AIC value when excluding specific variables (and their interactions).

Elevation is a significant variable describing the present level of tree cover, both in itself and in terms of the interactions with variables. Whilst we agree that the influence of elevation may have changed through time with changes in climate being expressed more strongly at higher elevations, the same could be said of each of our explanatory variables to some extent. This is perhaps a general point that should be emphasised within the Discussion section: reconstructing the past relies on an assumption that present day relationships are applicable to past situations. This point is equally applicable to the other reconstruction techniques. We agree that it would be interesting to include information on over- or under-represented taxa but this is difficult to do given the limited number of taxa for which RPP values are available, as well as raising questions regarding the appropriate RPP values for species across Europe (as with the REVEALS approach).

We will modify the text to cover these points as follows:

<L.255>. The influence of each variable on the quality of the statistical model is shown in Table 2, with the change in AIC value based on the removal of each variable. These values include the removal of interaction, polynomial and precision terms associated with each variable as applicable. Although AP and SP might be expected to be the most important explanatory variables, the model

only using AP and SP has an AIC value 568 greater than the final model, and Cox-Snell $R^2$ of only 0.27 (Table 2). Thus the inclusion of the other variables is important in fitting the final model.

**Table 2: Change in modern model AIC values and Cox-Snell $R^2$ model values when excluding specific variables (exclusion includes interactions, polynomials and precision variables) and for a model with only arboreal and shrub pollen percentage**

| Model | Δ AIC | Cox-Snell $R^2$ |
|---|---|---|
| Final model | 0 | 0.60 |
| excluding AP | 165 | 0.51 |
| excluding SP | 31 | 0.58 |
| excluding %needleshare | 121 | 0.55 |
| excluding AP Shannon index | 396 | 0.41 |
| excluding lake or bog site | 56 | 0.57 |
| excluding elevation | 204 | 0.48 |
| AP and SP model | 568 | 0.27 |

<L.426>. Our simple modelling approach yields a reasonably robust picture of changes in tree cover through the Holocene, largely consistent with known changes in climate. As with other statistical reconstruction techniques, it is predicated on the assumption of stationarity between tree cover and the explanatory variables. This may be problematic for variables such as elevation, where changes in elevational lapse rates (Mountain Research Initiative EDW Working Group, 2015) or atmospheric circulation patterns (Bartlein et al., 2017) could affect the relationship, but is less likely to be an issue for explanatory variables that reflect biophysical controls on pollen transport and deposition such as basin type or proportion of needleleaf trees. Our approach is less data-demanding...

**Additional references**

Bartlein, P.J., Harrison, S.P. and Izumi, K., 2017. Underlying causes of Eurasian mid-continental aridity in simulations of mid-Holocene climate. *Geophysical Research Letters* 44, doi: 10.1002/2017GL074476.

Mountain Research Initiative EDW Working Group. Elevation-dependent warming in mountain regions of the world. *Nature Clim Change* **5**, 424–430 (2015). https://doi.org/10.1038/nclimate2563

**R2:** It is also not clear to me in which way this regression model improves upon the modern analogue technique requiring the same input information and seemingly yielding a similar performance.

As we point out in the Discussion (line 433), we cannot assess how well the MAT technique performs quantitatively because Zanon et al. (2018) do not provide reconstructions at individual sites. Although the overall pattern of tree cover changes in the two reconstructions is similar, the magnitude of the recent decline shown by Zanon et al. (2018) appears to be too large and our reconstruction appears to be more realistic (Line 335). The main advantage that we see of our approach compared to the MAT reconstructions, as we say in the Discussion, is that it obviates the necessity to make methodological choices, such as the number of analogues used, that have been shown to affect MAT-based reconstructions.

**R3:** The manuscript is not explaining how the proposed regression model reduces the bias of simply using arboreal percentage, which may be dominated by pine and birch versus elm and lime.

We do not attempt to account for over- or under-representation of specific taxa, except in the inclusion of proportion of needleleaf as a predictor to account for gross differences between conifer and broad-leaved trees as shown by Serge et al. The new Table (see response to R1) shows that the final model is better than a model based on arboreal percentage data alone. However, we will add a new supplementary section (S6 – resulting in current S6 becoming S7 etc.) showing the AP % plotted against observed tree cover in the modern day, and we will add a comment about this issue in the main text as follows:

<L.260>. This correlation value compares favourably to the correlation between raw AP% and observed tree cover values (0.54): raw AP% values tend to overestimate observed tree cover (Supplementary information: S6).

**<Within supplement> S6: Arboreal pollen percentages compared with observed tree cover percentages**
Raw arboreal pollen percentages tend to overestimate tree cover, with the range in AP% greater for lower observed tree cover groups (Supp. Fig. 3).

[Figure]

**Supplementary Figure 3: Arboreal pollen percentage compared to observed tree cover: A - AP% compared to observed tree cover for each record; B - Differences between AP% and observations (residual), in bins of observed tree cover percentage**

**R4:** My second concern with the manuscript is the lack of appropriate recognition and citation of databases and initiatives that collected and curated the pollen data used here. Most of the modern and downcore pollen data used here was initially made available by the EPD/Neotoma or PANGAEA with a cc by 4 license requiring attribution and citation of this initial data release. Please see the recent discussion of the manuscript by Schild et al. (https://essd.copernicus.org/preprints/essd-2023-486/#discussion).

Much of the modern and fossil data used in this paper were derived from the EPD/Neotoma or PANGAEA, but the SMPDSv2 and the SPECIAL-EPD data sets contain additional data provided by individual palynologists (and duly acknowledged in the documentation of those data sets). Both data sets have also been cleaned to correct errors in EPD/Neotoma files. The most important improvement offered by these data sets, compared to the original files, is the construction of standardised Bayesian age models for all of the records. They also include additional metadata which is important for the interpretation and current use of these data. Thus, we think it is appropriate to refer to the SMPDSv2 and the SPECIAL-EPD as the source for the information used in this paper. However, and although the source of specific data sets is acknowledged in the documentation of those data sets, we take the point that it would be appropriate to include some further explanation about the sources in the current paper. We will therefore modify the text describing these data sources as follows (note that the inclusion of an additional Supplementary section changes S1 to S2 etc.):

<Line 114>. Modern pollen data (Fig. 2B) was obtained from version 2 of the SPECIAL Modern Pollen Dataset (Villegas-Diaz and Harrison, 2022). This data set was created from multiple different published regional datasets, from data repositories (Neotoma, PANGAEA) or directly from data collectors/authors (see Supplementary Information: S1 for sources and citations) but employs a

standardised taxonomy, and includes improvements to metadata and age models. The data set was further amended for the current analysis by including updated meta information (see *Code and data availability*).

**<Within supplement> S1: Sources for SMPDSv2 data**

Supplementary Table 1 provides the metadata source for the SMPDsv2 data, together with the number of entities/records and citations for each source.

**Supplementary Table 1: List of SMPDSv2 data sources and references**

| Source (metadata table) | Number of entities | Publications |
|---|---|---|
| AMSS | 38 | Jolly et al., 1996; Julier et al., 2019, 2018; Lebamba et al., 2009 |
| APD | 90 | Vincens et al., 2007 |
| Australasian pollen | 1540 | Adeleye et al., 2021b, 2021a; Beck et al., 2017; Field et al., 2018; Fletcher et al., 2014; Herbert and Harrison, 2016; Luly, 1993; Luly et al., 1986; Mariani et al., 2017; McWethy et al., 2010, 2014; Pickett et al., 2004; Prebble et al., 2019 |
| BIOME6000 Japan | 94 | Takahara et al., 2000 |
| Blyakharchuk | 144 | Author: Tatiana A. Blyakharchuk |
| Bush et al., 2021 | 636 | Bush et al., 2021 |
| CMPD | 4208 | Chen et al., 2021 |
| Dugerdil et al., 2021 | 48 | Dugerdil et al., 2021 |
| EMBSeCBIO | 149 | Harrison et al., 2021 |
| EMPDv2 | 3508 | Davis et al., 2020 |
| Gaillard et al., 1992 | 124 | Gaillard et al., 1992 |
| Harrison et al., 2022b | 3 | Harrison et al., 2022b |
| Herzschuh et al., 2019 | 595 | Herzschuh et al., 2019 |
| IBERIA | 243 | Harrison et al., 2022a |
| Neotoma | 6702 | Williams et al., 2018 |
| Phelps et al., 2020 | 106 | Phelps et al., 2020 |
| SMPDSv1 | 6345 | Harrison, 2019 |
| Southern Hemisphere pollen | 76 | Black, 2006; Dodson, 1978; Dodson and Intoh, 1999; Haberle, 1993, 1996; Hope, 2009; Hope et al., 1998, 1999; Macphail, 1975, 1979, 1980; Macphail and McQueen, 1983; Macphail and Mildenhall, 1980; Norton et al., 1986; Prebble et al., 2010; Shulmeister et al., 2003 |

**Additional References (supplement)**

Adeleye, M.A., Haberle, S.G., Harris, S., Hopf, F.V.-L., Connor, S., Stevenson, J., 2021a. Holocene heathland development in temperate oceanic Southern Hemisphere: Key drivers in a global context. Journal of Biogeography 48, 1048–1062. https://doi.org/10.1111/jbi.14057

Adeleye, M.A., Haberle, S.G., McWethy, D., Connor, S.E., Stevenson, J., 2021b. Environmental change during the last glacial on an ancient land bridge of southeast Australia. Journal of Biogeography 48, 2946–2960. https://doi.org/10.1111/jbi.14255

Beck, K.K., Fletcher, M.-S., Gadd, P.S., Heijnis, H., Jacobsen, G.E., 2017. An early onset of ENSO influence in the extra-tropics of the southwest Pacific inferred from a 14, 600 year high resolution multi-proxy record from Paddy's Lake, northwest Tasmania. Quaternary Science Reviews 157, 164–175. https://doi.org/10.1016/j.quascirev.2016.12.001

Black, M., 2006. The fire, human and climate nexus in the Sydney Basin, eastern Australia (Doctoral dissertation). University of New South Wales, New South Wales.

Bush, M.B., Correa-Metrio, A., van Woesik, R., Collins, A., Hanselman, J., Martinez, P., McMichael, C.N.H., 2021. Modern pollen assemblages of the Neotropics. Journal of Biogeography 48, 231–241. https://doi.org/10.1111/jbi.13960

Chen H-Y., Xu, D-Y., Liao M-N., Li. K., Ni, J., Cao, X.-Y., Cheng, B., Hao, X-D., Kong, Z-C.,, Li, S-F., Li, X-Q.,, Liu, G-X., Liu, P.-M., Liu, X.-Q., Sun, X-J., Tang, L.-Y., Wei, H-C., Xu, Q-H., Yan, S., Yang, X-D., Yang Z.-J., Yu, G., Zhang, Y., Zhang, Z-Y., Zhao, K-L., Zheng, Z., Herzschuh, U., 2021. A modern pollen dataset of China. Chinese Journal of Plant Ecology 45, 799. https://doi.org/10.17521/cjpe.2021.0024

Davis, B.A.S., Chevalier, M., Sommer, P., Carter, V.A., Finsinger, W., Mauri, A., Phelps, L.N., Zanon, M., Abegglen, R., Åkesson, C.M., Alba-Sánchez, F., Anderson, R.S., Antipina, T.G., Atanassova, J.R., Beer, R., Belyanina, N.I., Blyakharchuk, T.A., Borisova, O.K., Bozilova, E., Bukreeva, G., Bunting, M.J., Clò, E., Colombaroli, D., Combourieu-Nebout, N., Desprat, S., Di Rita, F., Djamali, M., Edwards, K.J., Fall, P.L., Feurdean, A., Fletcher, W., Florenzano, A., Furlanetto, G., Gaceur, E., Galimov, A.T., Gałka, M., García-Moreiras, I., Giesecke, T., Grindean, R., Guido, M.A., Gvozdeva, I.G., Herzschuh, U., Hjelle, K.L., Ivanov, S., Jahns, S., Jankovska, V., Jiménez-Moreno, G., Karpińska-Kołaczek, M., Kitaba, I., Kołaczek, P., Lapteva, E.G., Latałowa, M., Lebreton, V., Leroy, S., Leydet, M., Lopatina, D.A., López-Sáez, J.A., Lotter, A.F., Magri, D., Marinova, E., Matthias, I., Mavridou, A., Mercuri, A.M., Mesa-Fernández, J.M., Mikishin, Y.A., Milecka, K., Montanari, C., Morales-Molino, C., Mrotzek, A., Muñoz Sobrino, C., Naidina, O.D., Nakagawa, T., Nielsen, A.B., Novenko, E.Y., Panajiotidis, S., Panova, N.K., Papadopoulou, M., Pardoe, H.S., Pędziszewska, A., Petrenko, T.I., Ramos-Román, M.J., Ravazzi, C., Rösch, M., Ryabogina, N., Sabariego Ruiz, S., Salonen, J.S., Sapelko, T.V., Schofield, J.E., Seppä, H., Shumilovskikh, L., Stivrins, N., Stojakowits, P., Svobodova Svitavska, H., Święta-Musznicka, J., Tantau, I., Tinner, W.,

Tobolski, K., Tonkov, S., Tsakiridou, M., Valsecchi, V., Zanina, O.G., Zimny, M., 2020. The Eurasian Modern Pollen Database (EMPD), version 2. Earth System Science Data 12, 2423–2445. https://doi.org/10.5194/essd-12-2423-2020

Dodson, J.R., 1978. A vegetation history from north-east Nelson, New Zealand. New Zealand Journal of Botany 16, 371–378. https://doi.org/10.1080/0028825X.1978.10425144

Dodson, J.R., Intoh, M., 1999. Prehistory and palaeoecology of Yap, federated states of Micronesia. Quaternary International 59, 17–26. https://doi.org/10.1016/S1040-6182(98)00068-8

Dugerdil, L., Joannin, S., Peyron, O., Jouffroy-Bapicot, I., Vannière, B., Boldgiv, B., Unkelbach, J., Behling, H., Ménot, G., 2021. Climate reconstructions based on GDGT and pollen surface datasets from Mongolia and Baikal area: calibrations and applicability to extremely cold–dry environments over the Late Holocene. Climate of the Past 17, 1199–1226. https://doi.org/10.5194/cp-17-1199-2021

Field, E., Tyler, J., Gadd, P.S., Moss, P., McGowan, H., Marx, S., 2018. Coherent patterns of environmental change at multiple organic spring sites in northwest Australia: Evidence of Indonesian-Australian summer monsoon variability over the last 14,500 years. Quaternary Science Reviews 196, 193–216. https://doi.org/10.1016/j.quascirev.2018.07.018

Fletcher, M.-S., Wood, S.W., Haberle, S.G., 2014. A fire-driven shift from forest to non-forest: evidence for alternative stable states? Ecology 95, 2504–2513. https://doi.org/10.1890/12-1766.1

Gaillard, M.-J., Birks, H.J.B., Emanuelsson, U., Berglund, B.E., 1992. Modern pollen/land-use relationships as an aid in the reconstruction of past land-uses and cultural landscapes: an example from south Sweden. Veget Hist Archaebot 1, 3–17. https://doi.org/10.1007/BF00190697

Haberle, S.G., 1993. Late Quaternary environmental history of the Tari Basin, Papua New Guinea (PhD Thesis). Australian National University, Canberra.

Haberle, S.G., 1996. Explanations for palaeoecological changes on the northern plains of Guadalcanal, Solomon Islands: the last 3200 years. The Holocene 6, 333–338. https://doi.org/10.1177/095968369600600307

Harrison, S.P., 2019. Modern pollen data for climate reconstructions, version 1 (SMPDS). https://doi.org/10.17864/1947.194

Harrison, S.P., Marinova, E., Cruz-Silva, E., 2021. EMBSeCBIO pollen database. https://doi.org/10.17864/1947.309

Harrison, S.P., Shen, Y., Sweeney, L., 2022a. Pollen data and charcoal data of the Iberian Peninsula (version 3). https://doi.org/10.17864/1947.000369

Harrison, S.P., Villegas-Diaz, R., Cruz-Silva, E., Gallagher, D., Kesner, D., Lincoln, P., Shen, Y., Sweeney, L., Colombaroli, D., Ali, A., Barhoumi, C., Bergeron, Y., Blyakharchuk, T., Bobek, P., Bradshaw, R., Clear, J.L., Czerwiński, S., Daniau, A-L., Dodson, J., Edwards, K.J.,

Edwards, M.E., Feurdean, A., Foster, D., Gajewski, K., Gałka, M., Garneau, M., Giesecke, T., Gil Romera, G., Girardin, M.P., Hoefer, D., Huang, K., Inoue, J., Jamrichová, E., Jasiunas, N., Jiang, W., Jiménez-Moreno, G., Karpińska-Kołaczek, M., Kołaczek, P., Kuosmanen, N., Lamentowicz, M., Lavoie, M., Li, F., Li, J., Lisitsyna, O., López-Sáez, J.A., Luelmo-Lautenschlaeger, R., Magnan, G., Magyari, E.K., Maksims, A., Marcisz, K., Marinova, E., Marlon, J., Mensing, S., Miroslaw-Grabowska, J., Oswald, W., Pérez-Díaz, S., Pérez-Obiol, R., Piilo, S., Poska, A., Qin, X., Remy, C.C., Richard, P.J.H., Salonen, S., Sasaki, N., Schneider, H., Shotyk, W., Stancikaite, M., Šteinberga, D., Stivrins, N., Takahara, H., Tan, Z., Trasune, L., Umbanhowar, C.E., Väliranta, M., Vassiljev, J., Xiao, X., Xu, Q., Xu, X., Zawisza, E., Zhao, Y., Zhou, Z., Paillard, J., 2022b. The Reading Palaeofire database: an expanded global resource to document changes in fire regimes from sedimentary charcoal records Earth System Science Earth System Science Data 14: 1109-1124 https://doi.org/10.5194/essd-14-1109-2022

Herbert, A.V., Harrison, S.P., 2016. Evaluation of a modern-analogue methodology for reconstructing Australian palaeoclimate from pollen. Review of Palaeobotany and Palynology 226, 65–77. https://doi.org/10.1016/j.revpalbo.2015.12.006

Herzschuh, U., Cao, X., Laepple, T., Dallmeyer, A., Telford, R.J., Ni, J., Chen, F., Kong, Z., Liu, G., Liu, K.-B., Liu, X., Stebich, M., Tang, L., Tian, F., Wang, Y., Wischnewski, J., Xu, Q., Yan, S., Yang, Z., Yu, G., Zhang, Y., Zhao, Y., Zheng, Z., 2019. Position and orientation of the westerly jet determined Holocene rainfall patterns in China. Nat Commun 10, 2376. https://doi.org/10.1038/s41467-019-09866-8

Hope, G., 2009. Environmental change and fire in the Owen Stanley Ranges, Papua New Guinea. Quaternary Science Reviews 28, 2261–2276. https://doi.org/10.1016/j.quascirev.2009.04.012

Hope, G., Gillieson, D., Head, J., 1988. A Comparison of Sedimentation and Environmental Change in New Guinea Shallow Lakes. Journal of Biogeography 15, 603–618. https://doi.org/10.2307/2845439

Hope, G., O'Dea, D., Southern, W., 1999. Holocene vegetation histories in the Western Pacific: alternative records of human impact, in: The Pacific from 5000 to 2000 BP: Colonisation and Transformations. Paris, pp. 387–404.

Jolly, D., Bonnefille, R., Burcq, S., Roux, M., 1996. Représentation pollinique de la forêt dense humide du Gabon, tests statistiques, in: Comptes rendus de l'Académie des sciences. Presented at the Sciences de la terre et des plančtes, Paris, pp. 63–70.

Julier, A.C.M., Jardine, P.E., Adu-Bredu, S., Coe, A.L., Duah-Gyamfi, A., Fraser, W.T., Lomax, B.H., Malhi, Y., Moore, S., Owusu-Afriyie, K., Gosling, W.D., 2018. The modern pollen–vegetation relationships of a tropical forest–savannah mosaic landscape, Ghana, West Africa. Palynology 42, 324–338. https://doi.org/10.1080/01916122.2017.1356392

Julier, A.C.M., Jardine, P.E., Adu-Bredu, S., Coe, A.L., Fraser, W.T., Lomax, B.H., Malhi, Y., Moore, S., Gosling, W.D., 2019. Variability in modern pollen rain from moist and wet tropical forest plots in Ghana, West Africa. Grana 58, 45–62. https://doi.org/10.1080/00173134.2018.1510027

Lebamba, J., Vincens, A., Jolly, D., Ngomanda, A., Schevin, P., Maley, J., Bentaleb, I., 2009. Modern pollen rain in savanna and forest ecosystems of Gabon and Cameroon, Central Atlantic Africa. Review of Palaeobotany and Palynology 153, 34–45. https://doi.org/10.1016/j.revpalbo.2008.06.004

Luly, J.G., 1993. Holocene palaeoenvironments near Lake Tyrrell, semi-arid northwestern Victoria, Australia. Journal of Biogeography 20, 587–598. https://doi.org/10.2307/2845516

Luly, J.G., Bowler, J.M., Head, M.J., 1986. A radiocarbon chronology from the playa Lake Tyrrell, Northwestern Victoria, Australia. Palaeogeography, Palaeoclimatology, Palaeoecology, Palaeoenvironments of Salt Lakes 54, 171–180. https://doi.org/10.1016/0031-0182(86)90123-9

Macphail, M.K., 1975. The history of the vegetation and climate in southern Tasmania since the late Pleistocene (ca. 13. 000-0 BP) (Doctoral dissertation). University of Tasmania.

Macphail, M.K., 1979. Vegetation and Climates in Southern Tasmania since the Last Glaciation. Quaternary Research 11, 306–341. https://doi.org/10.1016/0033-5894(79)90078-4

Macphail, M.K., 1980. Fossil and modern Beilschmiedia (Lauraceae) pollen in New Zealand. New Zealand Journal of Botany 18, 453–457. https://doi.org/10.1080/0028825X.1980.10425165

Macphail, M.K., McQueen, D.R., 1983. The value of New Zealand pollen and spores as indicators of Cenozoic vegetation and climates. Journal of the Biological Society 26, 37–59.

Macphail, M.K., Mildenhall, D.C., 1980. *Dactylanthus taylori*: in North-West Nelson, New Zealand? New Zealand Journal of Botany 18, 149–152. https://doi.org/10.1080/0028825X.1980.10427242

Mariani, M., Connor, S.E., Fletcher, M.-S., Theuerkauf, M., Kuneš, P., Jacobsen, G., Saunders, K.M., Zawadzki, A., 2017. How old is the Tasmanian cultural landscape? A test of landscape openness using quantitative land-cover reconstructions. Journal of Biogeography 44, 2410–2420. https://doi.org/10.1111/jbi.13040

McWethy, D.B., Whitlock, C., Wilmshurst, J.M., McGlone, M.S., Fromont, M., Li, X., Dieffenbacher-Krall, A., Hobbs, W.O., Fritz, S.C., Cook, E.R., 2010. Rapid landscape transformation in South Island, New Zealand, following initial Polynesian settlement. Proceedings of the National Academy of Sciences 107, 21343–21348. https://doi.org/10.1073/pnas.1011801107

McWethy, D.B., Wilmshurst, J.M., Whitlock, C., Wood, J.R., McGlone, M.S., 2014. A High-Resolution Chronology of Rapid Forest Transitions following Polynesian Arrival in New Zealand. PLOS ONE 9, e111328. https://doi.org/10.1371/journal.pone.0111328

Norton, D.A., McGlone, M.S., Wigley, T.M.L., 1986. Quantitative analyses of modern pollen-climate relationships in New Zealand indigenous forests. New Zealand Journal of Botany 24, 331–342. https://doi.org/10.1080/0028825X.1986.10412681

Phelps, L.N., Chevalier, M., Shanahan, T.M., Aleman, J.C., Courtney-Mustaphi, C., Kiahtipes, C.A., Broennimann, O., Marchant, R., Shekeine, J., Quick, L.J., Davis, B.A.S., Guisan, A., Manning, K., 2020. Asymmetric response of forest and grassy biomes to climate variability across the African Humid Period: influenced by anthropogenic disturbance? Ecography 43, 1118–1142. https://doi.org/10.1111/ecog.04990

Pickett, E.J., Harrison, S.P., Hope, G., Harle, K., Dodson, J.R., Kershaw, A.P., Prentice, I.C., Backhouse, J., Colhoun, E.A., D'Costa, D., Flenley, J., Grindrod, J., Haberle, S., Hassell, C., Kenyon, C., Macphail, M., Martin, H., Martin, A.H., McKenzie, M., Newsome, J.C., Penny, D., Powell, J., Raine, J.I., Southern, W., Stevenson, J., Sutra, J.P., Thomas, I., van der Kaars, S., Ward, J., 2004. Pollen-based reconstructions of biome distributions for Australia, Southeast Asia and the Pacific (SEAPAC region) at 0,6000 and 18,000 14C yr B.P. Journal of Biogeography 31, 1381–1444.

Prebble, M., Anderson, A.J., Augustinus, P., Emmitt, J., Fallon, S.J., Furey, L.L., Holdaway, S.J., Jorgensen, A., Ladefoged, T.N., Matthews, P.J., Meyer, J.-Y., Phillipps, R., Wallace, R., Porch, N., 2019. Early tropical crop production in marginal subtropical and temperate Polynesia. Proceedings of the National Academy of Sciences 116, 8824–8833. https://doi.org/10.1073/pnas.1821732116

Prebble, M., Kennedy, J., Southern, W., 2010. Holocene lowland vegetation change and human ecology in Manus Province, Papua New Guinea, in: Altered Ecologies. ANU Press, pp. 299–321.

Shulmeister, J., McLea, W.L., Singer, C., McKay, R.M., Hosie, C., 2003. Late Quaternary pollen records from the Lower Cobb Valley and adjacent areas, North-West Nelson, New Zealand. New Zealand Journal of Botany 41, 503–533. https://doi.org/10.1080/0028825X.2003.9512867

Takahara, H., Sugita, S., Harrison, S.P., Miyoshi, N., Morita, Y., Uchiyama, T., 2000. Pollen-based reconstructions of Japanese biomes at 0, 6000 and 18,000 14C yr bp. Journal of Biogeography 27, 665–683. https://doi.org/10.1046/j.1365-2699.2000.00432.x

Villegas-Diaz, R., Harrison, S.P., 2022. smpds: The SPECIAL Modern Pollen Data Set for Climate Reconstructions. Software. https://doi.org/10.5281/zenodo.6598832

Vincens, A., Lézine, A.-M., Buchet, G., Lewden, D., Le Thomas, A., 2007. African pollen database inventory of tree and shrub pollen types. Review of Palaeobotany and Palynology 145, 135–141. https://doi.org/10.1016/j.revpalbo.2006.09.004

Williams, J.W., Grimm, E.C., Blois, J.L., Charles, D.F., Davis, E.B., Goring, S.J., Graham, R.W., Smith, A.J., Anderson, M., Arroyo-Cabrales, J., Ashworth, A.C., Betancourt, J.L., Bills, B.W.,

Booth, R.K., Buckland, P.I., Curry, B.B., Giesecke, T., Jackson, S.T., Latorre, C., Nichols, J., Purdum, T., Roth, R.E., Stryker, M., Takahara, H., 2018. The Neotoma Paleoecology Database, a multiproxy, international, community-curated data resource. Quaternary Research 89, 156–177. https://doi.org/10.1017/qua.2017.105

<Line 156>. Fossil pollen data were obtained from the SPECIAL-EPD dataset (Harrison et al., 2024), a database of pollen records from Europe, the Middle East and western Eurasia. It builds on a pollen compilation covering the Middle East (EMBSecBIO database: Cordova et al., 2009; Harrison and Marinova, 2017; Marinova et al., 2017; Harrison et al., 2021), data available from public data repositories (NEOTOMA: https://www.neotomadb.org/; PANGAEA: https://www.pangaea.de/) and data provided by the original authors for the Iberian peninsula (Liu et al., 2023) and other regions. It includes 1,758 records from 1573 sites. The data have been extensively quality-controlled and mistakes have been corrected and documented. New BACON Bayesian age models, based on the recalibration of radiocarbon ages using INTCAL2020 (Reimer et al., 2020) calibration curve as appropriate, are provided for all the records using the 'rbacon' R package (Blaauw et al., 2021) in the 'AgeR' R package (Villegas-Diaz et al., 2021).

**Additional References**

Blaauw, M., Christen, J. A., Lopez, M. A. A., Vazquez, J. E., Gonzalez V., O. M., Belding, T., Theiler, J., Gough, B., & Karney, C. (2021). *rba- con: Age-depth modelling using Bayesian statistics* (2.5.6) [R]. https:// CRAN.R-project.org/package=rbacon

Cordova, C.E., Harrison, S.P., Mudie, P.J., Riehl, S, Leroy, S.A.G., Ortiz, N., 2009. Pollen, plant macrofossil and charcoal records for palaeovegetation reconstruction in the Mediterranean-Black Sea Corridor since the Last Glacial Maximum. Quaternary International 197: 12-26.

Harrison, S.P. and Marinova, E., 2017. EMBSeCBIO modern pollen biomisation. University of Reading Dataset. http://dx.doi.org/10.17864/1947.109.

Harrison, S.P., Marinova, E., & Cruz-Silva, E. (2021). *EMBSeCBIO pollen database* [Data set]. University of Reading. https://doi.org/10. 17864/1947.309

Liu, M., Shen, Y., González-Sampériz, P., Gil-Romera, G., ter Braak, C.J.F. Prentice, I.C., Harrison, S.P., 2023. Holocene climates of the Iberian Peninsula. *Climate of the Past* 19: 803-834, https://doi.org/10.5194/cp-19-803-2023.

Marinova, E., Harrison, S.P., Bragg, F., Connor, S., de Laet, V., Leroy, S., Mudie, P., Atanassova, J., Bozilova , E., Caner, H., Cordova, C., Djamali, M., Filipova-Marinova, M., Gerasimenko, N., Kouli, K., Kotthoff, U., Kvavadze, E., Lazarova, M., Novenko, E., Ramezani, E., Röpke, A., Shumilovskikh, L., Tantau, I., Tonkov, S., 2017. Pollen-derived biomes in the eastern Mediterranean-Black Sea-Caspian corridor. *Journal of Biogeography* 45: 484–499 DOI: 10.1111/jbi.13128

Reimer, P., Austin, W. E. N., Bard, E., Bayliss, A., Blackwell, P. G., Ramsey, C. B., Butzin, M., Cheng, H., Edwards, R. L., Friedrich, M., Grootes, P. M., Guilderson, T. P., Hajdas, I., Heaton, T. J., Hogg, A. G., Hughen, K. A., Kromer, B., Manning, S. W., Muscheler, R., ... Talamo, S. (2020). The IntCal20 northern hemisphere radiocarbon age calibra- tion curve (0-55 cal kBP). *Radiocarbon*, *62*(4), 725–757. https://doi. org/10.1017/RDC.2020.41

Villegas-Diaz, R., Cruz-Silva, E., & Harrison, S. P. (2021). ageR: Supervised age models [R]. *Zenodo*, https://doi.org/10.5281/zenodo.4636716

Specific comments

**R5:** L. 57: The LRA includes local reconstructions (LOVE) which has not been applied on the European scale. Only the REVEALS model was used.

We agree that this is potentially confusing. We will make the following amendment:

<Line 57>. The most recent quantitative pan-European pollen-based reconstructions of Holocene vegetation changes have been made using the REVEALS approach (Sugita, 2007b, a) or the Modern Analogue Technique (MAT) (Overpeck et al., 1985; Guiot, 1990; Jackson and Williams, 2004; Zanon et al. 2018).

**R6:** L. 58: You should cite Zanon et al. (2018) already here.

We agree that Zanon et al. (2018) should be cited here, given their extension of MAT to reconstruct tree cover. See response to R5.

**R7:** L. 63: The main focus was on reconstructing the proportion of open versus forest land cover.

We agree that we should clarify this, and will rewrite the text as follows:

<Line 63>. They produced maps showing the changing patterns of open vegetation versus forest, as well as specific plant functional types, at record-containing 1º grid cells for five time periods during the Holocene, based on 636 sites and 25 pollen taxa.

**R8:** L. 65: As an introductory overview this is almost too detailed while it is lacking studies to work as a good review of all that has come before: e.g. Pirzamanbein et al. (2014, Ecological Complexity), Roberts et al. (2018) Scientific Reports 8:716. Some of these appear in the discussion, but it would be good to mention them here already.

There are indeed a number of other studies applying different approaches to forest reconstruction, but we were concerned that introducing all of the techniques and approaches would make the introduction too long and so focused on the REVEALS and MAT approaches we used for quantitative comparison.

However, we agree that this paragraph could be improved by adding some additional information. We will amend the text as follows:

<L56>. Several different techniques that have been applied to reconstruct regional and sub-regional vegetation in Europe using pollen such as biomization/pseudobiomization (e.g. Fyfe et al., 2015; Binney et al., 2017) or the application of MAT using plant functional types (e.g. Davis et al., 2014). Other studies have made reconstructions combining different approaches (e.g. Roberts et al., 2018) and by combining pollen-based reconstructions with simulated potential vegetation (Pirzamanbein et al., 2014). However, the most recent quantitative pan-European pollen-based reconstructions of Holocene vegetation changes have been made using the Landscape Reconstruction Algorithm (LRA) REVEALS approach (Sugita, 2007b, a) or the Modern Analogue Technique (MAT) (Overpeck et al., 1985; Guiot, 1990; Jackson and Williams, 2004; Zanon et al., 2018). The REVEALS method calculates regional vegetation cover based on modelled relationships between pollen abundance, estimated differences in species level pollen productivity and pollen transport, and differences in site characteristics. Initially used at individual sites or small regions (e.g. Gaillard et al., 2010; Nielsen et al., 2012; Marquer et al., 2014), REVEALS was first applied at a pan-European scale by Trondman et al. (2015) and later extended with additional sites, taxa and an improved temporal resolution by Githumbi et al. (2022). The most recent analysis by Serge et al. (2023), is based on 1607 records for 500-year intervals before 700 cal. BP and for the subsequent intervals of 700-350 cal. BP, 350-100 cal. BP and 100 cal. BP- present. They tested the impact of including additional taxa (n=46) on the vegetation reconstructions, producing maps of landcover and species abundance at record-containing 1º grid cells. In contrast, the MAT approach reconstructs past vegetation based on identifying modern analogues of fossil pollen assemblages, on the assumption that samples found in the fossil record that share a similar composition to those found in present-day pollen assemblages will have similar vegetation. Zanon et al. (2018) applied MAT to 2,526 individual fossil pollen samples from Europe to generate interpolated maps at 250-year intervals at 5 arc-minute resolution through the Holocene.

**R9:** L. 80: Fall speeds are not the major issue as they can be estimated based on pollen size.

We agree that the including FS with RPP here implies they are equally challenging. We will amend the text as follows:

<L80>. For instance, landscape-level reconstructions are problematic if RPP information is not available for relatively common taxa.

**R10:** L. 86: Since you mention PFTs you may want to include Davis et al. (2015) here already not only in the discussion.

We agree that Davis should be cited here in relation to the use of PFTs with MAT, but that the 2003

(Quaternary Science Reviews; *22*(15-17)) paper would be most appropriate given that this paper explicitly introduced this approach in relation to MAT.

<L86>.Techniques designed to minimise the number of samples for which no analogues are found, such as grouping species into plant functional types (PFTs) (see Davis, 2003), introduce further uncertainties since the allocation of pollen taxa to PFTs is often ambiguous (Zanon et al., 2018).

**Additional reference**

Davis, B.A.S., Brewer, S. , Stevenson, A.C. , Guiot, J.: The temperature of Europe during the Holocene reconstructed from pollen data, Quat. Sci. Rev., 22, 1701-1716, https://doi.org/10.1016/S0277-3791(03)00173-2, 2003

**R11:** L. 115: The SPECIAL Modern Pollen Dataset (Villegas-Diaz and Harrison, 2022) compiles samples from other data sources including Neotoma and PANGAEA which also have a CC-BY-4.0 license, hence you need to cite or acknowledge the original data source not just the data compilation. Please see response to R4 and additional text describing the data set

**R12:** L. 116: SMPDS needs to be introduced. It is not clear from the above that this refers to the surface sample data.
As noted, the SMPDS contains pollen samples from the post-industrial era. Records include a variety of different entity types, including surface samples, sediment samples, core tops, pollen traps etc. As indicated in L.125, we only included records from lakes and bogs in our analysis. Please see response to R4 for the additional text describing the data set.

**R13:** L. 119: Particularly where core tops were used this assumption is daring.
In general, the samples described simply as modern in the data set were surface samples and we have followed the authors of the various compilations in assuming that this is true.

**R14:** L. 122: Give a brief motivation not just a reference.
We will amend the text to provide an explanation for this as follows:
<L.122>. Depauperate samples with Hill's N2 values (Hill, 1973) of < 2 were excluded, following Wei et al. (2021). Wei et al. (2021) found that low taxa diversity produced unreliable estimates of reconstructed variables, in this case temperature via tolerance weighted partial least squares estimation.

**R15:** L. 122-124: Here you are referring to surface samples, core tops or Holocene records?
The data from the SMPDS that we used included lake and bog records. We included all entity types to build the modern tree cover model, except moss polsters and pollen traps which reflect very

localised pattern of pollen rain. We thank the reviewer for raising this point, as this should have been mentioned in the method and will revise the text as follows

<L.125>. Only samples from lakes and bogs were included, to ensure appropriate pollen source areas could be calculated, and samples gathered via moss polsters or pollen traps were excluded as these generally reflect only the very local pollen rain.

**R16:** L. 126: So you include small bogs but exclude large bogs? I cannot find this constraint discussed in Githumbi et al. (2022).

Githumbi et al. (2022) suggest that: "…*REVEALS estimates of plant cover using pollen assemblages from large bogs should only be interpreted with great caution (Mazier et al., 2012; see also Sect. 4, "Discussion").*" (Githumbi et al., 2022). They included estimates from large bogs in their analysis but flagged these as "lower quality" estimates. Given this caution, especially regarding the issue of surface level vegetation and our inclusion of Cyperaceae, Polypodiales and Ericaceae, we decided to exclude large bog sites from our analysis. We will amend the text as follows:

<L126>. However, bog records with a radius $\geq$ 400m were excluded from the analysis. Githumbi et al (2022) indicated that caution was necessary in interpreting REVEALS vegetation reconstruction estimates based on large bogs and, given that we included taxa that grow on bog surfaces in our analysis (see below), were excluded large bogs to reduce the potential for these to bias the regional vegetation reconstructions.

**R17:** L. 135: It would be useful to mention what is included in shrub pollen: Are you including dwarf shrubs like Calluna or rather taller perennial woody plants like Corylus and Juniperus?

The list of species included within each grouping is included within the Table in Supplementary information: S1. Calluna is included within the shrub group, whereas Colylus and Juniperus (as part of amalgamated group Cupressaceae) are included as trees.

**R18:** L. 139: How did you deal with situations where alien tree plantations make up most forest cover: e.g. Eucalyptus. Also plantations of Pseudotsuga (0.83 million ha in Europe) may be a potential problem.

Observed tree cover is based on the Copernicus Global Land Service maps (line 105). These maps do not distinguish between alien/natural species tree cover or plantation tree cover, and so we are unable to distinguish alien tree plantations. We will clarify this in the text as follows:

<Line 107>........ permanent water, snow, and crops (Fig. 2A). However, the Copernicus maps do not distinguish between natural forests and plantations and so the tree cover target may include planted species.

**R19:** L. 140: Large proportions of Cyperaceae and Polypodiales are limited to bogs, excluding them would reduce the biases from including bog samples.

As we mention, the inclusion of these species was to help prevent open environments being dominated by pollen from long distance transport. Site type (bog or lake) is included as a regressor, which is meant to reflect differences between bog and lake records.

**R20:** L. 145: It would be good if you mentioned here the range of resulting source areas considered.

We agree that it would be useful to indicate here the range of source areas considered.

<L.151>. Source area radii varied in size from 5,026 km to 418,894 km for the largest lake, with a median of 28,316 km.

**R21:** L. 153: What do you mean by "non-natural vegetation" here?

We agree that the terminology non-natural vegetation is somewhat ambiguous, so we will change this here, and at line 106 where the term was first used and defined, to other land-cover classes:

<Line 106>. A composite map of modern tree cover for the region 12°W to 45°E and 34-73°N was generated by averaging annual percentage tree cover data from Copernicus annual land cover maps from 2015 to 2019 (Buchhorn et al., 2020a, e, d, c, b), after removing cells dominated (> 50%) by other land-cover classes, including bare ground, built up areas, moss or lichen, permanent water, snow, and crops (Fig. 2A).

<Line 153>. There were 263 records where more than half of the contributing grid cells were masked as land-cover classes other than vegetation; these were excluded from the model construction.

**R22:** L. 154: How many from bogs?

There were 133 bog records used for the development of the modern tree cover model. We will include this information:

<L.154>. A total of 852 pollen records were included in the final model training dataset, of which 133 were bog records.

**R23:** L. 156: The same problem of attribution applies to the SPECIAL-EPD. Please cite and acknowledge the EPD. See https://www.neotomadb.org/data/data-use-and-embargo-policy

Please see response to R4 and the additional text.

**R24:** L. 167ff: I like the idea, but am skeptical about the predictors used. Rather than using % needleleaf, it would have been better to classify the pollen types according to high mid and low pollen producing plants. Needleleaf trees include the high pollen producing Pines and low producing Larix

(or Pseudotsuga). I am not sure elevation is a good predictor when thinking about the past as vegetation belts moved up and down the mountains during the Holocene. I would perhaps rather limit the inclusion of modern and fossil sites to below 500 m. I don't understand the need of including the Shannon Index, particularly I don't understand the provided motivation.

Please see response to R1.

**R25:** L. 233ff: We know that % tree pollen is a strong predictor of forest cover without any transformation so it would be useful to compare the model performance to the performance of a simple regression model of % tree and shrub pollen (depending on what is in the shrubs) versus forest cover.

Please see our response to R1.

**R26:** L. 233: The negative correlation between %needleleaf and tree cover is interesting and unexpected. Could that be due to frequent Pine pollen in generally open areas. Picea pollen should however correlate with high tree cover.

We agree that this could be an explanation for this relationship. The influence of open landscapes on the model at lower levels of needleleaf% is also potentially an explanation for the modelled relationships between SI tree cover and tree cover. For more open landscapes, tree species diversity may be limited. As these landscapes become more mixed, tree species diversity may increase. But at higher levels of tree species diversity, as evidenced by the negative quadratic term, the importance of this variable to increased tree cover values decreases and potentially could become negative for heavily wooded areas. We will add the following text:

<L.235>. This negative relationship may be a reflection of longer distance pollen transport of needleleaf species (e.g. *Pinus*) to open environments. As tree cover increases, this may imply an increased diversity of species, including broadleaf species. The positive quadratic term indicates that this relationship becomes positive at higher levels of tree cover, potentially reflecting higher tree cover in boreal needleleaf forests. Increased SI is positively related to tree cover, with the effect decreasing with elevation. However, the negative correlation for the quadratic term for the SI suggests that the relationship has less of an effect on tree cover as SI increases. Again, this relationship may be explained in the context of open environments, where tree species diversity may be limited to species with longer distance pollen transport. Tree species diversity may then increase with tree cover, with the negative quadratic term implying that the highest levels of tree cover are represented by relatively uniform species types.

**R27:** L. 258: The overestimation of tree cover in northern Scandinavia is interesting and expected as pollen productivity is lower. This is also the case for higher elevations, which is why elevation is a good covariable for the present, but this relationship may not hold true in the past where temperature

changes resulted in changing pollen productivities in the mountains.

Please see our response to R1.

**R28:** L. 330: The difference in tree cover between the reconstructions for the last 1000 years and the early Holocene is intriguing. As Zanon et al (2018) and Serge et al (2023) use completely different methodologies, but show the same trend, my initial response would be to trust them more, even if the absolute modern cover is off for both. Here it would be interesting to explore the reasons for the deviations of the current study. Could one reason be the separation of shrub pollen from tree pollen?

Within the Discussion section (L.380), we ascribe this difference as being potentially due a technical point regarding the modern map of observed tree cover. As we indicate, our model is trained on this map, which specifically excludes other land-cover classes. In contrast to the other reconstructions, we are effectively modelling tree cover without human influence, which is why our reconstructions deviate more for the later Holocene than at other periods.

**R29:** L. 341: Please see the recent manuscript by Schild et al. (https://essd.copernicus.org/preprints/essd-2023-486/#discussion ) who argue that the REVEALS method underestimates the forest cover. If that would be true then your new method would perform worse as it scores below the REVEALS estimates. If you argue that forest cover was generally lower then it would be useful to find supporting evidence and make that a point of discussion.

Thank you for highlighting this recent manuscript. However, this preprint does not appear to suggest that REVEALS underestimates forest cover, when validating modern reconstructions with satellite data. They argue that REVEALS successfully reduces the overestimation of tree cover based purely on pollen data in the modern day, with that adjustment further improved by optimising RPP values, particularly for North America (see Fig. 10, p14). Optimization of RPP values actually reduces tree cover estimates through time globally (see Figure 8, p12), which implies that the REVEALS estimates from Serge et al. (2023) would be on the higher, rather than lower side.

**R30:** L. 421: The main deforestation of Northwestern Europe took place during the Bronze Age and Medieval period leading to an all-time low around 1800 (see e.g. Bradshaw and Sykes 2014 Ecosystem Dynamics, Wiley).

This statement implies that we know the cause of forest loss in Europe and that it is primarily due to human destruction of the natural vegetation. Part of our reason for making forest cover reconstructions is to be able to test this assertion quantitatively. The Bronze Age is a somewhat loosely defined epoch somewhere between 5300 and 2700 BP. Since all three reconstructions show a decline in forest cover after 6ka, it could be argued that this is consistent with the idea that the main deforestation of Europe took place during the Bronze Age and is therefore potentially explained by human activities. But it is clear from our regional reconstructions and also from the REVEALs based

analysis by Roberts et al. (2017), that the timing of forest loss varied across Europe. This could, of course, be due to differences in the timing of landscape appropriation by people. But the latter part of the Holocene is also a time when climate was changing, and the timing of these changes is also non-synchronous across the continent. So, it would be equally plausible to argue that climate changes are responsible for (or have contributed to) changes in forest cover. This is the point we were trying to make, but we will take this opportunity to spell it out more clearly as follows:

<L411>. The late Holocene decline in tree cover is consistent with the orbitally-driven cooling. However, the more rapid decline in tree cover during the last millennium shown in the Boreal and Continental regions, and shown more dramatically in the Zanon et al. (2018) and Serge et al. (2023) reconstructions, is more difficult to explain as a function of climate changes - transient model simulations of the response to changes in orbital and greenhouse gas forcing (e.g. Liu et al., 2009; Zhang et al., 2016; Braconnot et al., 2019; Dallmeyer et al., 2020) generally indicate muted changes in either summer or winter temperatures during the most recent millennia. Human influence on the landscape has been identified in some regions of Europe from 6,000 cal. BP onwards (e.g. Roberts et al., 2018; Zapolska et al., 2023). Although this may have contributed to the climate-driven decline in forest cover, rapid population growth occurred only during the past 2000 years (Klein Goldewijk et al., 2010, 2017). The recent decline in tree cover may therefore reflect this rapid growth and the consequent increasing human influence on the landscape in some regions (see e.g. Marquer et al., 2017; Roberts et al., 2019). Much of the debate about the relative importance of climate and human activities on the environment during the Holocene has been based on correlations, often at a local scale. More formal modelling of these relationships, using quantitative information on climate and population size, is required to assign the impact of each on tree cover more confidently.

---

## Author Comment (AC2)

**Response to Reviewer 2**

We thank the reviewer for their positive review of the paper and for their specific comments and suggestions. Our response to these points are given below in blue and changes to the text are given in red.

General

Sweeney et al. present a new, simplified approach for deriving tree cover estimates from fossil pollen compositional data. The authors apply this approach to selected pollen records in Europe to reconstruct tree cover changes throughout the Holocene. They compare their results to similar previous studies that used different approaches, and discuss the new approach, recommending it for application in future studies in other regions or on a global scale. The manuscript is well written, and, especially important for a study focused on presenting such a new modeling approach, outlines the methods very clearly and understandably. With its interesting approach to understanding tree cover changes through time and a discussion-based nature, it fits within the scope of Biogeosciences. I would suggest the acceptance of this study after minor revisions, related mostly to the structure of the manuscript.

Specific comments

**R1:** "Forest cover" and "tree cover" from my point of view may not always refer to the same aspect. In the title, forest cover is mentioned, but throughout the manuscript the focus lies rather on tree cover. I suggest sticking to only one of these terms and including a brief definition in the introduction or methods section.

We agree that the two terms are not synonymous, and we will change the text to tree cover throughout (including in the title). We will explain that tree cover could reflect forest expansion but also growth of woodlands and more isolated trees in vegetation mosaics or in urban settings in the first paragraph, as follows:

<Line 35>. Tree cover in Europe has been expanding in recent decades (FAO, 2020; Turubanova et al., 2023), with potential implications for land-atmosphere energy exchanges, water and carbon cycles, and ultimately local and global climates (Bonan, 2008; Alkama and Cescatti, 2016). Changes in tree cover may reflect forest or woodland expansion, or the presence of trees in vegetation mosaics and urban settings.  Tree cover both affects and is affected by the environment (Moyes et al., 2015; Abis and Brovkin, 2017) and this two-way relationship leads to complex interactions between both.

**R2:** While reading the results section, I was sometimes wondering if I had unknowingly entered the discussion (e.g., in L345 onwards) – I think in this case it may be acceptable to have some level of discussion within the results section in this style of manuscript, since it relies heavily on comparisons

to previous, similar studies. However, maybe the writing style could be adapted to better differentiate the two sections.

In the results section, we were trying to explain what the differences are and where these might come from. However, these are also issues that are covered in the Discussion and should, perhaps, be more appropriately dealt with there. We will remove the more discussion-oriented parts of the text from the Results section and simply focus on pointing out the differences. Specifically, we will remove the following speculation from the Discussion:

<Line 349>. This probably reflects differences in data coverage between Serge et al. (2023) and the other two reconstructions.

**R3:** The supplement to this manuscript provides very relevant information, especially if a reproduction of the approach is wished, and seems well-organized. However, I would suggest the inclusion of some of the spatially plotted model results also in the main manuscript. Some patterns across Europe are described in the results and discussion sections, and having some of the time-slice plots close by would be very helpful (e.g., one each from the early, mid, and late Holocene, referring to the full range of plots in the supplement, or what may fit best to your discussion).

We will add a figure showing the time slice plots for 11,000, 9,000, 6,000 and 2,000 yr BP in the main text and reference these in line 298, as follows:

[Figure]

**Figure 6: Gridded maps of average reconstructed tree cover for selected periods, for 50km² grid cells. Bin ages are 200-years in width, with ages referring to mid-point of each bin.**

Technical comments

- L16: Not sure if "Earth System" needs to be written in capitals

No, this does not need to be capitalised here. We will correct this.

- L29: Re-phrase this section as it is a bit repetitive ("[…] our approach […] provides a better approach […]")

We will rewrite this section as:

<Line 29>. The reconstructed patterns of change in tree cover are similar to those shown by previous reconstructions, but our approach is more robust and does not require RPPs and therefore provides a useful way to reconstruct tree cover in regions where data limitations preclude the use of alternative methods.

- L45: This may be purely subjective, but I'd suggest to use long-dashes when writing ranges

We will correct this throughout.

- L50: When saying that something is commonly done, I expected to find some references at the end of the statement

This is really a ubiquitous approach in non-statistical papers, and a similar unreferenced statement is made by Zanon et al. However, we can add some examples here. Specifically, we will cite:

Adam, M., Weitzel, N., and Rehfeld, K.: Identifying Global-Scale Patterns of Vegetation Change During the Last Deglaciation From Paleoclimate Networks, Paleoceanogr Paleoclimatol, 36, https://doi.org/10.1029/2021PA004265, 2021.

Hicks, S.: The use of annual arboreal pollen deposition values for delimiting tree-lines in the landscape and exploring models of pollen dispersal, Rev Palaeobot Palynol, 117, 1–29, https://doi.org/10.1016/S0034-6667(01)00074-4, 2001.

Kaplan, J. O., Pfeiffer, M., Kolen, J. C. A., and Davis, B. A. S.: Large Scale Anthropogenic Reduction of Forest Cover in Last Glacial Maximum Europe, PLoS One, 11, e0166726, https://doi.org/10.1371/journal.pone.0166726, 2016.

- L70: Re-phrase to make clear that n=46 is the total amount of taxa and not the additional amount of taxa

We will rephrase this as:

<Line 70>.... is based on 1607 records and tested the impact of including 15 additional taxa (total n=46) on the vegetation reconstructions.

- L91: Suggest to standardize across the manuscript the way such lists are written using comma and only a single final "and" (e.g., "[…] data on tree cover, harmonized age models, and improved information […]"; similar cases e.g. in L181, L223)

We will check and ensure that the lists are presented in a standardised way

- L126: Standardize the use of spaces before writing units across the manuscript (e.g., different way can be found in L219 – personally, I prefer the use of spaces)

We will check and ensure that the units are presented in a standardised way

- Figure 2A: Suggest to add either "%" next to the legend, or otherwise state in the caption that it depicts relative data

We will redraw the figure to make sure that it is clear that the legend refers to % cover, as follows:

[Figure]

**Figure 2: A - Observed tree cover based on compositing annual tree cover maps from the Copernicus land cover data sets (2015-2019) and screening out cells where the dominant land cover was not natural; B - Modern pollen records used for model fitting; C - Fossil pollen sites used for tree cover reconstructions; D - Classification of the fossil pollen sites into climatic sub-regions**

- L198: I was not sure who "they indicate" was – is that information from the cited study, or personal communication with the authors?

This statement is in the Serge et al paper. We will clarify this as follows:

<Line 198>. We make comparisons to the Serge et al. (2023) reconstructions based on the 31 taxa originally used by Githumbi et al. (2022) since Serge et al. (2023) show that this produces better results than using the expanded data set of 46 taxa.

- L266: Dot missing in reference

We will correct this.

- L288: Comma or other separator missing in the R package citation

We will correct this.

- L324: Suggest to change "somewhat" to "slightly" and maybe separate this into two sentences

Thank you for this suggestion. We will rewrite as follows:

<Line 324>. Our reconstructed maximum cover in slightly lower (ca. 5-10%) than shown by the other reconstructions. However, the mid-Holocene timing of this maximum is broadly consistent across all of the reconstructions (although Zanon et al. (2018) show a double peak in tree cover, with an earlier peak at ca. 9,000 cal. BP) within the limitations of the age models and binning intervals used (see Supplementary Information: S10 and S11).

- L437: I think there's an "as" missing in "such how"

Yes, we will add this

---

## Author Comment (AC4)

We thank Marie-Jose for identifying where she is happy with our response to her original comments. Here we provide response to her additional comments (in blue), changes to the text are given in red.

1) p. 4. There are RPPs for Africa (see e.g. Gaillard et al., 2017) and southern America (see e.g. xxxx)

Other than the PAGES magazine article, we can't find a Gaillard et al. (2017) paper. However, there are some pilot studies or studies for specific ecosystems in the 2021 Quaternary Vegetation Dynamics book, which we can refer to. The problem still remains that even these studies contain RPPs for a very limited number of taxa. There are only 13 taxa from the Cameroon study and the other studies cited in Gaillard et al. (2021) for Africa and South America appear to also cover a limited number of taxa. However, we will further modify the text to clarify where there are additional RPP data, as follows:

Landscape-level reconstructions are problematic if RPP information is not available for relatively common taxa (Harrison et al., 2020). RPP values have been estimated for common taxa in Europe and China, and there are a limited number of studies from North America (see e.g. Wieczorek and Herzschuh, 2020). Some studies have been conducted for some ecosystems in South America and Africa, but these only provide RPPs for a very limited number of taxa (e.g. Duffin & Bunting, 2008; Whitney et al, 2018; Gaillard et al., 2021; Hill et al., 2021; Tabares et al., 2021; Piraquive Bermúdez et al., 2022) and even this level of information is not readily available for other regions.

*We will include the additional references*:

Duffin, K. I. and Bunting, M. J.: Relative pollen productivity and fall speed estimates for southern African savanna taxa, Veg Hist Archaeobot, 17, 507–525, https://doi.org/10.1007/s00334-007-0101-2, 2008.

Gaillard, M.-J., Githumbi, E., Achoundong, G., Lézine, A.-M., Hély, C., Lebamba, J., Marquer, L., Mazier, F., Li, F., and Sugita, S.: The challenge of pollen-based quantitative reconstruction of Holocene plant cover in tropical regions: A pilot study in Cameroon, in: Quaternary Vegetation Dynamics: the African Pollen Database, edited by: Runge, J., Gosling, W., Lézine, A.-M., and Scott, L., CRC Press, London, 259–279, 2021.

Hill, T. R., Duthie, T. J., and Bunting, J.: Pollen productivity estimates from KwaZulu-Natal Drakensberg, South Africa, in: Quaternary Vegetation Dynamics: the African Pollen Database, edited by: Runge, J., Gosling, W., Lézine, A.-M., and Scott, L., CRC Press, London, 259–274, 2021.

Piraquive Bermúdez, D., Theuerkauf, M., and Giesecke, T.: Towards quantifying changes in forest cover in the Araucaria forest-grassland mosaic in southern Brazil, Veg Hist Archaeobot, 31, 107–122, https://doi.org/10.1007/s00334-021-00841-2, 2022.

Tabares, X., Ratzmann, G., Kruse, S., Theuerkauf, M., Mapani, B., and Herzschuh, U.: Relative pollen productivity estimates of savanna taxa from southern Africa and their application to reconstruct shrub encroachment during the last century, Holocene, 31, 1100–1111, https://doi.org/10.1177/09596836211003193, 2021.

Whitney, B. S., Smallman, T. L., Mitchard, E. T., Carson, J. F., Mayle, F. E., and Bunting, M. J.: Constraining pollen-based estimates of forest cover in the Amazon: A simulation approach, Holocene, 29, 262–270, https://doi.org/10.1177/0959683618810394, 2019.

2) P5. My question on crops still remains. Do the remote-sensed data provide details on the crops. Do all the crop areas you deleted correspond to areas that do not produce much pollen?

The Copernicus land-cover data set that we used does indeed include percentage of 100m pixel covered by all types of and, as we state in the text, we have removed these from our data set. The data set also includes the percentage of forest/tree cover for each pixel. It also identifies forest type (broad biome classification) but we do not use this in our analysis.

3) P6. I am sorry for this misunderstanding. I was thinking about REVEALS applications when I was writing this. My apologies. Your choice of using the Prentice's source area formula for 70% pollen as an estimate of the "appropriate area for the calculation of mean tree cover" is fine. I would however write that it is an estimate, because knowing wether this area is appropriate for all types of sites, large or small, lakes or bog, would need to be tested.

We will clarify that this is an estimate, as follows:

The source area for each record, and hence the appropriate area for the calculation of mean tree cover, was calculated using Prentice's (1985) source area formula for 70% of pollen, and lake or bog area from the SMPDS, noting that the original formula makes no distinction for different site types.

4) P7. what do you mean by "plausible"? How do you know that this is the "appropriate area", i.e. the vegetation area represented by the pollen assemblages in all your sites, large or small lakes, or small bogs?

By plausible, we mean that the final model has reasonably good (though not perfect) metrics (see Section 3.1). As we point out in the text, basin size is not available for all of the sites used for the reconstructions (both modern and fossil) which precludes a comprehensive evaluation of the

impact of basin size on source area. The problem is compounded for bog sites, where even when there are estimates of size they are somewhat approximate.

5) P7 a pragmatic attempt …. You assume the pollen assemblages reflect the average vegetation composition within the Prentice's source area for 70% of pollen, for any type of site. It's an estimate or approximation of the area.

And P7 but taking a source area that is too large ….. Well, I do not see the problem.

We are indeed making the assumption that the pollen assemblages reflect the average vegetation composition using the Prentice' source area formula, and we agree that this is an estimate or approximation of the actual area. We also agree that using the median FS is an approximation. We have made these assumptions because of the limitations of the available data, both on basin size and on fall speeds. Hence we regard this as a "pragmatic" approach, but nevertheless the model does seem to give results that are consistent with the modern observations of tree cover.

6) P13 our reconstruction is of tree cover ….. I am not convinced that you reconstruct tree cover. Does your vegetation data provide the actual % cover of trees within all land-cover types you are using? Note that REVEALS estimate tree cover for the trees for which RPPs are available and nothing else, and it includes trees in woodlands and trees in mixed wooded/open land-cover types.

Also … see my comment above; I am not still not convinced that what you reconstruct is tree cover; I may misunderstand something re your vegetation data. In that case, I apologize; haven't the time to dig further into your vegetation data and how you handled it to get "tree cover".

We have clearly not explained the Copernicus data set well enough. The data set provides information on land cover classes. One of these land-cover classes is forest/tree cover. It provides information on forest/tree cover as a percentage for each 100m pixel. By using the percentage forest/tree cover at this high resolution, we are indeed reconstructing tree cover. We will expand the description of this data set to clarify this, as follows:

We used the Copernicus Dynamic Land Cover Dataset to source information on tree cover. This data set provides percentage land cover estimates for individual land-cover classes at a spatial resolution of 100m. We used the land-cover class designated as forest/tree cover, which at this spatial resolution can be regarded as an estimate of actual tree cover. A composite map of modern tree cover for the region 12°W to 45°E and 34-73°N was generated by averaging annual percentage forest/tree cover data from Copernicus annual land cover maps from 2015 to 2019 (Buchhorn et al., 2020a, e, d, c, b), after removing cells dominated (> 50%) by other land-cover classes, including bare ground, built up areas, moss or lichen, permanent water, snow, and crops (Fig. 2A). However, the Copernicus maps do

not distinguish between natural forests and plantations and so the tree cover target may include planted species. This modern tree cover map has a resolution of 100m.

P15 to model them quatitatively ….. sorry, but these will also be only as good as the models themselves...... I can understand that you choose to be "cautious about attributing any of the reconstructed changes to human activities etc...." but I have difficulties to find it relevant to ignore all research on human impact on the Holocene vegetation before 1000 BP. Arguing that all this knowledge is based primarily on vegetation changes correlated to archaeological records and that correlation is not causation is a bit arrogant. I really don't see how you could replicate these Holocene pollen records from ca. 7-6 BP until today using climate, CO2 and records of natural fire (climate-induced fire). I really can't  "buy" these arguments for erasing a enormous amount of knowledge in the fields of archaeology, vegetation history and palaeoecology in general. I am very sorry.

We agree that the reconstructions are only as good as the models themselves, and this is one reason that we have been at pains to provide metrics for model fit. We are not claiming that there has been no human impact on European vegetation cover during the Holocene and we are not dismissing the archaeological and palaeoecological literature on this out of hand, but we remain cautious about the spatial scale of this influence and its relative importance compared to other factors. We have also been careful to point out in the discussion that climate alone cannot explain all of the observed changes. We have not investigated or discussed the potential impact of other factors, such as disturbance, but agree this should not be neglected. (There may also be a role for $CO_2$, given that the change between the mid-Holocene and pre-industrial is about 11 ppm, but this really has not been investigated and in any case would tend to increase tree cover rather than the opposite.) We will treat the discussion of the climate influence on the tree cover changes, including what these changes do not appear to explain, as a separate paragraph. We will create a new paragraph to discuss the potential influence of other factors, including human impacts, as follows:

There are several other factors that could have influenced tree cover during the Holocene. Human influence on the landscape has been identified in many regions of Europe from 6,000 cal. BP onwards (e.g. Roberts et al., 2018; Zapolska et al., 2023). Although this may have contributed to observed decline in tree cover from the mid-Holocene onwards, the most rapid population growth occurred only during the past 2000 years (Klein Goldewijk et al., 2010, 2017). The recent decline in tree cover may therefore reflect this rapid growth and the consequent increasing human influence on the landscape in some regions (see e.g. Marquer et al., 2017; Roberts et al., 2019). Climate-driven changes in disturbance (wildfires, windthrow) may also contribute to the inferred changes in tree

cover. Changes in the frequency or intensity of storms has, for example, been shown in maritime Europe (e.g. Pouzet et al., 2018; Sjöström et al., 2024) during the late Holocene; storms are a major cause of widespread forest damage in Europe today (Senf and Siedl, 2020) and could have been important during the Holocene. Changing wildfire regimes could also have been an important influence on tree cover (Marlon et al., 2013; Kuosmanen et al., 2014). Much of the debate about the relative importance of climate and human activities on the environment during the Holocene has been based on local-scale correlations; other contributing factors have been largely ignored. More formal modelling of these relationships, using quantitative information on climate, population size, and disturbance is required to assign the impact of each on tree cover more confidently.

We will add these references;

Kuosmanen, N., Fang, K., Bradshaw, R. H., Clear, J. L., and Seppä, H.: Role of forest fires in Holocene stand-scale dynamics in the unmanaged taiga forest of northwestern Russia, Holocene, 24, 1503–1514, https://doi.org/10.1177/0959683614544065, 2014.

Marlon, J. R., Bartlein, P. J., Daniau, A.-L., Harrison, S. P., Maezumi, S. Y., Power, M. J., Tinner, W., and Vannière, B.: Global biomass burning: a synthesis and review of Holocene paleofire records and their controls, Quat Sci Rev, 65, 5–25, https://doi.org/10.1016/j.quascirev.2012.11.029, 2013.

Pouzet, P., Maanan, M., Piotrowska, N., Baltzer, A., Stéphan, P., and Robin, M.: Chronology of Holocene storm events along the European Atlantic coast, Progress in Physical Geography: Earth and Environment, 42, 431–450, https://doi.org/10.1177/0309133318776500, 2018.

Senf, C. and Seidl, R.: Mapping the forest disturbance regimes of Europe, Nat Sustain, 4, 63–70, https://doi.org/10.1038/s41893-020-00609-y, 2020.

Sjöström, J. K., Gyllencreutz, R., Martínez Cortizas, A., Nylund, A., Piilo, S. R., Schenk, F., McKeown, M., Ryberg, E. E., and Kylander, M. E.: Holocene storminess dynamics in northwestern Ireland: Shifts in storm duration and frequency between the mid- and late Holocene, Quat Sci Rev, 337, 108803, https://doi.org/10.1016/j.quascirev.2024.108803, 2024.

---

## Author Response (AR2)

**Point-by-point response to referee comments (3)**

We would like to thank the reviewer for their continued help in improving this manuscript.

Our responses to individual comments are highlighted in blue, with any proposed changes highlighted in red. Note that red page numbers refer to the location in the updated marked-up manuscript.

**R1**. I am pleased to see that the authors took many of my comments into account for the revision of the manuscript. Unfortunately no changes were made to the regression model. I accept the argument made that classifying all pollen taxa into under and overrepresented taxa may be difficult, but what would be possible is using the pollen dispersal syndrome: wind, insect, both. Also the fall speed could be obtained for all pollen taxa and this combination could really yield something new.

We tested the inclusion of dispersal syndrome instead of %needleleaf within the regression model. Categorisation of tree species as wind or biotically pollinated is a challenge, particularly for pollen only identified at the family level. We primarily used the categorisation from Tong et al. (2023), with some amendments based on expected primary dispersal methods. Models with %wind pollination and %wind and dual pollination produced a worse fit than the %needleleaf model. We also tested the inclusion of %*Pinus* rather than the %needleleaf proportion within the model. Here there was less of a difference with the %needleleaf, with the model including %*Pinus* only very marginally worse. We have included this additional analysis within the manuscript and supplement.

L228. As an additional approach to address differences in pollen production and transport, we tested the implications of including broad pollen dispersal syndromes in the model in place of %needleleaf. We calculated the percentage of the tree pollen based on wind pollination as opposed to biotic (insect, bird) or dual (wind or biotic) pollination methods, following the categorisations by Tong et al. (2023), and Kling and Ackerly (2021) (see Supplementary Information: S2). *Pinus* is the most ubiquitous pollen type recorded in the modelled data set, and is widely recognised as a potential contaminant in more open vegetation due to long-distance transport. We therefore also tested the impact of including %*Pinus* as a predictor in the regression model.

L323. Replacing %needleleaf with either %wind pollination tree species or %*Pinus* as explanatory variables related to pollen transport resulted in a poorer model fit (Supplementary Information: S3).

**S3: Coefficients and model fit including %wind pollination or %Pinus**

Including the %wind pollinated within the regression model instead of %needleleaf, also with a second-degree polynomial, led to a reduction in the model pseudo (Cox-Snell) R2 to 0.56 (compared to 0.60). LOOCV MAE increased to 0.12 (from 0.11), RMSE increased to 0.15 from 0.14 and the squared correlation (R2) of the predictions to the observations was reduced to 0.57 (from 0.63). The model coefficients are shown in Supplementary Table 3; although the coefficient values change compared to the model with %needleleaf, the directions of the coefficients remain the same. The slight reduction in model fit reflects the fact that %needleleaf also partially accounts for differences in pollen productivity, as well as differences in pollen transport. As well as selecting the tree species classified as wind pollinated only, we also investigated the model fit when including both those species classified as wind pollinated and those species classified as both wind and biotically pollinated. In this case the model fit was similarly worse than using %needleleaf, with the pseudo (Cox-Snell) R2 0.55, and LOOCV MAE 0.12, RMSE 0.15 and the squared correlation (R2) of the predictions to the observations 0.57.

*Supplementary Table 1: Modern tree cover model coefficients, including %wind pollinated rather than %needleleaf*

| Coefficients (mean model with logit link) | Estimate | Standard Error | P Value |
|---|---|---|---|
| (Intercept) | -1.045 | 1.899 | 0.582 |
| Tree pollen % | 2.592 | 0.233 | 8.66e-29 *** |
| Shrub pollen % | -3.834 | 0.642 | 2.38e-09 *** |
| Wind pollination of AP% | -11.090 | 4.486 | 0.013 * |
| Wind pollination of AP%^2 | 7.264 | 2.701 | 0.007 ** |
| AP Shannon index | 4.413 | 0.481 | 4.61e-20 *** |
| AP Shannon index^2 | -1.237 | 0.145 | 1.16e-17 *** |
| Lake or bog site | -0.034 | 0.139 | 0.807 |
| Elevation | 0.003 | 0.001 | 0.007 ** |
| AP pollen:elevation interaction | -0.001 | 0.001 | 0.003 ** |
| SP pollen:elevation interaction | 0.004 | 0.001 | 0.003 ** |
| AP Shannon:elevation interaction | -0.004 | 0.001 | 0.001 ** |
| AP Shannon^2:elevation interaction | 0.002 | 0.000 | 3.05e-05 *** |
| Lake or bog site:elevation interaction | -0.001 | 0.000 | 3.32e-04 *** |
| | | | |
| **Precision submodel (log link; after variable selection^^)** | | | |
| (Intercept) | -2.827 | 0.988 | 0.004 ** |
| Wind pollination of AP% | 3.382 | 0.978 | 5.41e-04 *** |

| | | | |
|---|---|---|---|
| AP Shannon index | 0.938 | 0.116 | 4.84e-16 *** |
| Lake or bog site | 0.537 | 0.126 | 1.97e-05 *** |

Significance codes: 0 = '***';  0.001 = '**';  0.01 = '*'; 0.05 = '"' 0.1;  ' ' = 1
^^Only significant covariates were included (at 5% significance)

Similarly, including %Pinus within the model instead of %Needleleaf also slightly reduced the quality of the model fit, reducing the pseudo (Cox-Snell) R2 to 0.59 (vs 0.60). Although the LOOCV MAE (0.11) and RMSE (0.14) were approximately the same as the model with %needleleaf, the squared correlation (R2) of the predictions to the observations reduced slightly to 0.62 (from 0.63). The model coefficients are shown in Supplementary Table 4. This slight reduction in model fit supports the use of %needleleaf, although this marginal difference suggests that %Pinus could also be used and tested for other geographic contexts.

*Supplementary Table 2: Modern tree cover model coefficients, including %Pinus pollinated rather than %needleleaf*

| Coefficients (mean model with logit link) | Estimate | Standard Error | P Value |
|---|---|---|---|
| (Intercept) | -6.008 | 0.453 | 3.91e-40 *** |
| Tree pollen % | 2.428 | 0.222 | 1.47e-28 *** |
| Shrub pollen % | -3.539 | 0.629 | 1.82e-08 *** |
| *Pinus* of AP% | -0.010 | 0.005 | 0.04272 * |
| *Pinus* of AP%^2 | 0.000 | 0.000 | 2.77E-07 *** |
| AP Shannon index | 5.530 | 0.482 | 1.77e-30 *** |
| AP Shannon index^2 | -1.497 | 0.144 | 3.62e-25 *** |
| Lake or bog site | -0.039 | 0.133 | 0.76985 |
| Elevation | 0.002 | 0.001 | 0.045 * |
| AP pollen:elevation interaction | -0.001 | 0.000 | 0.029 * |
| SP pollen:elevation interaction | 0.004 | 0.001 | 0.002 ** |
| AP Shannon:elevation interaction | -0.003 | 0.001 | 0.002 ** |
| AP Shannon^2:elevation interaction | 0.001 | 0.000 | 3.65e-05 *** |
| Lake or bog site:elevation interaction | -0.001 | 0.000 | 0.002 ** |
| | | | |
| **Precision submodel (log link; after variable selection^^)** | | | |
| (Intercept) | 0.082 | 0.264 | 0.755 |
| *Pinus* of AP% | 0.013 | 0.003 | 8.62e-07 * |
| AP Shannon index | 0.959 | 0.128 | 5.98e-14 *** |
| Lake or bog site | 0.582 | 0.125 | 3.45e-06 *** |

Significance codes: 0 = '***';  0.001 = '**';  0.01 = '*'; 0.05 = '"' 0.1;  ' ' = 1
^^Only significant covariates were included (at 5% significance)

As far as we are aware, estimates of pollen fall speeds are not readily available beyond those published by Serge et al. (2023), and will depend on pollen size and density. The REVEALS approach makes use of FS and RPP estimates, and our method is an attempt to provide reconstruction estimates that do not require quite the same level of species level information.

I continue to have the following concerns with the presented manuscript:

**R2.** % needleleaf: The argument made in the manuscript really only applies to Pinus so I would suggest rerunning the analysis with only Pinus. Picea and Abies will not create large biases and Larix is a problem at the other extreme. I cannot see how combining these plus Taxus Cedrus or Juniperus should improve the reconstructions.

As discussed in our response to R1, the model containing %needleleaf very slightly outperforms that with %*Pinus* instead.

**R3.** Shannon index: Tree pollen diversity seems to have a strong influence in the final model. More efforts should be made to explain this and evaluating how that may influence the reconstructions e.g. running a model without SI for the past and evaluating the differences.

We have performed further analysis around the use of the Shannon index in the model. We divided the modelled data into Hengl's (2017) potential natural vegetation biomes and investigated whether there was a difference between the biomes and Shannon index values. At the biome level, the difference between observed tree cover and AP% is greatest in the tundra biome, and the Shannon index value for AP lowest. This implies that the Shannon index may be helping to adjust tree cover predictions towards observations, where high AP values may reflect longer distance transport rather than localised cover. We also investigated the impact of excluding the Shannon index on the subsequent reconstructions of tree cover. Although the general pattern of mid-Holocene increase in tree cover is observable, the timing of the tree cover peak, and early Holocene tree cover values are quite different. We have included this additional analysis within the manuscript and supplement.

L324. Increased tree SI is positively related to tree cover, with the effect decreasing with elevation. However, the negative correlation for the quadratic term for the SI suggests that the relationship has less of an effect on tree cover as tree SI increases. Again, this relationship may be explained in the context of open environments, where tree species diversity may be limited to species with longer distance pollen transport. For example, records in tundra tend to have a greater average disconnect between observed tree cover and AP%, as well as the lowest average tree SI values by biome (Supplementary Information: S4). Tree species diversity may then increase with tree cover, with the negative quadratic term implying that the highest levels of tree cover are represented by relatively uniform species types.

**S4: Modern tree species diversity and tree cover**

The tree pollen Shannon index (SI) is an important component of the regression model (Table 2). With increased tree cover, it is more likely that tree species diversity increases, although the negative second-degree polynomial (Table 1) suggests that at higher levels of tree cover this relationship becomes less important. To investigate this relationship, we divided the modelled modern data into biomes, and explore whether the SI for tree pollen varies for more open biomes. We divided the records into biomes according to Hengl's (2018) map of potential natural vegetation, extracting the modal biome with a 5km buffer around each record. We grouped some similar biomes, to simplify comparison; this grouping, and the division of modelled modern records, is shown in Supplementary Table 5.

*Supplementary Table 3: Biome groups for included modelled modern records. Parenthesis indicates grouped biomes*

| Biome group (incl. biomes) | Number of modern records |
|---|---|
| Cool evergreen needleleaf forest (Cold evergreen needleleaf forest; cool evergreen needleleaf forest) | 363 |
| Cool mixed forest | 255 |
| Temperate deciduous broadleaf forest | 117 |
| Warm temperate evergreen and mixed forest | 84 |
| Tundra (Low and high shrub tundra; Erect dwarf shrub tundra) | 29 |
| Xerophytic scrub and woodlands (Steppe; Xerophytic woods scrub) | 4 |

The distribution of arboreal pollen percentages and observed tree cover percentages are shown for each biome group in Supplementary Figure 1. In general, arboreal pollen percentages are greater than observed tree cover for each biome group, but the difference is most stark for the tundra biome (and xerophytic scrub and woodlands, although there are very few records). Supplementary Figure 2 shows the arboreal pollen Shannon index values for each group. The lowest median value and distribution is for tundra, suggesting that the higher pollen values compared to observed cover reflects inputs from a few species through long-distance transport. In fact, it is generally the case that the larger the difference between arboreal pollen and observed tree cover percentages, the lower the tree SI value. This highlights the importance of including this variable in the regression model.

[Figure]

*Supplementary Figure 1: Boxplots of arboreal pollen and observed tree cover by biome group*

[Figure]

*Supplementary Figure 2: Boxplots of Shannon index values for tree pollen for each biome group*

L433. Given the importance of the tree SI to the regression model (Table 2), we also ran the reconstructions based on a model excluding this variable. Although the reconstruction followed the same broad mid-Holocene increase and decline in tree cover, median tree cover prior to 7,000 cal. BP was much greater than shown in Fig. 5, with a less marked increase to the mid-Holocene (see Supplementary Information: S16). As shown in Supplementary Information 16, there is a slight increase in tree SI through time, which may imply a slight underestimation in tree cover during the early part of the Holocene.

**S15: AP Shannon index through time and implications for tree cover reconstructions**

The Shannon index (SI) value of arboreal pollen has a substantial impact on the quality of the model fit (Table 2). To investigate the implications of this variable for tree cover reconstructions, we re-ran the downcore reconstructions based on a model excluding tree cover SI (Supplementary Figure 9B) compared to the original model (Supplementary Figure 9A). Although the general mid-Holocene peak is visible in both reconstructions, there are differences between the two models. At the beginning of the Holocene, the median tree cover is around 20% higher when excluding tree SI. As a result, the increase in tree cover to the mid-Holocene is less dramatic. Additionally, the peak in tree cover occurs earlier, between ca. 9,000 and 7,000 cal. BP, compared to ca. 6,000 cal. BP when including the tree SI. In addition, the median tree cover values towards the present are around 5% lower when excluding tree SI.

[Figure]

*Supplementary Figure 3: A - Median reconstructed tree cover for Europe from 12,000 to 0 cal. BP, with 95% confidence intervals for 1000 bootstrap resampling of records; B - Median reconstructed tree cover for Europe from 12,000 to 0 cal. BP excluding AP Shannon index, with 95% confidence intervals for 1000 bootstrap resampling of records*

The tree SI values generally increased from the early Holocene to ca. 7,000 cal. BP (Supp. Fig. 10) and then remain relatively stable. Given that the tree SI has a positive relationship with observed tree cover in the modern regression model, this may imply that tree cover predictions for this earlier period of the Holocene are underestimated. However, since tree SI values are a means to correct for long distance transport, it is more likely that this implies there were more open environments earlier in the Holocene than is now the case.

[Figure]

*Supplementary Figure 4: AP Shannon index for Europe from 12,000 to 0 cal. BP, average for individual records binned in 200-year bins. Blue line is a LOESS line of best fit*

**R4.** Removal of landcover classes in the training data: may have lead to the underestimation of forest cleared for agriculture. Thus the reconstructions should be sensitive to climate induced changes in tree cover, while they may be less useful evaluating human induced deforestation. This should be more explored in the data and better communicated. If it is indeed that case that clearance for crop production is less readily detected this would render the data inappropriate for some interpretations and that should be clearly stated.

We tested the implications of not excluding cells from the tree cover map that had a majority of crop cover with the cell. There was a reasonable (negative) impact on the model fit, supporting our exclusion of these cells in the training data.

L.254. Finally, as excluding cells dominated by crop cover in the modern observed tree cover map may have implications for downcore reconstructions, we also tested the implications of removing this crop cover restriction when calculating observed tree cover on the regression model fit.

L. 371. We tested the influence of excluding cells dominated by crop cover when calculating observed tree cover values for each record location by re-running the regression using the same variables but without excluding these cells. The number of records used to train the model increased to 1050 (from 852), but the model pseudo (Cox-Snell) $R^2$ was reduced from 0.60 to 0.47 (Supplementary Information: S8), supporting out decision not to include these cells in the model training.

**S8: Excluding crop cells from the calculation of observed tree cover**

Cells dominated by crop cover (>50%) were excluded from the forest/tree cover map that was used to calculate observed tree cover for each modern pollen record. The rationale behind this methodological choice was to try to ensure that the relationship between tree cover and modern pollen was as representative as possible. Areas with high crop cover may affect the subsequent regression model, implying lower observed tree cover which is not reflected in the pollen record. To explore the implications of this exclusion, we re-ran the same regression model without excluding cells dominated by crop cover. The direction of the model coefficients (Supplementary Table 9) are the same as the standard regression model, except for whether the site was a bog or lake (which now becomes significant). However, some variables, such as the shrub pollen %, and the interactions between AP pollen and elevation, and SP pollen and elevation, are no longer significant. Note that although the needleleaf share of AP% is also insignificant, a model with orthogonal polynomials (i.e. reducing multicollinearity with the second polynomial of needle share of AP%) is significant. Additionally, the needleleaf share of AP% and whether the site is a lake or bog are no longer significant in the precision sub-model.

*Supplementary Table 4: Modern tree cover model coefficients, without exclusion of crop dominated cells when calculating observed tree cover*

| Coefficients (mean model with logit link) | Estimate | Standard Error | P Value |
|---|---|---|---|
| (Intercept) | -5.068 | 0.405 | 5.38e-36 *** |
| Tree pollen % | 2.199 | 0.186 | 3.01e-32 *** |
| Shrub pollen % | -0.731 | 0.508 | 0.150 |
| Needle share of AP% | -0.031 | 0.433 | 0.943 |
| Needle share of AP%^2 | 1.480 | 0.499 | 0.003 ** |
| AP Shannon index | 3.576 | 0.483 | 1.36e-13 *** |
| AP Shannon index^2 | -1.085 | 0.153 | 1.25e-12 *** |
| Lake or bog site | 0.386 | 0.095 | 5.04e-05 *** |
| Elevation | 0.002 | 0.001 | 0.046 * |
| AP pollen:elevation interaction | -0.001 | 0.000 | 0.110 |
| SP pollen:elevation interaction | 0.000 | 0.001 | 0.624 |
| AP Shannon:elevation interaction | -0.003 | 0.001 | 0.012 * |
| AP Shannon^2:elevation interaction | 0.001 | 0.000 | 6.74e-05 *** |
| Lake or bog site:elevation interaction | -0.001 | 0.000 | 6.70e-09 *** |
| | | | |

| Precision submodel (log link; after variable selection^^) | | | |
|---|---|---|---|
| (Intercept) | 1.459 | 0.230 | 2.4e-10 *** |
| Needle share of AP% | 1.191 | 0.205 | 0.351 |
| AP Shannon index | 0.406 | 0.108 | 1.74e-04 *** |
| Lake or bog site | -0.039 | 0.107 | 0.714 |

Significance codes: 0 = '***';  0.001 = '**';  0.01 = '*'; 0.05 = '"' 0.1;  ' ' =  1
^^Only significant covariates were included (at 5% significance)

The model that does not exclude cells with >50% crops includes more records (1050 compared to 852) but the pseudo (Cox-Snell) $R^2$ model fit was worse than the standard model (0.47 compared to 0.60), with LOOCV values similarly worse (MAE = 0.13 vs 0.11; RMSE = 0.16 vs 0.14; $R^2$ of predictions to observations = 0.50 vs 0.63).

**Line specific comments**

**R5.** L. 30: For a global application the % needleleaf would need to be evaluated.

We agree that the modelling approach would have to be tested in other regions. However, we do not feel that this statement in the abstract is controversial.

**R6.** L. 59: All the newly added references are no or no good examples of the prior statement. Hicks 2001 uses absolute pollen, Kaplan used plant functional types and Adam used the trends of change during the Lateglacial, a period for which nobody has yet dared to try quantitative reconstructions.

We thank the referee for this. We have amended the references as follows:

L50. The relative abundance of arboreal pollen has often been used to infer changes in tree abundance at a site (e.g. Thorley, 1981; Eastwood et al. 1999; Gil-Romera et al.).

Additional references:

Eastwood, W. J., Roberts, N., Lamb, H. F., and Tibby, J. C.: Holocene environmental change in southwest Turkey: a palaeoecological record of lake and catchment-related changes, Quat Sci Rev, 18, 671–695, https://doi.org/10.1016/S0277-3791(98)00104-8, 1999.

Gil-Romera, G., García Antón, M., and Calleja, J. A.: The late Holocene palaeoecological sequence of Serranía de las Villuercas (southern Meseta, western Spain), Veg Hist Archaeobot, 17, 653–666, https://doi.org/10.1007/s00334-008-0146-x, 2008.

Thorley, A.: Pollen Analytical Evidence Relating to the Vegetation History of the Chalk, J Biogeogr, 8, 93, https://doi.org/10.2307/2844552, 1981.

**R7**. L. 106: I continue to not see how the MAT approach would be more complicated than what is

proposed here. To my mind it even provides more information as it provides a dissimilarity measure to evaluate the trustworthiness of reconstructions.

We do not suggest that the MAT is more complicated, just that there are a number of arbitrary decisions that need to be made in the application of the technique, such as the choice of an appropriate number of analogues, the thresholds used for identifying analogues, and identifying non-analogue samples. To highlight the high-level differences between the techniques, we have included the following table:

L102. The key elements of each technique, including the approach applied within this paper, are summarised in Table 1.

**Table 1: Key elements of the reconstruction technique from this study, the REVEALS approach (i.e. Serge et al., 2023) and MAT (i.e. Zanon et al., 2018)**

|  | This paper | REVEALS (Serge) | MAT (Zanon) |
|---|---|---|---|
| **Training data** | Modern pollen data Modern tree cover | NA | Modern pollen data Modern tree cover |
| **Training model** | Regression based model linking modern pollen to modern tree cover | NA |  |
| **Downcore data** | Pollen data; Site characteristics | Pollen data; Site characteristics; RPP and FS per taxa | Pollen data |
| **Main Assumptions and Challenges** | Regression model applicability and included variables | Accuracy of RPP and FS values; Limited set of taxa | Number of analogues used (commonly 3-5); Threshold of similarity; Non-analogues |
| **Scale** | Site-based | Typically 1º where sites are located | Site-based; (Zanon: spatio-temporal interpolation) |

**R8**. L. 235: These numbers go way beyond Europe please check your calculations. From my own research the area around the lake generating the signal is much smaller than the theoretical source of 70% of all pollen (e.g. Matthias & Giesecke 2014 QSR).

We thank the reviewer for pointing out this error. The distances indicated should be in m, not km.

L194. Source area radii varied in size from 5,026m to 418,894m for the largest lake, with a median of 28,316m.

**R9.** L. 173: It needs to be clearly stated which (all) landcover classes were removed.

We have amended to the text to clarify this.

L126. A composite map of modern tree cover for the region 12°W to 45°E and 34–73°N was generated by averaging annual percentage forest/tree cover data from Copernicus annual land cover maps from 2015 to 2019 (Buchhorn et al., 2020a, e, d, c, b), after removing cells dominated (> 50%) by other land-cover classes, which include bare ground, built up areas, moss or lichen, permanent water, snow, and crops (Fig. 2A).

**R10**. L. 280 ff: The motivation of why to include %needleleaf is misleading. Since Larix is included in the Needleleaf group the spread is similar to that of broadleaf trees. Moreover, Table 1 from Serge et al., 2023 only includes a small part of the here included needleleaf trees and is also not using that label.

Please see our response to R1. Additionally, we have amended the text to highlight that the %needleleaf is attempting to capture both productivity and dispersion.

L170. To take into account broadscale differences in pollen productivity and pollen transport for species at a site level, the pollen data were also used to calculate the needleleaf share of the AP (%needleleaf)…

**R11**. L. 288: I am still not convinced by the reasons provided for the inclusion of the Shannon Index. Please explain how this should "account of potential impacts of very localised tree cover or long-distance transport influencing" and "Increased species diversity may reflect less fragmented landscape (Hill and Curran, 2003) and the likelihood that the recorded AP% reflects regional tree cover." Also note that the reference to Hill and Curran, 2003 is not relevant here as they were looking at trees in the landscape, while you look at a pollen count.

Please see R1 and the additional analysis performed that justifies the inclusion of this variable within the model. Additionally, we have amended the text, and removed the speculative wording around landscape fragmentation.

L244. We also included tree SI, to take account of potential impacts of very localised tree cover or long-distance transport from a single taxa or few taxon influencing the recorded AP%.

**R12**. L. 425: This is a result of modern forestry where even in the boreal forest you find monocultures in areas with highest forest cover. Could this parameter in the regression have caused lower tree cover in the early Holocene when tree composition was less divers?

It is true that forestry practices tend to create high cover of single species, but then some natural forest types such as beechwoods also have high cover of single species. As per R1, we acknowledge that lower Shannon tree cover values in the early Holocene could lead to somewhat lower values of tree

cover than may have been the case but also point out that this could be a realistic indication of more open vegetation at this time.

**R13**. L. 455: The influence of the Shannon index on the final model seems important and it is therefore important to understand the relationship in order to evaluate whether this is robust though time or a potential bias when reconstructing the early Holocene tree cover.
Please see our answer to R1

**R14**. L. 630: Thus by ignoring areas dominated by crops in the analysis of the modern data this approach may not detect the manmade reduction in tree cover.
Please see our answer to R4.

**R15**. L. 766: Please add the required acknowledgment: "Much of the fossil data were obtained from the Neotoma Paleoecology Database (http://www.neotomadb.org) and its constituent databases: European Pollen Database (EPD) and The Alpine Palynological Database (ALPADABA). The work of data contributors, data stewards, and the Neotoma community is gratefully acknowledged."
We have added this acknowledgement as requested. We have also added an acknowledgement to other palynologists who have contributed to the modern data set as follows:

Much of the fossil data were obtained from the Neotoma Paleoecology Database (http://www.neotomadb.org) and its constituent databases: European Pollen Database (EPD) and The Alpine Palynological Database (ALPADABA). The work of data contributors, data stewards, and the Neotoma community is gratefully acknowledged. We also thank members of the EMBSECBIO data community and other palynologists who have contributed records to the modern and fossil data set used here.

---

## Author Response (AR3)

**Point-by-point response to referee comments (4)**

We would like to thank the reviewer and editor for their continued help in improving this manuscript.

Our responses to individual comments are highlighted in blue, with any proposed changes highlighted in red. Note that red page numbers refer to the location in the updated marked-up manuscript.

**Editor comments**

**E1.** Line 55 - consider adding "such as basin size and type"

We have amended the text as follows:

L53. …. and site characteristics that affect the pollen source area, such as basin size and type (Bradshaw and Webb, 1985; Prentice and Webb, 1986; Prentice, 1988; Sugita, 1993).

**E2.** Table 1 - in my view would better fit under Chapter 2, where a subchapter could summarize these methodological baselines.

We have moved this into the Methods section. We have created a new sub-section, 2.4 Comparison of Reconstructions. We have moved the text describing the methods used to extract information from the Serge et al (2023) and Zanon et al. (2018) reconstructions into this new section. The text of the new section is as follows:

L295. Our new approach shares some features with the methods used in previous reconstructions (Table 1), but is less data-demanding than the REVEALS-based technique and does not require a priori decisions about the selection of analogues. We compare our final predictions to both modern and fossil reconstructions of tree cover by Serge et al. (2023) and Zanon et al. (2018). Modern is defined as the interval 100 cal. BP and the present day in Serge et al. (2023) and between 125 cal. BP and present for Zanon et al. (2018). We make comparisons to the Serge et al. (2023) reconstructions based on the 31 taxa originally used by Githumbi et al. (2022) since Serge et al. (2023) show that this produces better results than using the expanded data set of 46 taxa.

**Table 1: Key elements of the reconstruction technique from this study, the REVEALS approach (i.e. Serge et al., 2023) and MAT (i.e. Zanon et al., 2018)**

|                | This paper                                                    | REVEALS (Serge)                                                                               | MAT (Zanon)                                            |
| -------------- | ------------------------------------------------------------- | --------------------------------------------------------------------------------------------- | ----------------------------------------------------- |
| Training data  | Modern pollen data
Modern tree cover                       | NA                                                                                            | Modern pollen data
Modern tree cover               |
| Training model | Regression based model linking modern pollen to modern tree cover | NA, although defined relationship between RPP and FS per taxa from modern data underpins technique | Calibration of modern pollen assemblages to tree cover |

| Downcore data requirements | Pollen data; Site characteristics | Pollen data; Site characteristics; RPP and FS per taxa | Pollen data |
|---|---|---|---|
| **Main Assumptions and Challenges** | Regression model applicability and included variables | Accuracy of RPP and FS values; Limited set of taxa | Number of analogues used (commonly 3-5); Threshold of similarity; Non-analogues |
| **Scale** | Site-based | Typically 1º where sites are located | Site-based; (Zanon: spatio-temporal interpolation) |

For each of the 1º grid cells in Serge et al. (2023), tree cover was calculated from the sum of the appropriate vegetation types. Time series of the change in median tree cover were constructed using median tree cover corresponding to the pollen source area of each of our individual modern reconstructions. As the Serge et al. (2023) and Zanon et al. (2018) data is available in gridded format, comparison with our site-based predictions is not straightforward. Where the site location source areas straddled multiple grid cells, a median was calculated, weighted by the proportion of grid cell coverage using R package exactextractr (function: exact_extract) (Baston, 2023; version 0.10.0). The tree cover time series for the Zanon et al. (2018) and Serge et al. (2023) data were initially constructed using all of the extracted tree cover values for each of our model training site locations. However, since there can be multiple sites within some of these grid cells, we tested whether this affected the comparisons by taking an average of extracted tree cover values for locations sharing the same grid cell values from Zanon et al. (2018) or from Serge et al. (2023), and using this to create new time series for these two reconstructions.

**E3.** Can you also mention versions of R and packages used?

We have included R and package versions in the text and references.

**E4.** I think that the mismatch between the timing of tree cover increases in your data (Figs 7, 8, 9) and those of Serge (or Zanon) is sometimes 2000 years. You did not discuss sufficiently what was wrong with your or their approach.

Differences between the reconstruction methodologies, and the lack of primary (non-interpolated) data for previous reconstructions, make strict comparisons between the three approaches difficult or to diagnose what is wrong with the previous reconstructions. We have pointed out the differences, including for example the double peak in tree cover in the Zanon et al reconstruction, in the Results section (Lines 525 to 564). We have expanded the text in the Discussion to make it clear what the potential sources of these differences are, as follows:

L576. There are some differences between our reconstructions and those from other studies. Firstly, the maximum tree cover from our reconstructions is around 5–10% less than the maximum calculated from the other reconstructions. This could reflect the conservative nature of our modern-day tree cover model, which underestimates tree cover at the high end despite the application of quantitative mapping adjustment to model predictions. However, Zanon et al. (2018) also underestimate tree cover at high levels of tree cover. Alternatively, the difference may reflect the exclusion of higher elevation records from fossil dataset in order to minimise the impact of upslope pollen transport, which was not done in the other two reconstructions and would tend to reduce overall median tree cover. Secondly, the timings of peak tree cover vary between the reconstructions, with the MAT-based estimate peaking 2000 years earlier and the REVEALS estimate ca. 500 years later than shown in our reconstruction. The small difference between the peak timing shown by REVEALS and our reconstructions likely reflect differences in coverage through time and differences in the binning procedure. This may also partially explain the difference with the MAT-based reconstructions. However, this reconstruction also shows a marked decline between ca. 8000 and 7000 yr BP and a second peak in tree cover slightly after the peak shown by our reconstructions. It is difficult to assess the robustness of the reconstructed decline, which is strongly affected by a single point, but this would affect the overall shape of the curve and hence the timing of peak cover. Thirdly, the increase in tree cover in the early Holocene is less rapid in our reconstruction than the other reconstructions. This could reflect an underestimation of tree cover because of the lower tree SI values in our reconstructions, but the rapid increase in tree cover in the MAT- and REVEALS-based reconstructions is more likely to reflect an overestimation of tree cover because of long-distance pollen transport into the more open landscape characteristic of the early Holocene. The major difference at the pan-European scale is the reduction in tree cover from ca. 2000 cal. BP to present, which is less marked in our reconstructions and more consistent with observed tree cover. The observed tree cover values used in the model construction exclude areas dominated by land-cover types such as built areas or areas dominated by crops. We account for this in defining modern source areas in our model. Not accounting for changes in these other land-cover types, which through anthropogenic land use have increased substantially over the past 1000 years (Klein Goldewijk et al., 2017) would result in a steeper decline in tree cover, as seen in the other two reconstructions.

**Referee comments**

**R1.** I appreciate the effort of testing the influence of the pollination syndrome and the influence of Pinus instead of % Needleleaf pollen. I am surprised by the negative result, which leaves me puzzled as to why % Needleleaf pollen is the best predictor. Here and also for the Shannon index I would still

find it interesting to carefully discuss the results and reflect on how the modern situation may differ from the past for which this regression is applied.

We included a brief consideration of our assumption of stationarity between tree cover and the explanatory variables in the Discussion section, but we have now expanded this and commented on the importance of %needleleaf and species diversity as predictors, as follows:

L687. Our simple modelling approach yields a reasonably robust picture of changes in tree cover through the Holocene, largely consistent with known changes in climate. We have shown that both %needleleaf and the SI are important predictors of tree cover. These measures are, to some extent, surrogates for pollen productivity and factors affecting pollen transport distance as explicitly addressed in the REVEALS-based reconstructions. Thus, their importance in the model is not surprising and we have demonstrated that the final model is able to capture modern day patterns. However, our approach is predicated on the assumption of stationarity between tree cover and the explanatory variables through time, as indeed are the other statistical reconstruction techniques considered here. This may be problematic for variables such as elevation, where changes in elevational lapse rates (Mountain Research Initiative EDW Working Group, 2015) or atmospheric circulation patterns (Bartlein et al., 2017) could affect the relationship, but is less likely to be an issue for explanatory variables that reflect biophysical controls on pollen transport and deposition such as basin type or proportion of needleleaf trees. Tree diversity is somewhat more problematic, since the influence of long-distance transport into the more open landscapes likely to have been characteristic of the colder, drier climate at the start of the Holocene may not be adequately captured in the modern training data. The relatively slow rate of the initial increase in tree cover may be a reflection of this. This could be explored using macrofossil of sedimentary DNA data, to discriminate between local diversity and potential long-distance contamination of the SI index. Nevertheless, the overall pattern of changes in tree cover during the Holocene appear to robust and explicable, supporting the idea that modern relationships between tree cover and the explanatory variables provide a reasonable basis for reconstructions.

**R2.** Now there is a lot of information in the supplements which could be better integrated into the main manuscript.

We agree that there is a lot of information in the Supplements, but this documents the multiple additional tests that we have made in the light of the reviewers' comments. We have outlined these results in the appropriate place in the main text, but we think that it is appropriate to keep the methodological testing in the supplement, since these results are largely negative and this supports the key methodological decisions that we have made in making the reconstructions. This allows us to be

more confident about our results and conclusions, but we think including this material in the main text would obscure the main message.

**R3.** L. 52: Gil-Romera et al. – year missing

Thank you for noticing this omission, this has been amended.

**R4.** L. 105 ff: The table could be improved. Specifics: NA for training model MAT, RPP and FS per taxa is more similar to Training model then to downcore as these are assessed by modern situations.

We have revised Table 1 and moved this table into the Methods section (see response E2)

**R5.** L. 174: This is a posterior interpretation that SI may be sensitive to long distance cover no prior reasoning.

We agree and have modified this statement, since we have text in the Results section that discusses the impact of the Shannon Index on the model and how this could be interpreted in terms of either local species diversity or long-distance transport, as follows:

L175. We also calculated the Shannon Index (SI) of tree species diversity from the pollen data.

We have also modified the following sentence:

L251. We also included tree SI, as a way to evaluate potential impacts of very localised tree cover or long-distance transport from a single taxa or few taxon influencing the recorded AP%.

**R6.** L. 437: SI changes during the early Holocene due to the new establishment of trees generating a more divers forest with little change in the importance of long distance transported pollen.

The influence of arboreal SI in our model is based on the relationships between tree species diversity and tree cover in the modern day. In more open areas, longer distance pollen may upwardly bias tree cover values within the record. During the earlier Holocene, both the reconstructions by Serge and Zanon, as well as our reconstructions, suggest lower general tree cover. Increases in tree SI through time may therefore reflect both (1) a general increase in tree species diversity, and (2) a more open environment generally, which implies long distance transport may have a bigger impact on inflating tree cover values. Within the manuscript we cautiously suggest that tree cover in general may be slightly underestimated (i.e. suggesting that (1) may be more important), but in some areas the issue of long-distance transport may be more important, suggesting a slight overestimation of tree cover.

**Additional changes**

Please rename the supplementary figures according to the example: Supplementary Figure 1 -> Figure S1, etc.

We did not refer to the Supplementary Figures and Tables in the main text, but rather to the section dealing with each issue. We have re-named the sections in the Supplementary, and now refer to these in the main text as Supplementary Information, Section S1 etc. The figures and tables in the Supplementary Information are all labelled sequentially and have been renamed (Table S1, Figure S1 etc). We will list the figures and tables in each section at the beginning of the Supplementary Information file. We will also add references to these supplementary figures and tables in the appropriate place in the main text (e.g. Supplementary Information, Section S1, Table S1).